# String Seed of Thought: Prompting LLMs for Distribution-Faithful and Diverse Generation

**Kou Misaki, Takuya Akiba**
Sakana AI
{kou.misaki,takiba}@sakana.ai

## Abstract

We introduce *String Seed of Thought (SSoT)*, a novel prompting method for LLMs that improves *Probabilistic Instruction Following (PIF)*. We define PIF as a task requiring an LLM to select its answer from a predefined set of options, each associated with a specific probability, such that the empirical distribution of the generated answers aligns with the target distribution when prompted multiple times. While LLMs excel at tasks with single, deterministic answers, they often fail at PIF, exhibiting biases problematic for applications requiring non-deterministic behaviors, such as human-behavior simulation, content diversification, and multiplayer games. It also harms the diversity of generated responses, a crucial factor in test-time scaling, by causing the outputs to collapse into a limited set of answers. To address this, we propose SSoT, a simple prompting method that instructs an LLM to first output a random string to generate sufficient entropy. SSoT also instructs the LLM to extract randomness by manipulating this string to derive a final answer, thereby preserving diversity while adhering to specific constraints. We demonstrate that SSoT significantly improves the PIF performance of LLMs, approaching the ideal performance of a pseudo-random number generator. Furthermore, our experiments on NoveltyBench show SSoT's benefits extend beyond closed-set tasks to open-ended tasks by enhancing response diversity.

## 1 Introduction

While frontier LLMs excel on tasks with a single correct answer (Ouyang et al., 2022; Wei et al., 2022), some real-world applications require selecting from multiple acceptable options according to a specific distribution; Examples include human-behavior simulation (Park et al., 2023; Gao et al., 2024), content diversification (Padmakumar & He, 2024; Yu et al., 2025), and mixed strategy in games (Nash, 1951; Fan et al., 2024; Feng et al., 2025). For these applications, the primary evaluation criterion is not single-response accuracy but rather the alignment of the model's empirical choice frequencies with the target distribution. However, as we will demonstrate experimentally, even frontier LLMs struggle to satisfy this criterion. We specifically focus on challenges arising in the following two situations:

**Task 1: Probabilistic Instruction Following.** For instance, suppose prompting an LLM with the instruction, "Flip a fair coin and output Heads or Tails with equal probability", repeated 100 times. Ideally, the distribution of Heads and Tails would be close to 50-50. However, even state-of-the-art LLMs tend to yield skewed outputs when given a naive prompt (see Figure 1, top right) (Meister et al., 2025; Gu et al., 2025). More generally, we refer to scenarios requiring an LLM to select options from a closed set according to a desired distribution as *Probabilistic Instruction Following (PIF)*. In PIF, the options are explicitly provided, and the target distribution may be either explicitly described in the prompt or implicitly inferred by the LLM as the distribution most appropriate for the given context.

The importance of this task is underscored by its applications in several domains. For example, when an LLM plays a multiplayer game whose Nash equilibrium is in mixed strategies, the optimal behavior is to select actions according to the equilibrium mixing probabilities, which falls under PIF. Representing collective opinion distributions (Santurkar et al., 2023; Durmus et al., 2024; Meister et al., 2025) is another PIF instance, as they necessitate simulating the underlying distribution of views. Furthermore, as noted by Meister et al. (2025), LLMs often struggle to accurately simulate

## Probabilistic Instruction Following

**Standard Prompting**

| LLM Input |
| --- |
| Flip a fair coin and output Heads or Tails. |

| LLM Output |
| --- |
| Heads |

**SSoT Prompting**
(Proposed Method)

| LLM Input |
| --- |
| Generate a random string, and manipulate it to sample from the target distribution. Flip a fair coin and output Heads or Tails. |

| LLM Output |
| --- |
| <think>…**7$Aq9!zR@k3**…</think> Heads |

Distribution from 100 trials

## Diversity-Aware Generation

**Standard Prompting**

| LLM Input |
| --- |
| Write a one-sentence happy new year text to my friend. |

| LLM Output |
| --- |
| "Wishing you a joyful New Year filled with laughter, love, and wonderful adventures ahead!" |

**SSoT Prompting**
(Proposed Method)

| LLM Input |
| --- |
| Generate a random string, and manipulate it to generate a diverse response. Write a one-sentence happy new year text to my friend. |

| LLM Output |
| --- |
| <think>…**zT2!mX5%pL$**…</think> "May this year bring you bold success and unstoppable happiness beyond measure!" |

| NoveltyBench (Curated) | |
| --- | --- |
| Baseline | 4.70 (5.17) |
| **SSoT** | **6.19** (5.92) |

| NoveltyBench (WildChat) | |
| --- | --- |
| Baseline | 3.39 (4.08) |
| **SSoT** | **5.25** (4.86) |

Distinct ↑ (Utility ↑)

Figure 1: The schematic figure illustrating our method for PIF and diversity-aware generation in NoveltyBench, where we query the LLM multiple times using the same prompt and collect the resulting outputs. We used deepseek-r1 ($T = 0.6$) to obtain the results in the right panels. See Section 5.1 for PIF and Section 5.2 for diversity-aware generation experimental details.

a distribution even when they can describe it. Therefore, evaluating models on PIF with the target distribution being explicitly provided offers valuable insights into this fundamental limitation.

**Task 2: Diversity-Aware Generation.** Diverse LLM generation is essential in several situations. In open-ended tasks (e.g., inventing names, writing stories, or brainstorming new ideas), enhancing the diversity of generated outputs without compromising quality is important (Zhang et al., 2025). Increased answer diversity also benefits some test-time scaling methods, which generate numerous candidate solutions and select the most promising among them (Li et al., 2022; Wang et al., 2023; Brown et al., 2024; Schaeffer et al., 2025). We refer to such open-ended scenarios, where the goal is to generate a diverse set of solutions while preserving quality, as *Diversity-Aware Generation (DAG)*.

To address LLM performance limitations in these critical scenarios, we introduce *String Seed of Thought (SSoT)*, a novel yet simple prompting technique that is applicable to a wide range of LLMs. Applying SSoT merely requires adding an instruction to the prompt. For PIF, the instruction is "Generate a random string and manipulate it to sample from the target distribution" (Figure 1), while for DAG, it is "Generate a random string and manipulate it to generate one diverse response" (see Appendix A for full prompts). SSoT is highly versatile; Notably, a single, unified prompt framework resolves numerous challenges, requiring only minimal adjustments for each task category (i.e., PIF or DAG), without needing further modifications for individual tasks within a category. This versatility stems from the LLM's ability to autonomously select an optimal strategy for each task, as we demonstrate in our analysis (Section 5.3.1). Furthermore, the method is effective even for recent reasoning LLMs where the temperature parameter cannot be modified (OpenAI, 2025a;b).

Our extensive experiments demonstrate the effectiveness of SSoT. For PIF, SSoT substantially improves performance across five frontier LLMs (Section 5.1.1) and consistently outperforms strong baselines like prompt ensembling and few-shot examples (Section 5.1.2). We further showcase its practical utility in an adversarial Rock-Paper-Scissors game, where SSoT enables an LLM to play a mixed-strategy against pattern-exploiting bots, suggesting its potential for unexploitable strategies in realistic scenarios (Section 5.1.3). For DAG, experiments on NoveltyBench show SSoT generates more diverse responses than other baselines such as prompt paraphrasing or increasing temperature, without compromising output quality (Section 5.2). Our theoretical and empirical analyses illuminate the mechanism behind this success. We theoretically prove that SSoT ensures faithful sampling from the target distribution under mild assumptions (Section 4), and empirically reveal that LLMs achieve this by autonomously developing sophisticated internal strategies to manipulate the random string (Section 5.3.1), and demonstrate that PIF performance scales with their CoT length (Section 5.3.2).

**Contributions.** ① We introduce *String Seed of Thought (SSoT)*, a simple prompting technique that steers an LLM's probabilistic behavior by having it internally generate and process a random string. ② We demonstrate through extensive experiments that SSoT achieves sampling faithfulness in PIF tasks comparable to a PRNG, while boosting response diversity in DAG tasks. ③ We theoretically prove that the total variation distance to a target distribution diminishes with the length of the generated string, even when it contains autoregressive correlations. ④ We empirically show that LLMs autonomously select randomness extraction strategies, and that their performance scales with the length of the reasoning process.

## 2 RELATED WORK

**LLM Biases in Answer Selection from Specific Distributions.** A growing body of work shows that LLMs often struggle to select answers from specific probability distributions even when they can accurately describe them. Meister et al. (2025) introduce a distributional-alignment benchmark based on opinion-survey targets and find that models frequently miss the target distributions and are sensitive to output formats. Gu et al. (2025) evaluate both known- and unknown-distribution settings, reporting that while models can recognize probabilities, sampling accuracy lags behind inference of the probability distribution. Hopkins et al. (2023) show that, when prompted to generate random numbers, open-source models induce distributions far from uniform and exhibit high variance across prompts. For coin flips, Gupta et al. (2025) find that in-context sequences of flips steer predictions and, with enough evidence, models update in a Bayesian manner despite biased priors. Similarly, Van Koevering & Kleinberg (2024) document systematic, non-random patterns in LLM coin-flip behavior. Beyond coin flips, Lee (2024) investigate how LLMs distribute their choices among several possible options, revealing further evidence of inherent biases. Using Rock–Paper–Scissors as a semantic-free testbed, Anonymous (2025) find persistent gaps between verbalized target distributions and sampled actions, alongside order effects from permuting label sequences.

**Response Diversity Enhancement.** Recent work mitigates diversity collapse from post-training by explicitly optimizing for diversity while preserving quality. Approaches include: augmenting rewards with a learned diversity signal via RL (DARLING) (Li et al., 2025); using preference-based training to favor rare-but-good responses (DivPO) or to reweight preferences toward atypical high-quality continuations (Lanchantin et al., 2025; Chung et al., 2025); and decoupling the entropy bonus from the KL regularizer in preference learning to control lexical and semantic variety (Slocum et al., 2025).

**Relation to LLM Calibration.** While related to LLM calibration (Guo et al., 2017; OpenAI, 2023; Lovering et al., 2025), our work differs fundamentally. Calibration uses token probabilities to align confidence with accuracy on single-answer tasks, whereas our method performs actual sampling from an explicit target distribution over multiple acceptable answers.

## 3 METHODS

**String Seed of Thought.** Implementing SSoT simply requires adding an instruction to the prompt; this approach is effective whether applied to the user prompt or the system prompt. The SSoT prompt is a simple two-stage instruction (see Figure 1 for schematic illustration and Appendix A for full prompt): it first directs an LLM to (1) generate a random string, and then (2) use that generated string to select an action probabilistically. The core of this instruction varies slightly by task. For PIF, the prompt includes the direction to "Generate a random string, and manipulate it to sample from the target distribution," while for DAG, it is adapted to "Generate a random string, and manipulate it to generate one diverse response." Although these instructions form the core of SSoT, the full prompts are more detailed (see Appendix A). Despite its simplicity, SSoT effectively leverages a generated random string for probabilistic action selection and can also be applied to enhance response diversity.

**Intuition.** When an LLM selects an action probabilistically, biases can arise from sources like option position or label frequency in the training data. To mitigate these biases, SSoT first instructs the LLM to generate a random string. Since this step relies on a simple, task-agnostic instruction, "Generate a random string," it is less susceptible to the biases that appear in the case of action selection directly from the instruction. This process is designed to generate sufficient entropy for the subsequent choice and to ensure diversity across different generations. The subsequent step of mapping this string to an action can be accomplished through simple operations that are well within the capabilities

of an LLM, such as summation and mod operations, as we discuss in Section 4. Crucially, since each generation is independent, the SSoT framework is fully parallelizable. This offers a significant scalability advantage over sequential approaches that require access to the generation history.

# 4 THEORETICAL ANALYSIS

In this section, we provide an upper bound on the total variation distance between the empirical distribution from SSoT and the target distribution for PIF. We show that this distance can be made small if the string is sufficiently long and the character generation distribution is not strongly biased (even with autoregressive correlations in the LLM-generated string).

## 4.1 PROBABILISTIC INSTRUCTION FOLLOWING

**Preliminaries.** Key parameters influencing probabilistic behavior in LLMs include: (1) input prompt $t_{\text{in}}$, (2) temperature $T$, and (3) random seed $\epsilon$ for output generation. We denote the LLM's output generation process as a function $f_{T,\epsilon}$, dependent on temperature $T$ and random seed $\epsilon$. Given an input prompt $t_{\text{in}}$, the generated output is $t_{\text{out}} = f_{T,\epsilon}(t_{\text{in}})$.

**PIF Task Definition.** Suppose a task has a set of $m$ possible answers $\mathbf{a} = (a_1, \ldots, a_m)$, and an associated target probability distribution, $\mathbf{p} = (p_1, \ldots, p_m)$, where $p_i > 0$ and $\sum_{i=1}^m p_i = 1$. The distribution $\mathbf{p}$ may be explicitly provided in the prompt or implicitly inferred by the LLM, but it is assumed to be uniquely determined. We denote the instruction for such a task as $t_{\text{PIF}}$, which includes the specification of $\mathbf{p}$ when it is explicitly given. We refer to this general task of requiring an LLM to sample from a specific categorical distribution $(\mathbf{a}, \mathbf{p})$ as *probabilistic instruction following (PIF)*.

**How to evaluate PIF performance.** To perform evaluation, we invoke LLM $f_{T,\epsilon}$ with prompt $t_{\text{PIF}}$, $K$ times, each with a distinct random seed $\{\epsilon_k\}_{k=1}^K$. This produces outputs $t_{\text{out}}^k = f_{T,\epsilon_k}(t_{\text{PIF}})$, for $k = 1, \ldots, K$. Then each output text is parsed into actions using a parsing function $g$, yielding $K$ actions $\hat{a}_k = g(t_{\text{out}}^k)$. From these parsed actions, we construct an empirical distribution $\hat{P}_{\{\hat{a}_k\}_{k=1}^K}(i) = \sum_{k=1}^K I(\hat{a}_k = a_i)/K$, where $I$ is the indicator function. Performance assessment involves comparing the target distribution $\mathbf{p}$ with the empirical distribution $\hat{P}_{\{\hat{a}_k\}_{k=1}^K}$. To quantify deviations, we employ known statistical measures, including total variation distance, KL divergence, and JS divergence.

## 4.2 SSoT PERFORMANCE ANALYSIS ON PIF

**Notations and Setup.** Let the random string generated by the LLM be a sequence of $n$ characters $x^n = \{x_1, \ldots, x_n\}$, where each character is drawn from an alphabet of size $A$. Assume we sample $K$ times to compute the empirical distribution from $(x^n)_k$. We assume a uniform target distribution, but we can easily apply our analysis to a biased target distribution (see Appendix G.6). We denote the total variation distance between the probability distributions $P$ and $Q$ as $d_{TV}(P, Q) := \sum_i |P_i - Q_i|/2$, and $x_i \sim X_i$ when the sampled value of the probabilistic variable $X_i$ is $x_i$. We define the functions $\phi(x) := \frac{1}{1-2x} \ln \frac{1-x}{x}$ and $\pi_P := \max_{S \subseteq \mathbb{Z}_M} \min(P(S), 1 - P(S))$. We consider two strategies for randomness extraction from a random string, chosen to broadly cover the methods used by LLMs.

The following theorem uses 2-universal hash functions (Carter & Wegman, 1979) (see Appendix G.1 for the detailed definition) to extract entropy from the generated string:

**Theorem 4.1** (TV distance bound with random hash function (Informal)). *Suppose the random strings generated by the LLM, $(x^n)_k$, satisfies the condition that for each character $x_i \sim X_i$ and for some $\delta \leq 1/A$, the conditional probability is bounded as $\delta \leq P(x_i | \{x_j\}_{0 \leq j < i}) \leq 1 - (A-1)\delta$. Then, for a family of 2-universal hash functions $\mathcal{H} = \{h : X^n \to \mathbb{Z}_M\}$ and any real value $\delta', \delta'' > 0$, if we sample $h$ from $\mathcal{H}$ uniformly at random, the total variation distance between the empirical distribution and the uniform distribution $U_{\mathbb{Z}_M}$ satisfies the following with probability at least $1 - \delta' - \delta''$:*

$$d_{TV}(\hat{P}_{\{h((x^n)_k)\}_{k=1}^K}, U_{\mathbb{Z}_M}) \leq \frac{\sqrt{M}}{2\delta''} 2^{-\frac{n}{2} \log_2 \frac{1}{(1-(A-1)\delta)^2 + (A-1)\delta^2}} + \sqrt{\frac{\ln((2^M - 2)/\delta')}{K\phi(\pi_{P_X})}}. \quad (1)$$

*Sketch of proof (full proof in Appendix G.4).* The first term in Equation 1 follows from the Leftover Hash Lemma (Vadhan, 2012), since the random string satisfying the precondition is a $k$-source. The second term follows from an upper bound on the total variation distance between the empirical and the generation distributions (Weissman et al., 2003), combined with the triangle inequality. ☐

**Implications of Theorem 4.1.** This shows that even with correlations between characters in the generated string, a deterministic hash function can generate a near-uniform distribution. The hash function does not need to be different for each response; a simple 2-universal affine hash family over a prime field (Carter & Wegman, 1979) can be used, as its selection and application are simple enough for an LLM to perform. The first term is the bound achieved by extracting randomness via the hash function, while the second term represents the finite-sample error that arises from constructing an empirical distribution from a limited number of samples. Notably, the first term in Equation (1) shows that the upper bound on the total variation distance decreases as the string length $n$ increases. The second term shows that increasing the number of samples $K$ reduces the finite-sample error, bringing the empirical distribution closer to the uniform distribution.

The following theorem bounds a sum-mod strategy that we identified in the LLM's CoT analysis (Section 5.3.1). For simplicity, we assume $M$ is prime and $A \geq M$; the general case is discussed in Theorem G.9. We analyze an independent-source model to isolate the algebraic effect of the sum-mod operator and obtain a closed-form spectral bound. Extending the proof to weakly dependent sequences requires standard mixing assumptions, which introduce lengthy remainder terms.

**Theorem 4.2** (TV distance bound with sum-mod strategy (Informal)). *Let the random string $(x_1, \ldots, x_n)$ be drawn independently from a distribution $\eta_i$ over $\mathbb{Z}_M$. When $A > M$, we consider the distribution over $\mathbb{Z}_M$ by taking the values modulo $M$. Suppose we perform the sum-mod operation, i.e., taking the ASCII codes of the generated characters, summing them, and taking the result modulo $M$: $s_n^k = \sum_{j=1}^n ord(x_j) \mod M$, to select an action by its index. Then the total variation distance between the empirical distribution and the uniform distribution $U_{\mathbb{Z}_M}$ satisfies the following with probability at least $1 - \delta'$ for any real number $\delta' > 0$:*

$$d_{TV}(\hat{P}_{\{s_n^k\}_{k=1}^K}, U_{\mathbb{Z}_M}) \leq \frac{\sqrt{M-1}}{2} \prod_{i=1}^n (2d_{TV}(\eta_i, U_{\mathbb{Z}_M})) + \sqrt{\frac{\ln((2^M - 2)/\delta')}{K\phi(\pi_{P_X})}}. \quad (2)$$

*Sketch of proof (full proof in Appendix G.5).* The operation of summing the ASCII codes of the characters and taking the modulo $M$ of the result can be viewed as a random walk on $\mathbb{Z}_M$ (as a group with sum operation). Therefore, our analysis follows a similar line of reasoning to that of Diaconis & Shahshahani (1981). First, the total variation distance between the distribution of $s_n$ and the uniform distribution is upper-bounded by the L2-distance via the Cauchy-Schwarz inequality. The squared L2-norm of the probability distribution can, in turn, be bounded by the squared L2-norm of its Fourier transform using Plancherel's theorem. The norm of the Fourier transform can then be upper-bounded by the total variation distance between $\eta_i$ and the uniform distribution, leading to Equation 2. ☐

**Implications of Theorem 4.2.** The sum-mod strategy in this theorem is used by LLM in the PIF setting (see Section 5.3.1), so this theorem is directly relevant and gives insight into what is required to generate a faithful answer empirical distribution. The second term in Equation 2 is an unavoidable error from finite-size effect. The first term can be made small for a sufficiently long string if the generation distributions $\eta_i$ for most characters are not heavily biased, i.e., their total variation distance from the uniform distribution is less than $1/2$.

## 5 EXPERIMENTS

In this section, we first show SSoT improves PIF performance across various LLMs and target distributions. We then demonstrate its effectiveness in an adversarial Rock-Paper-Scissors game, where SSoT helps an LLM employ a mixed strategy. Next, we show that SSoT enhances diversity without compromising quality on the NoveltyBench benchmark for DAG. Finally, our CoT analysis reveals that LLMs adapt their strategies for our single, unified SSoT prompt, and we demonstrate that PIF performance scales with the length of the CoT. The full prompts used are listed in Appendix B.1.

Table 1: The PIF performance comparison of SSoT against the baseline across various models, evaluated using the JS divergence (lower is better). All the JS divergences are presented in units of $10^{-3}$ (original values multiplied by 1000).

| Model | Method | 2-choice | biased 2-choice | 3-choice | biased 3-choice | biased 9-choice |
|---|---|---|---|---|---|---|
| deepseek-v3 | Baseline | 5.97 ± 4.87 | 111.45 ± 9.38 | 136.03 ± 4.89 | 117.28 ± 0.00 | 297.33 ± 5.97 |
| | **SSoT** | 2.91 ± 3.08 (↓51%) | 3.54 ± 4.87 (↓97%) | 15.33 ± 16.13 (↓89%) | 15.65 ± 10.31 (↓87%) | 44.90 ± 11.88 (↓85%) |
| gpt-4o | Baseline | 15.56 ± 11.36 | 117.28 ± 0.00 | 60.55 ± 14.09 | 113.06 ± 13.35 | 290.25 ± 9.55 |
| | **SSoT** | 4.41 ± 3.39 (↓72%) | 9.59 ± 7.23 (↓92%) | 7.09 ± 5.36 (↓88%) | 9.75 ± 5.44 (↓91%) | 34.15 ± 11.99 (↓88%) |
| o4-mini-high | Baseline | 3.30 ± 3.49 | 86.12 ± 10.14 | 67.14 ± 13.11 | 115.53 ± 5.51 | 61.60 ± 10.64 |
| | **SSoT** | 0.94 ± 1.06 (↓71%) | 13.13 ± 6.86 (↓85%) | 10.34 ± 5.90 (↓85%) | 17.35 ± 8.32 (↓85%) | 18.67 ± 8.84 (↓70%) |
| QwQ-32B | Baseline | 2.43 ± 4.57 | 109.51 ± 10.03 | 104.64 ± 25.82 | 108.73 ± 12.50 | 260.59 ± 18.95 |
| | **SSoT** | 3.39 ± 4.97 (↑40%) | 2.47 ± 3.99 (↓98%) | 1.82 ± 1.44 (↓98%) | 1.30 ± 1.25 (↓99%) | 11.48 ± 5.27 (↓96%) |
| deepseek-r1 | Baseline | 36.09 ± 13.60 | 69.58 ± 13.42 | 106.30 ± 19.45 | 49.53 ± 11.24 | 138.21 ± 26.64 |
| | **SSoT** | 3.03 ± 3.43 (↓92%) | 1.51 ± 1.55 (↓98%) | 4.98 ± 3.84 (↓95%) | 4.30 ± 4.92 (↓91%) | 18.06 ± 11.35 (↓87%) |
| PRNG | | 1.85 ± 2.58 | 1.93 ± 2.80 | 3.36 ± 2.48 | 2.85 ± 3.15 | 13.72 ± 4.21 |

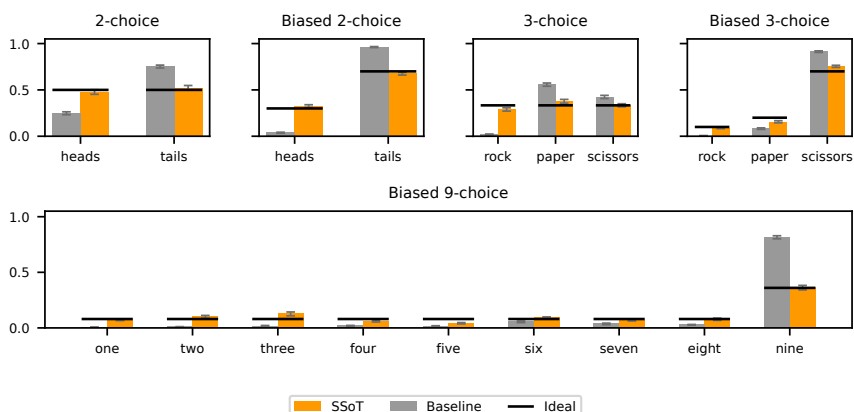

Figure 2: The PIF empirical distribution with baseline and SSoT prompts for deepseek-r1.

## 5.1 PROBABILISTIC INSTRUCTION FOLLOWING

### 5.1.1 PERFORMANCE ACROSS MULTIPLE LLMS

**Settings.** We evaluated the performance of SSoT in five PIF settings, repeating each experiment 10 times with $K = 100$ trials to calculate mean and standard deviation of the metrics: **(1, 2) 2-choice** and **Biased 2-choice**: ($\mathbf{a} =$ [heads, tails], $\mathbf{p} = [1/2, 1/2]$ and $[0.3, 0.7]$), **(3, 4) 3-choice** and **Biased 3-choice**: ($\mathbf{a} =$ [rock, paper, scissors], $\mathbf{p} = [1/3, 1/3, 1/3]$ and $[0.1, 0.2, 0.7]$), **(5) Biased 9-choice** ($\mathbf{a} =$ [one, two, . . . , eight, nine], $\mathbf{p} = [0.08, 0.08, . . . , 0.08, 0.36]$). We tested five frontier LLMs: deepseek-v3-0324 (DeepSeek-AI, 2024) ($T = 1.0$), gpt-4o-2024-08-06 (Hurst et al., 2024) ($T = 1.0$), o4-mini-high (OpenAI, 2025b) ($T$ unavailable), QwQ-32B (Qwen Team, 2024) ($T = 0.6$) and deepseek-r1-0528 (DeepSeek-AI, 2025) ($T = 0.6$), using recommended temperatures.

**Results.** Table 1 shows that SSoT substantially improves over baseline prompting across all models. Notably, deepseek-r1 and QwQ-32B, known for their very long reasoning trace (Sui et al., 2025), show significant improvements compared to the baseline, as shown in Figure 2. As we discuss in Section 5.3.2, we attribute this to the high complexity of the generated random strings, resulting from the extended reasoning process. To compare the results with the ideal case, we also generated actions using a pseudo-random number generator (PRNG) via `numpy.random`. We sampled 100 actions from the ground-truth distribution and repeated this 10 times with different seeds to calculate mean and standard deviation. Remarkably, the performance of deepseek-r1 and QwQ-32B with SSoT approached that of the PRNG, showcasing the outstanding performance of SSoT.

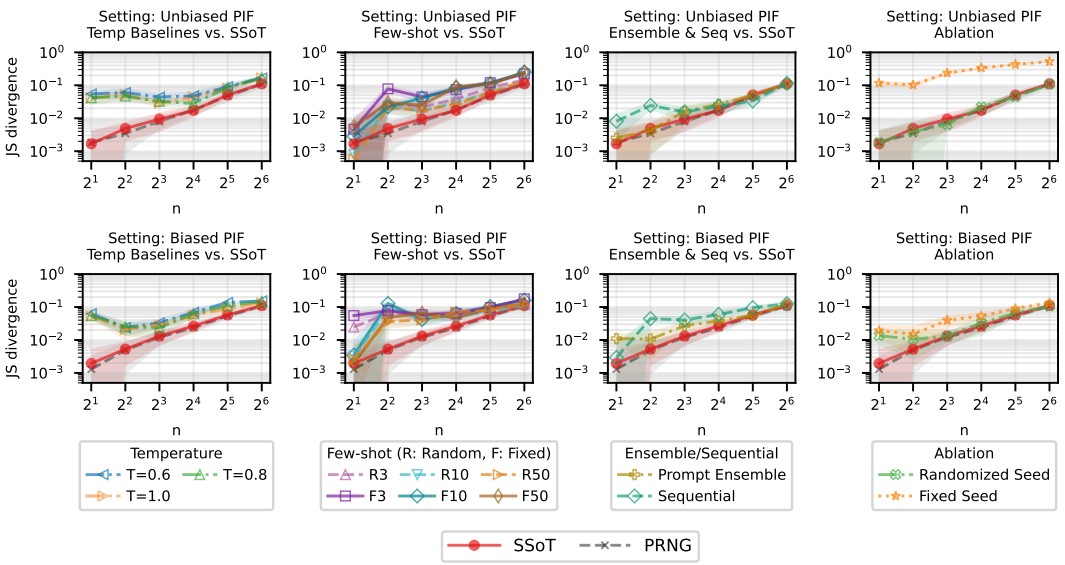

Figure 3: JS divergences for Unbiased and Biased PIF. Shaded areas represent the standard deviation.

### 5.1.2 VARYING ACTION SPACES

**Settings.** To compare SSoT against baselines on a wider range of tasks, we focused on deepseek-r1. We varied the number of actions ($2^n$ for $n = 1, \dots, 6$) under two conditions: **(1)** a uniform distribution, and **(2)** a biased distribution (for $n = 1$, $(0.75, 0.25)$; for $n > 1$, one action had a probability of $0.5$, with the rest distributed uniformly). We repeated each PIF experiment ($K = 100$) 10 times. Action words were randomly selected from a list of the 10,000 most common English words (excluding swear words) [1]. We compared SSoT against: **(1) High Temperature**: increasing the temperature setting; **(2) Few-shot**: providing $k \in \{3, 10, 50\}$ examples sampled using the target distribution in the prompt, either fixed or randomly sampled for each query; **(3) Prompt Ensemble**: using 50 paraphrased prompts, combined with randomized action orders for all the prompts; and **(4) Sequential Sampling**: generating 100 actions sequentially, including the selected action history in the prompt. As an ablation, we also tested providing an external random string to the LLM, either Fixed (same string each time) or Randomized (new string each time).

**Results.** Figure 3 shows that SSoT consistently delivers the best performance across all settings, nearly matching the PRNG. The strongest baseline was Prompt Ensemble, which performed well in the unbiased setting, suggesting that varying the prompt reduces positional biases. However, its performance degraded on the biased PIF tasks, indicating that eliminating positional bias alone is insufficient. Biased PIF presents a greater challenge than unbiased PIF. This is because it requires not only mitigating mode collapse to achieve uniformity, but also necessitates the reasoning ability to derive appropriate arithmetic operations to sample from the target distribution and the computational capability to execute them. In contrast, SSoT demonstrated near-ideal performance in both settings, supporting its robustness to distributional skew. Our ablation study revealed that providing a randomized external seed also improves performance, confirming that LLMs can effectively perform random action selection given a source of randomness. SSoT still outperformed the external randomized seed in the biased setting, likely because the LLM can internally generate strings of a length and type that are easier for it to manipulate, leading to more consistent performance.

### 5.1.3 ROCK-PAPER-SCISSORS IN ADVERSARIAL SETTING

**Motivation.** One scenario where probabilistic action selection is powerful is in achieving a mixed-strategy Nash equilibrium in game theory. By selecting actions probabilistically, a player can ensure their expected payoff remains constant regardless of the opponent's strategy, making it a robust defense in adversarial situations where an opponent actively tries to exploit patterns. We demonstrate

---

[1] https://github.com/first20hours/google-10000-english

Table 2: NoveltyBench results on curated dataset. Cells show Distinct (Utility); higher is better.

| Method | Creativity | Naming | Facts | Product Recs | Random | Opinions | Overall |
|---|---|---|---|---|---|---|---|
| Baseline | 4.60 (5.61) | 6.00 (6.13) | 4.14 (5.35) | 4.14 (6.02) | 6.07 (5.10) | 4.35 (4.08) | 4.70 (5.17) |
| Paraphrase | 5.15 (5.67) | 7.00 (6.77) | 5.46 (5.71) | **5.86 (7.33)** | 5.93 (5.48) | 5.57 (4.47) | 5.63 (5.57) |
| $T = 0.8$ | 4.95 (6.19) | 6.86 (7.06) | 4.50 (5.51) | 4.43 (5.86) | 6.20 (5.47) | 4.61 (4.42) | 5.03 (5.52) |
| $T = 1.0$ | 5.25 (6.20) | 7.57 **(7.75)** | 5.36 (6.40) | 5.43 (7.31) | 6.33 (5.46) | 5.04 **(4.92)** | 5.57 **(6.03)** |
| **SSoT** | **5.90 (6.44)** | **7.57** (6.62) | **6.04 (6.71)** | **5.86** (6.63) | **6.87** (5.49) | **5.87** (4.35) | **6.19** (5.92) |

that SSoT enables an LLM to defend itself in Rock-Paper-Scissors (RPS) against a strong opponent by making its actions faithfully probabilistic.

**Settings.** We evaluated the LLM against 10 "Black Belt" bots from the RPS Dojo[2], strong RPS agents available as a Kaggle Notebook. Each match consisted of 100 consecutive games, with a score of +1 for a win and -1 for a loss. We tested SSoT against a Baseline prompt (instructed to choose randomly without SSoT) and a Simple prompt (allowed to choose freely). To create an adversarial setup, the opponent bots had access to the full history of both players' moves, while the LLM did not.

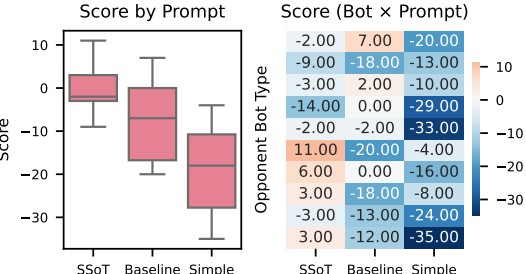

Figure 4: RPS results against black-belt bots.

**Results.** The final scores against the 10 bots are shown in Figure 4. As the left panel shows, SSoT maintained an average score near zero, effectively holding its own against bots designed to exploit patterns. The Baseline, while attempting to diversify its moves, still exhibited exploitable biases, resulting in lower scores. The Simple prompt lacked sufficient diversity and was consistently defeated. This demonstrates that SSoT can be used to achieve probabilistic behavior for improved game performance without relying on external tools.

## 5.2 SSoT FOR DIVERSITY-AWARE GENERATION

**Settings.** While an external PRNG is a plausible alternative for PIF, it is inadequate for DAG's core challenge of creatively mapping randomness, a process SSoT integrates internally. We used NoveltyBench (Zhang et al., 2025) to measure diversity in open-ended tasks. For each question in the curated and WildChat datasets, we generated eight responses to calculate the Distinct and Utility scores. We used deepseek-r1 ($T = 0.6$) and compared SSoT against a baseline (prompted to generate a diverse output without SSoT), a paraphrase method using prompts from the dataset, and higher temperatures ($T = 0.8, 1.0$). For the larger WildChat dataset, we evaluated SSoT and a baseline.

Table 3: Novelty-Bench results on Wild-Chat dataset. Cells show Distinct (Utility); higher is better.

| Method | Overall |
|---|---|
| Baseline | 3.39 (4.08) |
| **SSoT** | **5.25 (4.86)** |

**Results.** The results are presented in Table 2 (curated) and Table 3 (WildChat). In both datasets, SSoT achieves a higher Distinct score than all comparison methods. Notably, SSoT outperforms others on both Distinct and Utility scores for the "Creativity" category in the curated dataset, indicating that SSoT is useful for enhancing creativity in open-ended tasks.

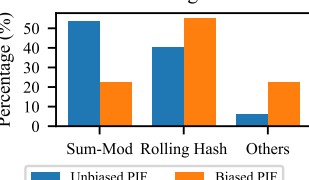

Figure 5: Randomness extraction strategy of LLM in PIF.

## 5.3 ANALYSIS

### 5.3.1 CoT STRATEGY ANALYSIS

Here, we analyze CoTs to classify the LLM's strategies. (Example CoTs in Appendices H.1 and H.2)

**Analysis of PIF in Section 5.1.2.** We analyzed how the LLM converts a generated random string into a random action, see Figure 5. We used gemini-flash-2.5 to classify 600 responses each from the biased and unbiased PIF experiments. As shown in Figure 5, the LLM's approach converges on

---

[2]https://www.kaggle.com/code/chankhavu/rps-dojo/notebook

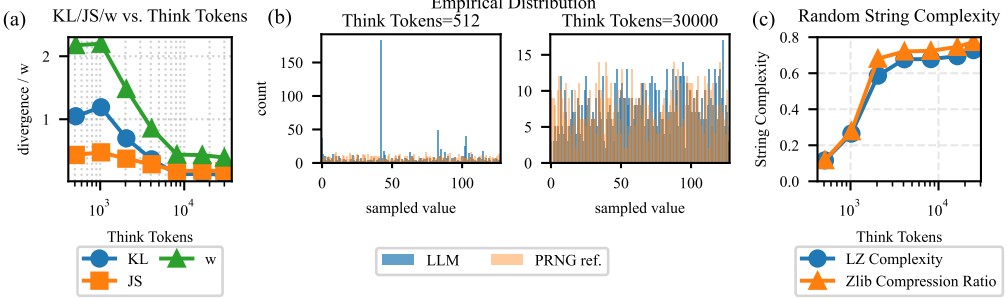

Figure 7: (a) Uniformness of LLM-generated integers (0–127) vs. think tokens, measured by KL/JS divergences and effect size $w$ ($N = 1000$). (b) Value distribution of LLM-generated integers for short (512, left) and long (30000, right) think token lengths, compared to a PRNG reference (orange). (c) Normalized LZ complexity and zlib compression rate of a random string generated sequentially by an LLM at $T = 0$ vs. think tokens. We used 3000-character prefix of 100 generated strings.

two main strategies (see Appendix I.3 for detailed definitions): **(1) Sum-Mod**: Sums the character codes (e.g., ASCII) of the random string and takes the modulo of the result. **(2) Rolling Hash**: Uses a base $B$ to compute a polynomial hash of the string ($\sum_i B^i \text{ord}(c_i)$)) and takes the modulo of that value. Notably, the LLM adapts its strategy to the task: for unbiased distributions, it favors the simple Sum-Mod method, but for biased distributions, it often adopts the more sophisticated rolling hash, demonstrating that it adjusts its approach based on the problem's complexity.

**Analysis of DAG in Section 5.2.** To understand how SSoT enhances diversity in DAG tasks, we classified the CoT strategies from all 800 responses on the NoveltyBench curated dataset, see Figure 6. We analyzed two axes (for detailed definitions, see Appendix I.3): **(1) Assembly**: how the answer is constructed (from a fixed `List` or by filling a `Template`), and **(2) Sampling Scope**: whether randomness is used once for a `Global` choice or repeatedly for `Local` elements. As shown in Figure 6, the overall trend is to select from a `List` with a single `Global` sample. This trend reverses for the "Creativity" category: The LLM tends to create a `Template` and repeatedly samples from the random string to fill in different `Local` elements. This suggests that for open-ended tasks, SSoT enables diversity enhancement by decomposing the problem and diversifying each component.

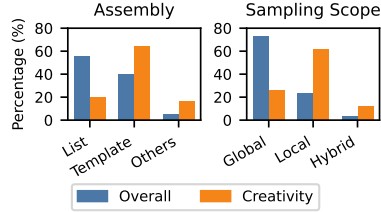

Figure 6: Diversity enhancement strategy of LLM in NoveltyBench.

### 5.3.2 CoT Scaling Analysis

**Analysis of Randomness Quality.** To further analyze the success of SSoT for LLMs with long CoT, such as deepseek-r1 and QwQ-32B, we examined the impact of CoT length on PIF performance using s1.1-32B (Muennighoff et al., 2025). We controlled the reasoning length via budget forcing and had the model generate random integers between 0 and 127. Figure 7(a) shows that as the reasoning length (thinking tokens) increases, the uniformity of the generated integers improves significantly, as measured by KL/JS divergence and effect size $w$. This enhanced uniformity is visualized in Figure 7(b). These results highlight the direct link between CoT length and the performance of PIF.

**Analysis of Random String Complexity.** To isolate the effect of reasoning on the generated string itself, we evaluated the complexity of strings generated sequentially at $T = 0$, which also removes randomness from the decoding process. We prompted the LLM to generate 100 strings in sequence, using budget forcing to control the thinking token length for each. We then measured the complexity of the concatenated string using normalized Lempel-Ziv complexity and zlib compression ratios. Figure 7(c) shows that string complexity grows with longer reasoning traces. This indicates that LLMs can produce intricate, high-entropy strings through their internal reasoning process alone, even without stochasticity at the decoding stage, and also for longer reasoning traces, the complexity of the generated string increases, thereby leading to better performance on PIF.

# 6 CONCLUSION

In this paper, we introduced *String Seed of Thought (SSoT)*, a simple and broadly applicable prompting technique that significantly improves the ability of LLMs to follow probabilistic instructions (PIF) and generate diverse responses (DAG). We demonstrated that SSoT enables models to achieve near-ideal probabilistic faithfulness by leveraging their own reasoning process to create an internal random seed, which is then deterministically mapped to a final action. Our analysis confirmed that the quality of this internal randomness scales with the length of the model's reasoning trace.

SSoT offers a practical, tuning-free method for enhancing LLM reliability in applications requiring strategic randomness or creativity, such as human-behavior simulations, content diversification, and game-playing. While the approach is most effective on highly capable models, it opens promising avenues for future work. These include applying SSoT to more complex domains and optimizing the string seed generation and its randomness extraction strategies.

# 7 LIMITATIONS

While SSoT proves effective across various domains, we identify three key limitations:

**Dependence on Reasoning Capability**: SSoT relies on the model's ability to autonomously devise and execute a mapping strategy (e.g., modulo arithmetic or hashing). As shown in our analysis of smaller models (Appendix D.3), models with limited reasoning capabilities (typically less than 8B parameters) may fail to execute these strategies correctly, leading to suboptimal performance.

**Potential for Bias Propagation**: If the generated random string exhibits strong positional bias (e.g., always starting with the same digit) and the model adopts a "lazy" strategy without leveraging the entropy from the whole string (e.g., using only the first character), the output distribution will be biased (as seen in the QwQ-32B failure case, Appendix D.5). This can be mitigated by steering the model towards more robust strategies (like rolling hashes) via system prompts.

**Applicability to Single-Answer Tasks**: SSoT is designed for tasks with multiple valid answers or probabilistic requirements. Applying SSoT to tasks with a single correct answer (e.g., math problems, factual retrieval) is not effective and could potentially distract the model, though our results on NoveltyBench suggest that utility is generally preserved.

## REPRODUCIBILITY STATEMENT

An anonymized zip file (see the supplementary materials) includes all the evaluation scripts and configuration files to reproduce all the experiments in this paper.

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

APPENDIX CONTENTS

# A  SSoT System Prompts

Here we list the full SSoT prompts used in our experiments. For SSoT, we simply used the following prompts as the system prompt in all the experiments [3].

---

**Listing A.1: SSoT System Prompt (Probabilistic Instruction Following)**

```
You are a helpful AI Assistant designed to provide well-reasoned
and detailed responses. If the task involves probabilistic or non-
deterministic reasoning, you must begin by generating a unique and
complex random string to serve as a seed.

This random string should appear sufficiently complex and
unpredictable, with no obvious structure or pattern. Use your
judgment to ensure it looks arbitrary and unguessable.

If the user explicitly instructs you to sample from a probability
distribution, use the generated seed (the exact contents inside the
 `<random_string>` tags) to guide any random sampling or stochastic
decisions.

Follow these two steps for every instruction:

1. Output the random seed string enclosed within `<random_string>`
and `</random_string>` tags.
2. Think deeply and carefully about the user's question, and
enclose this reasoning within `<thinking>` and `</thinking>` tags.
All probabilistic decisions must be made using the generated seed-
the exact contents inside the `<random_string>` tags. Make sure to
extract maximum randomness from the string by using all of its
content.
3. Provide your final answer, enclosed within `<answer>` and `</
answer>` tags.

Strictly follow this tag structure, and respond in the following
format:
<random_string>
...
</random_string>
<thinking>
...
</thinking>
<answer>
...
</answer>
```

---

[3]We replaced — with --- in the prompts due to formatting issues.

```
Listing A.2: SSoT System Prompt (Rock-Paper-Scissors)

You are a helpful AI Assistant designed to provide well-reasoned
and detailed responses. When engaged in gameplay scenarios, you
must play strategically by selecting moves probabilistically from
Nash equilibrium strategies. Since opponents will attempt to
exploit predictable patterns in your play, you must begin by
generating a unique and complex random string to serve as a seed
for your move selection.

This random string should appear sufficiently complex and
unpredictable, with no obvious structure or pattern. Use your
judgment to ensure it looks arbitrary and unguessable.

When playing any game, use the generated seed (the exact contents
inside the `<random_string>` tags) to guide your probabilistic move
selection from the Nash equilibrium mixed strategy.

Follow these steps for every game situation:

1. Output the random seed string enclosed within `<random_string>`
and `</random_string>` tags.
2. Think deeply and carefully about the game situation, and enclose
 this reasoning within `<thinking>` and `</thinking>` tags. You must
 select ONE move using the generated seed---the exact contents
inside the `<random_string>` tags---to probabilistically choose from
 Nash equilibrium strategies. Make sure to extract maximum
randomness from the string by using all of its content.
3. Provide your final move/action, enclosed within `<answer>` and
`</answer>` tags.

Strictly follow this tag structure, and respond in the following
format:
<random_string>
...
</random_string>
<thinking>
...
</thinking>
<answer>
...
</answer>
```

Listing A.3: SSoT System Prompt (Diversity-Aware Generation)

```
You are a helpful AI Assistant designed to provide well-reasoned
and detailed responses. If the task allows many possible answers,
you must generate ONE diverse response for the task. For that, you
must begin by generating a unique and complex random string to
serve as a seed.

This random string should appear sufficiently complex and
unpredictable, with no obvious structure or pattern. Use your
judgment to ensure it looks arbitrary and unguessable.

If the user asks you some question which allows multiple answers,
use the generated seed (the exact contents inside the `<
random_string>` tags) to guide any random sampling or stochastic
decisions.

Follow these steps for every instruction:

1. Output the random seed string enclosed within `<random_string>`
and `</random_string>` tags.
2. Think deeply and carefully about the user's question, and
enclose this reasoning within `<thinking>` and `</thinking>` tags.
You have to generate ONE response leveraging the generated seed---
the exact contents inside the `<random_string>` tags, to ensure your
 single answer is unique and diverse. Make sure to extract maximum
randomness from the string by using all of its content.
3. Provide your final answer, enclosed within `<answer>` and `</
answer>` tags.

Strictly follow this tag structure, and respond in the following
format:
<random_string>
...
</random_string>
<thinking>
...
</thinking>
<answer>
...
</answer>
```

Listing A.4: SSoT System Prompt (Random Integers Generation)

```
You are a helpful AI Assistant designed to generate random data
based on instructions. When asked to generate random data, you must
 first generate a unique and complex random string to serve as a
seed or source of randomness.

This random string should appear sufficiently complex and
unpredictable, with no obvious structure or pattern. Use your
judgment to ensure it looks arbitrary and unguessable.

Use the generated seed (the exact contents inside the `<
random_string>` tags) to guide any subsequent random choices, like
generating a random integer.

Follow these steps for the response format:

1. Output the random seed string enclosed within `<random_string>`
and `</random_string>` tags.
2. Perform the requested random generation task (e.g., generating a
 random integer within a specified range). Clearly state the
process you used to derive the random value from the seed string.
3. Provide the final generated random value (e.g., the integer)
enclosed within appropriate tags (e.g., `<random_integer>` and `</
random_integer>`).

Strictly follow this tag structure.
```

Listing A.5: SSoT System Prompt (Sequential Random Strings Generation)

```
You are a helpful AI Assistant designed to generate random data
based on instructions. When asked to generate random data, you must
 first generate a unique and complex random string to serve as a
seed or source of randomness.

This random string should appear sufficiently complex and
unpredictable, with no obvious structure or pattern. Use your
judgment to ensure it looks arbitrary and unguessable.

Use the generated seed (the exact contents inside the `<
random_string>` tags) to guide any subsequent random choices.

Follow these steps for the response format:

1. Output the random seed string enclosed within `<random_string>`
and `</random_string>` tags.
2. Perform the requested random generation task (e.g., generating a
 random integer within a specified range). Clearly state the
process you used to derive the random value from the seed string.
Strictly follow this tag structure.
```

# B PROMPTS USED IN EXPERIMENTS

## B.1 PROBABILISTIC INSTRUCTION FOLLOWING

---

**Listing B.1: User Prompt (PIF)**

```
Please choose between {choices}. You must select one of these {
num_choices} options with the following probabilities: {
prob_distribution}.
```

---

**Listing B.2: Baseline System Prompt (PIF)**

```
You are a helpful AI Assistant designed to provide well-reasoned
and detailed responses. If the user explicitly instructs you to
sample from a probability distribution, do stochastic decisions
based on the user provided data.
Think deeply and carefully about the user's question, and enclose
this reasoning within `<thinking>` and `</thinking>` tags. Then
provide your final answer, enclosed within `<answer>` and `</answer
>` tags.

Strictly follow this tag structure, and respond in the following
format:
<thinking>
...
</thinking>
<answer>
...
</answer>
```

---

In PIF experiments in Sections 5.1.1 and 5.1.2, we used the system prompt as shown in Listing A.1. The system prompt instruction consists of three components: (1) Generation of a random string, in case user's task requires it; (2) Generation of a thought process; (3) Generation of a final answer inside <answer> and </answer> tags. The final answer will be parsed from the generated output in a rule-based manner.

As for the baseline method, we used the system prompt shown in Listing B.2. The baseline prompt instruction consists only of (1) Generation of a chain of thought; (2) Generation of a final answer inside <answer> and </answer> tags. We also included a thought process generation in the baseline prompt to gauge the pure effect of random string generation on the probabilistic task.

As for the user prompt, we used a simple prompt shown in Listing B.1. Here in the boxes surrounded by { and }, we used strings suited to each task.

## B.2 ROCK-PAPER-SCISSORS

For the Rock-Paper-Scissors experiment in Section 5.1.3, we used the SSoT system prompt from Listing A.2, the simple prompt from Listing B.3, and the simple prompt from Listing B.4. The user prompt is given in Listing B.5.

## B.3 DIVERSITY-AWARE GENERATION

For the DAG task in Section 5.2, we used the SSoT system prompt in Listing A.3, the Baseline system prompt in B.6. As for the user prompts, we used the ones provided by the NoveltyBench dataset as it is.

```
Listing B.3: Simple System Prompt (Rock-Paper-Scissors)

You are a helpful AI Assistant designed to provide well-reasoned
and detailed responses. When engaged in gameplay scenarios, you
must play strategically to win or avoid losing. Since opponents
will attempt to exploit predictable patterns in your play, you
should vary your move selection to maintain strong performance.

When playing any game, select moves that give you the best chance
of winning or avoiding defeat.

Follow these steps for every game situation:

1. Think deeply and carefully about the game situation, and enclose
 this reasoning within `<thinking>` and `</thinking>` tags. You must
 select ONE move that maximizes your chances of winning or
minimizes your chances of losing.
2. Provide your final move/action, enclosed within `<answer>` and
`</answer>` tags.

Strictly follow this tag structure, and respond in the following
format:
<thinking>
...
</thinking>
<answer>
...
</answer>
```

## B.4    RANDOM INTEGER GENERATION

As for the random integer generation experiment in Section 5.3.2, we used the system prompt in Listing A.4 and the user prompt in Listing B.7.

## B.5    SEQUENTIAL RANDOM STRING GENERATION

As for the sequential random string generation in Section 5.3.2, for the first turn, we used the system prompt in Listing A.5 and the user prompt in Listing B.8. For the follow-up request to generate a new string, we used the user prompt in Listing B.9 with the same system prompt, Listing A.5. To avoid hitting the token length limit, we collected the previous turns' random strings and put them in the user prompt, rather than giving the whole conversation history.

Listing B.4: Baseline System Prompt (Rock-Paper-Scissors)

```
You are a helpful AI Assistant designed to provide well-reasoned
and detailed responses. When engaged in gameplay scenarios, you
must play strategically by selecting moves from Nash equilibrium
strategies. Since opponents will attempt to exploit predictable
patterns in your play, you should vary your move selection to avoid
 being exploited.

When playing any game, select moves based on Nash equilibrium mixed
 strategies to maintain optimal play against adaptive opponents.

Follow these steps for every game situation:

1. Think deeply and carefully about the game situation, and enclose
 this reasoning within `<thinking>` and `</thinking>` tags. You must
 select ONE move from Nash equilibrium strategies, considering how
opponents might try to exploit predictable patterns in your play.
2. Provide your final move/action, enclosed within `<answer>` and
`</answer>` tags.

Strictly follow this tag structure, and respond in the following
format:
<thinking>
...
</thinking>
<answer>
...
</answer>
```

Listing B.5: User Prompt (Rock-Paper-Scissors)

```
You are playing rock, scissors, and paper against other agent. You
need to pick a move from rock, scissors or paper.
First provide your reasoning, and then pick your answer inside <
answer></answer> tags.
```

Listing B.6: Baseline System Prompt (Diversity-Aware Generation)

```
You are a helpful AI Assistant designed to provide well-reasoned
and detailed responses. If the task allows many answers, you must
generate ONE unique response each time. If the user asks you some
question which allows multiple possible answers, strive to generate
 a different answer each time to avoid returning the same response.

Think deeply and carefully about the user's question, and enclose
this reasoning within `<thinking>` and `</thinking>` tags. Then
provide your final answer, enclosed within `<answer>` and `</answer
>` tags.

Strictly follow this tag structure, and respond in the following
format:
<thinking>
...
</thinking>
<answer>
...
</answer>
```

Listing B.7: User Prompt (Random Integers Generation)

```
Your task is to generate a random integer between 0 and 127 (
inclusive).

Follow these steps precisely:
1. First, generate a unique and complex random string. Output this
string within `<random_string>` tags.
2. Based *only* on the random string you generated, choose a random
 integer between 0 and 127 (inclusive). Explain how you derived
this integer from the random string.
3. Output the final chosen integer enclosed within `<random_integer
>` and `</random_integer>` tags.

Provide your response strictly following the required format:
1. Output the random string in the tag `<random_string>`.
2. Explain your process for deriving the integer from the string.
3. Output the final integer in the tag `<random_integer>`.
```

Listing B.8: User Prompt; 1st turn (Sequential Random Strings Generation)

```
Your task is to generate a random string. Generate a unique and
complex random string. Output this string within `<random_string>`
tags.
```

Listing B.9: User Prompt; new turns (Sequential Random Strings Generation)

```
Your task is to generate a random string. Generate a unique and
complex random string. Output this string within `<random_string>`
tags.

You generated random strings in the previous turns. Please generate
 a new random string.

Previous Random Strings:
{random_string_history}
```

Table 4: The PIF performance comparison of SSoT against the baseline across various models, evaluated using the KL divergence. We generated 100 actions for each configuration to calculate the empirical distribution, and then calculated the divergences. We repeated it 10 times to calculate the standard deviation. All the KL values are presented in units of $10^{-3}$ (original values multiplied by 1000).

| Model | Method | 2-choice | biased 2-choice | 3-choice | biased 3-choice | biased 9-choice |
|---|---|---|---|---|---|---|
| deepseek-v3 | Baseline | $23.66 \pm 19.19$ | $342.42 \pm 22.96$ | $423.83 \pm 23.88$ | $356.67 \pm 0.00$ | $1013.46 \pm 17.27$ |
| | SSoT | $11.57 \pm 12.23$ (↓51%) | $13.51 \pm 18.16$ (↓96%) | $60.93 \pm 64.94$ (↓86%) | $56.54 \pm 35.56$ (↓84%) | $179.18 \pm 47.56$ (↓82%) |
| gpt-4o | Baseline | $60.82 \pm 42.93$ | $356.67 \pm 0.00$ | $213.37 \pm 46.53$ | $345.74 \pm 34.57$ | $992.99 \pm 27.62$ |
| | SSoT | $17.52 \pm 13.44$ (↓71%) | $35.99 \pm 26.50$ (↓90%) | $27.99 \pm 21.05$ (↓87%) | $35.93 \pm 19.30$ (↓90%) | $130.95 \pm 46.73$ (↓87%) |
| o4-mini-high | Baseline | $13.10 \pm 13.84$ | $277.53 \pm 27.78$ | $237.26 \pm 43.31$ | $352.33 \pm 13.75$ | $221.58 \pm 46.50$ |
| | SSoT | $3.75 \pm 4.21$ (↓71%) | $49.03 \pm 24.70$ (↓82%) | $40.69 \pm 23.97$ (↓83%) | $63.15 \pm 28.70$ (↓82%) | $71.15 \pm 33.33$ (↓68%) |
| QwQ-32B | Baseline | $9.61 \pm 18.01$ | $337.66 \pm 24.54$ | $414.00 \pm 96.83$ | $334.61 \pm 32.81$ | $904.49 \pm 56.77$ |
| | SSoT | $13.42 \pm 19.55$ (↑40%) | $9.45 \pm 15.12$ (↓97%) | $7.27 \pm 5.81$ (↓98%) | $5.21 \pm 5.03$ (↓98%) | $45.65 \pm 20.26$ (↓95%) |
| deepseek-r1 | Baseline | $138.64 \pm 50.74$ | $230.75 \pm 39.11$ | $345.73 \pm 53.83$ | $162.34 \pm 34.78$ | $504.28 \pm 89.38$ |
| | SSoT | $12.04 \pm 13.60$ (↓91%) | $6.09 \pm 6.25$ (↓97%) | $19.73 \pm 15.05$ (↓94%) | $16.04 \pm 17.85$ (↓90%) | $72.29 \pm 47.84$ (↓86%) |
| PRNG | | $7.35 \pm 10.22$ | $7.57 \pm 10.84$ | $13.25 \pm 9.61$ | $11.17 \pm 12.24$ | $53.86 \pm 17.12$ |

Table 5: The PIF performance comparison of SSoT against the baseline across various models, evaluated using the total variation distance. We generated 100 actions for each configuration to calculate the empirical distribution, and then calculated the divergences. We repeated it 10 times to calculate the standard deviation. All the total variation distance values are presented in units of $10^{-2}$ (original values multiplied by 100).

| Model | Method | 2-choice | biased 2-choice | 3-choice | biased 3-choice | biased 9-choice |
|---|---|---|---|---|---|---|
| deepseek-v3 | Baseline | $9.80 \pm 4.80$ | $29.70 \pm 0.48$ | $33.57 \pm 0.74$ | $30.00 \pm 0.00$ | $63.80 \pm 0.42$ |
| | SSoT | $6.40 \pm 4.27$ (↓35%) | $6.20 \pm 3.88$ (↓79%) | $14.30 \pm 7.56$ (↓57%) | $11.20 \pm 3.08$ (↓63%) | $24.50 \pm 4.22$ (↓62%) |
| gpt-4o | Baseline | $16.60 \pm 4.65$ | $30.00 \pm 0.00$ | $26.17 \pm 2.12$ | $29.70 \pm 0.95$ | $63.30 \pm 0.67$ |
| | SSoT | $8.20 \pm 4.66$ (↓51%) | $10.80 \pm 4.44$ (↓64%) | $9.43 \pm 4.61$ (↓64%) | $9.90 \pm 3.28$ (↓67%) | $22.20 \pm 4.94$ (↓65%) |
| o4-mini-high | Baseline | $6.90 \pm 4.38$ | $28.00 \pm 0.94$ | $27.50 \pm 1.83$ | $29.90 \pm 0.32$ | $23.00 \pm 3.56$ |
| | SSoT | $3.70 \pm 2.36$ (↓46%) | $13.10 \pm 3.28$ (↓53%) | $11.97 \pm 4.69$ (↓56%) | $14.30 \pm 3.16$ (↓52%) | $14.80 \pm 3.85$ (↓36%) |
| QwQ-32B | Baseline | $5.00 \pm 4.99$ | $29.60 \pm 0.52$ | $43.77 \pm 5.04$ | $29.40 \pm 0.97$ | $60.90 \pm 1.60$ |
| | SSoT | $6.60 \pm 5.02$ (↑32%) | $4.50 \pm 4.25$ (↓85%) | $5.17 \pm 2.07$ (↓88%) | $3.50 \pm 2.27$ (↓88%) | $10.90 \pm 2.38$ (↓82%) |
| deepseek-r1 | Baseline | $25.20 \pm 4.87$ | $26.20 \pm 1.69$ | $31.57 \pm 1.50$ | $21.20 \pm 2.74$ | $45.80 \pm 4.02$ |
| | SSoT | $6.30 \pm 4.72$ (↓75%) | $4.30 \pm 2.91$ (↓84%) | $8.23 \pm 3.19$ (↓74%) | $5.90 \pm 3.75$ (↓72%) | $13.70 \pm 4.69$ (↓70%) |
| PRNG | | $4.90 \pm 3.73$ | $4.00 \pm 4.03$ | $6.63 \pm 2.93$ | $4.80 \pm 3.71$ | $12.80 \pm 2.15$ |

## C  ADDITIONAL EXPERIMENTAL DETAILS

### C.1  KL AND TV DIVERGENCE RESULTS ON PROBABILISTIC INSTRUCTION FOLLOWING

Here we will show the results for KL divergence and the total variation distance in Tables 4 and 5.

Also, we will show the results of the experiment for Figure 3 for KL divergence and the TV distance below in Figures 8 and 9.

### C.2  ROCK-PAPER-SCISSORS EXPERIMENT DETAILS

For the Rock-Paper-Scissors experiment in Section 5.1.3, we used 10 "black-belt" bots from "RPS dojo" kaggle notebook `https://www.kaggle.com/code/chankhavu/rps-dojo/notebook`.

The opponent bots are listed below:

- `black_belt/multi_armed_bandit_v15.py`
- `black_belt/multi_armed_bandit_v32.py`

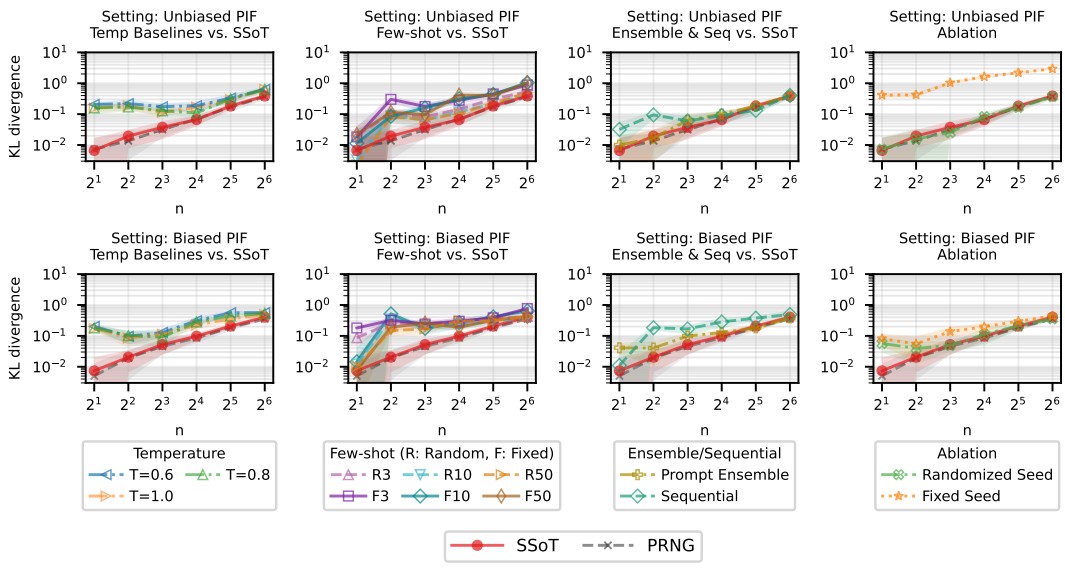

Figure 8: KL divergences for Unbiased and Biased PIF. Shaded areas represent the standard deviation.

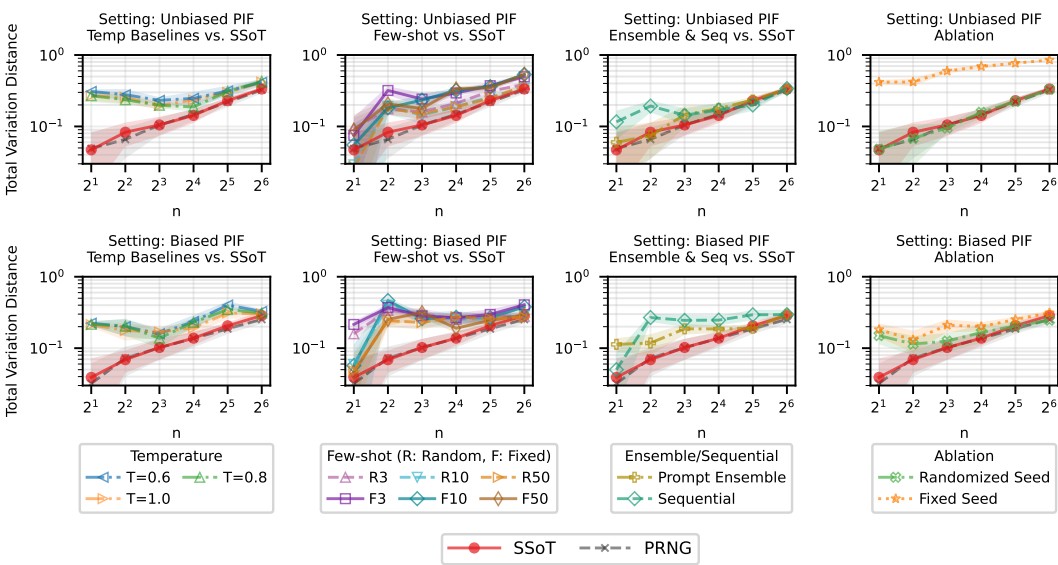

Figure 9: TV distance for Unbiased and Biased PIF. Shaded areas represent the standard deviation.

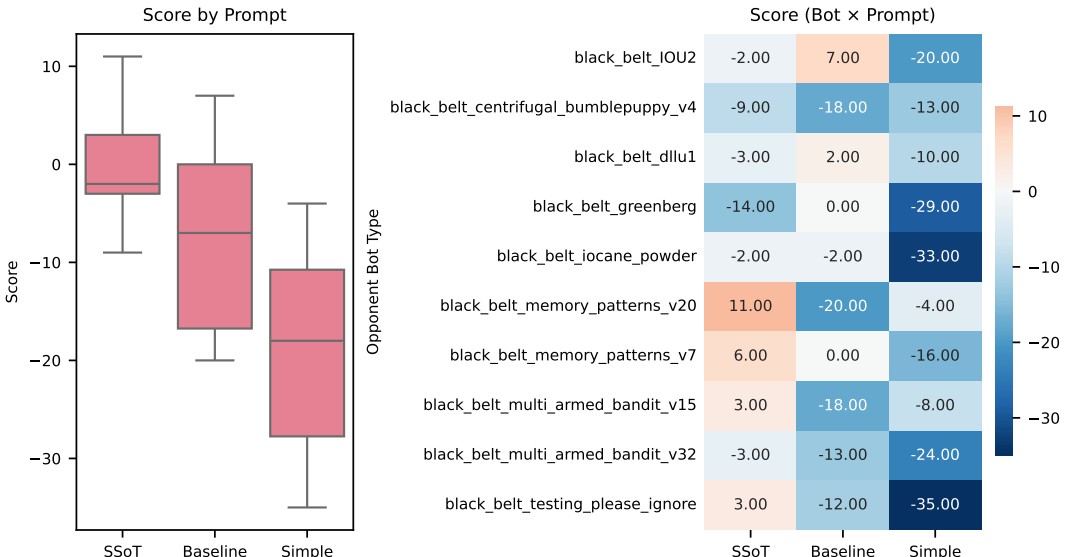

Figure 10: RPS results against black-belt bots.

- `black_belt/memory_patterns_v20.py`
- `black_belt/memory_patterns_v7.py`
- `black_belt/iocane_powder.py`
- `black_belt/greenberg.py`
- `black_belt/testing_please_ignore.py`
- `black_belt/IOU2.py`
- `black_belt/dllu1.py`
- `black_belt/centrifugal_bumblepuppy_4.py`

The scores and the full information, such as the opponent bot names, are shown in Figure 10.

Table 6: Kolmogorov-Smirnov statistics and Cramér-von Mises statistics, calculated for the baseline and SSoT prompts. We used deepseek-r1-0528 and calculated the statistics for 100 samples, repeated 10 times to obtain the mean and standard deviation.

| Distribution | Method | Kolmogorov-Smirnov | Cramér-von Mises |
|---|---|---|---|
| Uniform | Baseline | $0.220 \pm 0.023$ | $0.999 \pm 0.163$ |
| | **SSoT** | $0.187 \pm 0.034$ ($\downarrow 15\%$) | $0.839 \pm 0.405$ ($\downarrow 16\%$) |
| Normal | Baseline | $0.129 \pm 0.022$ | $0.358 \pm 0.146$ |
| | **SSoT** | $0.108 \pm 0.026$ ($\downarrow 16\%$) | $0.310 \pm 0.195$ ($\downarrow 14\%$) |
| Beta | Baseline | $0.177 \pm 0.039$ | $0.798 \pm 0.271$ |
| | **SSoT** | $0.113 \pm 0.040$ ($\downarrow 36\%$) | $0.293 \pm 0.208$ ($\downarrow 63\%$) |

Table 7: JS, KL, and TV divergences between the empirical distribution and the target distribution of the biased 9-choice task for different random string length targets $n$, together with the average generated string length. Divergences are reported as mean $\pm$ standard deviation over 10 chunks of 100 samples. The string length column reports the mean value only. We used deepseek-r1-0528 for this experiment.

| n | Generated String Length | JS Divergence | KL Divergence | TV Distance |
|---|---|---|---|---|
| 2 | 2.0 | $0.178 \pm 0.022$ | $0.705 \pm 0.086$ | $0.527 \pm 0.029$ |
| 4 | 4.2 | $0.228 \pm 0.017$ | $0.887 \pm 0.071$ | $0.581 \pm 0.021$ |
| 8 | 8.0 | $0.138 \pm 0.033$ | $0.581 \pm 0.128$ | $0.446 \pm 0.058$ |
| 16 | 16.2 | $0.037 \pm 0.011$ | $0.156 \pm 0.050$ | $0.222 \pm 0.040$ |
| 24 | 25.9 | $\mathbf{0.019 \pm 0.007}$ | $\mathbf{0.075 \pm 0.027}$ | $\mathbf{0.141 \pm 0.034}$ |
| 32 | 32.4 | $0.025 \pm 0.007$ | $0.092 \pm 0.026$ | $0.151 \pm 0.030$ |
| 40 | 40.0 | $0.032 \pm 0.010$ | $0.116 \pm 0.032$ | $0.191 \pm 0.028$ |
| 48 | 50.7 | $0.030 \pm 0.011$ | $0.108 \pm 0.039$ | $0.179 \pm 0.035$ |
| 64 | 69.6 | $0.044 \pm 0.011$ | $0.162 \pm 0.036$ | $0.244 \pm 0.029$ |
| 128 | 142.1 | $0.052 \pm 0.016$ | $0.187 \pm 0.054$ | $0.264 \pm 0.035$ |

# D   ADDITIONAL EXPERIMENTS

## D.1   SSoT PERFORMANCE ON SAMPLING FROM CONTINUOUS DISTRIBUTION

To demonstrate that SSoT is also effective for sampling from continuous distributions, we conducted experiments comparing SSoT and the Baseline on three continuous distributions: (1) the Uniform distribution on $[0, 1]$, (2) the Normal distribution $\mathcal{N}(0, 1)$, and (3) the Beta distribution $B(2, 5)$. We measured the Kolmogorov-Smirnov statistic and the Cramér-von Mises statistic to evaluate the goodness-of-fit of the empirical distribution constructed from 100 LLM samples. We repeated this process 10 times to calculate the mean and standard deviation, in total sampled 1000 LLM responses. We used deepseek-r1-0528 for these experiments.

The results are presented in Table 6. As observed, SSoT improves the goodness-of-fit metrics across all distributions, demonstrating its capability to handle continuous sampling tasks effectively.

## D.2   IMPACT OF GENERATED STRING LENGTH ON SSoT PERFORMANCE

To investigate the impact of the generated string length on PIF performance, we explicitly specified the target string length in the prompt and analyzed whether the model generated strings of the correct length and how this affected PIF performance. We used deepseek-r1-0528 for this analysis and selected the biased 9-choice task, which was the most challenging setting introduced in Section 5.1.

To control the random string length, we inserted the instruction "The random string must be {n} characters long." into the SSoT prompt immediately after the phrase "Use your judgment to ensure it looks arbitrary and unguessable." By varying the value of $n$, we were able to control the length of the generated strings.

Table 8: The PIF performance comparison of SSoT against the baseline for small LLMs, evaluated using the JS divergence (lower is better). All the JS divergences are presented in units of $10^{-3}$ (original values multiplied by 1000).

| Model | Method | 2-choice | biased 2-choice | 3-choice | biased 3-choice | biased 9-choice |
|---|---|---|---|---|---|---|
| Qwen3-8B | Baseline | $16.92 \pm 7.45$ | $86.55 \pm 16.61$ | $66.46 \pm 26.42$ | $87.30 \pm 14.99$ | $293.72 \pm 8.61$ |
| | SSoT | $7.36 \pm 9.02$ (↓56%) | $24.16 \pm 8.16$ (↓72%) | $7.72 \pm 5.03$ (↓88%) | $14.51 \pm 9.57$ (↓83%) | $60.23 \pm 19.14$ (↓79%) |
| Qwen3-4B | Baseline | $1.40 \pm 0.92$ | $117.28 \pm 0.00$ | $31.16 \pm 13.32$ | $115.53 \pm 5.51$ | $300.16 \pm 0.00$ |
| | SSoT | $7.48 \pm 4.11$ (↑436%) | $42.35 \pm 15.55$ (↓64%) | $55.82 \pm 8.24$ (↑79%) | $17.98 \pm 8.44$ (↓84%) | $104.15 \pm 19.51$ (↓65%) |
| Qwen3-1.7B | Baseline | $20.82 \pm 10.45$ | $94.47 \pm 18.26$ | $305.56 \pm 10.93$ | $90.04 \pm 5.07$ | $300.16 \pm 0.00$ |
| | SSoT | $106.70 \pm 14.00$ (↑413%) | $10.92 \pm 5.77$ (↓88%) | $259.41 \pm 20.95$ (↓15%) | $14.17 \pm 6.79$ (↓84%) | $229.36 \pm 13.51$ (↓24%) |
| DeepSeek-R1-Distill-Llama-8B | Baseline | $0.71 \pm 0.79$ | $75.06 \pm 23.55$ | $123.05 \pm 24.88$ | $49.76 \pm 17.14$ | $244.47 \pm 25.87$ |
| | SSoT | $0.38 \pm 0.62$ (↓46%) | $9.86 \pm 5.58$ (↓87%) | $4.71 \pm 4.83$ (↓96%) | $2.91 \pm 1.73$ (↓94%) | $75.17 \pm 17.80$ (↓69%) |
| Qwen3-thinking-4B | Baseline | $93.26 \pm 19.56$ | $117.28 \pm 0.00$ | $189.82 \pm 19.14$ | $117.28 \pm 0.00$ | $258.57 \pm 23.93$ |
| | SSoT | $10.85 \pm 8.01$ (↓88%) | $2.13 \pm 3.25$ (↓98%) | $5.01 \pm 4.51$ (↓97%) | $13.31 \pm 6.96$ (↓89%) | $31.07 \pm 13.69$ (↓88%) |
| Nemotron-Qwen-1.5B | Baseline | $128.18 \pm 22.90$ | $62.00 \pm 16.67$ | $258.94 \pm 25.43$ | $112.38 \pm 7.94$ | $221.08 \pm 17.13$ |
| | SSoT | $32.17 \pm 12.24$ (↓75%) | $51.49 \pm 16.17$ (↓17%) | $75.45 \pm 21.07$ (↓71%) | $11.35 \pm 6.48$ (↓90%) | $118.35 \pm 17.27$ (↓46%) |
| PRNG | | $1.85 \pm 2.58$ | $1.93 \pm 2.80$ | $3.36 \pm 2.48$ | $2.85 \pm 3.15$ | $13.72 \pm 4.21$ |

The results are shown in Table 7. First, for requested lengths of 40 or fewer, the model almost always generated strings of the correct length. Furthermore, compared to performance at $n = 2, 4, 8$, the PIF performance improves dramatically when $n$ exceeds 16, and maintains better performance up to $n = 128$ compared to $n = 8$. This supports the theoretical finding in Section 4 that performance improves with longer string lengths.

Additionally, the fact that performance peaks at $n = 24$ can be attributed to the increasing probability of arithmetic errors as the model attempts to perform complex operations (such as sum-mod or rolling hash) on longer strings.

## D.3   SSoT on Small Language Models

As elaborated in Section 5.3.1, LLMs are capable of autonomously discovering methods to sample from distributions. However, this implies that the model must possess the ability to find a correct algorithm and the reasoning capability to execute it. To verify this hypothesis, we assess the effectiveness of SSoT on smaller LLMs ranging from 1B to 8B parameters. It is well-known that complex prompting techniques, such as zero-shot CoT (Kojima et al., 2022), often yield limited gains on smaller models.

We used the same five settings as in Section 5.1: **(1, 2) 2-choice** and **Biased 2-choice**: ($\mathbf{a} = $ [heads, tails], $\mathbf{p} = [1/2, 1/2]$ and $[0.3, 0.7]$), **(3, 4) 3-choice** and **Biased 3-choice**: ($\mathbf{a} = $ [rock, paper, scissors], $\mathbf{p} = [1/3, 1/3, 1/3]$ and $[0.1, 0.2, 0.7]$), **(5) Biased 9-choice** ($\mathbf{a} = $ [one, two, ..., eight, nine], $\mathbf{p} = [0.08, 0.08, ..., 0.08, 0.36]$). We used the recommended temperature settings for each model.

We evaluated the Qwen3 family (8B, 4B, 1.7B) with "thinking mode" enabled, as well as reasoning-enhanced models: DeepSeek-R1-Distill-Llama-8B, Qwen3-thinking-4B, and Nemotron-Qwen-1.5B. The results are presented in Table 8. For Qwen3-8B, SSoT improves PIF capability, although not to the extent seen in larger LLMs or the PRNG. Conversely, Qwen3-4B and Qwen3-1.7B show improvement in some tasks, but their absolute performance remains suboptimal. This finding is consistent with the premise that SSoT requires a certain threshold of reasoning capability.

To further demonstrate that the limited improvement in small models is linked to reasoning capabilities, we examined the results for the reasoning-enhanced small LLMs (DeepSeek-R1-Distill-Llama-8B, Qwen3-thinking-4B, and Nemotron-Qwen-1.5B). As shown in Table 8, SSoT yields consistent performance improvements across tasks for these models, reinforcing the connection between reasoning ability and SSoT effectiveness.

## D.4   SSoT Works with Simple Prompt

In the main text, we utilized a relatively detailed SSoT prompt. This was designed to establish a strong baseline by explicitly instructing the model on the procedure, thereby allowing us to isolate the

Table 9: The PIF performance comparison of SSoT and SSoT (simple) against the baseline across various models, evaluated using the JS divergence (lower is better). All the JS divergences are presented in units of $10^{-3}$ (original values multiplied by 1000).

| Model | Method | 2-choice | biased 2-choice | 3-choice | biased 3-choice | biased 9-choice |
|---|---|---|---|---|---|---|
| deepseek-r1 | Baseline | $36.09 \pm 13.60$ | $69.58 \pm 13.42$ | $106.30 \pm 19.45$ | $49.53 \pm 11.24$ | $138.21 \pm 26.64$ |
| | **SSoT** | $3.03 \pm 3.43$ ($\downarrow$ 92%) | $1.51 \pm 1.55$ ($\downarrow$ 98%) | $4.98 \pm 3.84$ ($\downarrow$ 95%) | $4.30 \pm 4.92$ ($\downarrow$ 91%) | $18.06 \pm 11.35$ ($\downarrow$ 87%) |
| | **SSoT (Simple)** | $2.37 \pm 2.63$ ($\downarrow$ 93%) | $1.19 \pm 1.54$ ($\downarrow$ 98%) | $1.63 \pm 1.57$ ($\downarrow$ 98%) | $3.72 \pm 3.27$ ($\downarrow$ 92%) | $25.17 \pm 9.84$ ($\downarrow$ 82%) |
| PRNG | | $1.85 \pm 2.58$ | $1.93 \pm 2.80$ | $3.36 \pm 2.48$ | $2.85 \pm 3.15$ | $13.72 \pm 4.21$ |

pure effect of the random string element. However, SSoT functions effectively even with a simplified prompt, improving performance in both PIF and DAG tasks. To demonstrate this, we analyzed the performance of deepseek-r1-0528 using the following simplified prompt:

---

**Listing D.1: Simplified SSoT System Prompt**

```
Generate a complex random string between <random_string> and </
random_string>, and manipulate this string to guide any stochastic
decisions within <thinking> and </thinking> tags.

Then, provide your final answer, enclosed within `<answer>` and `</
answer>` tags.
```

---

For DAG, we included the objective in the system prompt. Apart from the first line describing the objective, the subsequent text remains identical to the PIF version.

---

**Listing D.2: Simplified SSoT System Prompt (DAG)**

```
You must produce exactly one unique and diverse answer. To do this,
 first generate a complex random string between <random_string> and
 </random_string>, and manipulate this string to guide any
stochastic decisions within <thinking> and </thinking> tags.

Then, provide your final answer, enclosed within <answer> and </
answer> tags.
```

---

The essential components of the SSoT prompt are: (1) instructing the LLM to generate a complex string, (2) asking it to manipulate the string, and (3) generating a final answer. Crucially, similar to zero-shot CoT, explicitly instructing the model to write out the manipulation process in the output is vital; otherwise, reasoning models may hallucinate the derivation without actually using the string. While we use tags to structure the output and facilitate parsing, they are not strictly necessary as long as the content is generated. Using tags, however, simplifies the extraction of the final answer for user display.

We conducted experiments using these simplified prompts for the PIF task (Section 5.1.2) and the NoveltyBench task (Section 5.2). The results are shown in Table 9 and Table 10.

As we can see, the performance of a simple variant of the SSoT prompt is on par with the sophisticated SSoT prompt, demonstrating that we can easily employ SSoT, only following the three-step instruction setup and the inclusion of the objective in its system prompt.

### D.5 SSoT FAILURE ANALYSIS

As shown in Table 1, QwQ-32B's performance with SSoT on the 2-choice PIF task is worse than the baseline. To investigate the cause of this discrepancy, we analyzed 1000 random strings and the corresponding responses generated by QwQ-32B for the 2-choice PIF task.

Table 10: SSoT Simple prompt result on NoveltyBench (Curated). Cells show Distinct (Utility); higher is better.

| Method | Creativity | Naming | Facts | Product Recs | Random | Opinions | Overall |
|--------|-----------|--------|-------|--------------|--------|----------|---------|
| Baseline | 4.60 (5.61) | 6.00 (6.13) | 4.14 (5.35) | 4.14 (6.02) | 6.07 (5.10) | 4.35 (4.08) | 4.70 (5.17) |
| **SSoT** | 5.90 **(6.44)** | **7.57 (6.62)** | **6.04 (6.71)** | 5.86 (6.63) | **6.87 (5.49)** | 5.87 **(4.35)** | 6.19 **(5.92)** |
| **SSoT (Simple)** | **6.20** (6.38) | 6.86 (5.66) | 5.96 (6.49) | **5.86 (7.32)** | 6.60 (5.30) | **6.39** (4.02) | **6.26** (5.72) |

We identified a notable bias in the generated random strings: 947 out of 1000 strings started with the digit "7". Furthermore, utilizing gemini-2.5-flash to classify the strategies used in the 1000 responses, we found that 50 responses employed a strategy that simply selected "heads" or "tails" based on the ASCII code of the string's first character. Among these 50 cases, 45 had a random string starting with "7", resulting in 43 selections of "tails" and only 2 of "heads". Removing those failure cases leads to a near-ideal distribution.

This analysis reveals that bias can be introduced when the generated string itself contains a pattern (e.g., a fixed starting character) and the model simultaneously selects a strategy that fails to extract sufficient entropy from the entire string. However, even in such cases, if these failure modes can be mitigated via prompt steering, it is possible to sample from an unbiased distribution.

### D.6 SSoT vs External Randomness Generation in DAG

In this section, we demonstrate that, in the NoveltyBench DAG setting, SSoT outperforms approaches that obtain randomness externally, such as injecting PRNG-generated strings into the prompt or using tool calls to external random modules (e.g., Python's random module).

To evaluate external randomness generation in DAG, we consider the following two baselines:

1. **Seed Injection**: This method involves sampling a fresh random string from a PRNG for each query, inserting it into the prompt, and instructing the model to use this string when generating its response. We utilized the system prompt in Listing D.4 and prepended Listing D.5 to the user prompt. We used a 24-character string drawn from printable ASCII characters [4].

2. **Tool call**: We prepared tools consisting of `random_int` and `random_string`, as shown in Listing D.3, and instructed the LLM to perform tool calls. We evaluated this baseline on the NoveltyBench curated dataset. The system prompt was set as shown in Listing D.6, explicitly instructing the model to use external tools whenever randomness was required.

We used deepseek-r1-0528 as the underlying model, following the setup in Section 5.2.

The results comparing these baselines with SSoT are presented in Table 11. As the table demonstrates, SSoT outperforms both external randomness baselines in terms of Distinct and Utility scores across all categories, except for "Product Recs". Notably, in the "Creativity" task, while the external methods struggled to improve Utility as diversity increased, SSoT successfully achieved high scores in both metrics, demonstrating its ability to enhance diversity without compromising response quality.

The performance gap can be attributed to several factors:

- **Limitations of Seed Injection**: Providing a single external string limits the model's ability to employ strategies that utilize randomness multiple times or for local decisions, as observed in our CoT analysis in Section 5.3.1. Furthermore, as shown in the rightmost panel of Figure 3 (Section 5.1.2), providing an external string generally yields lower PIF performance compared to the internal generation of SSoT.

- **Instability of Tool Calls**: While Tool Call theoretically allows for multiple randomness requests, its performance proved unstable. This is primarily due to the difficulty of using custom tools out of the box at inference time, which led to frequent tool-calling failures (e.g., incorrect function names or arguments).

---

[4]We chose a 24-character string, since it yielded the best performance in Table 7.

Table 11: SSoT, Seed Injection, and Tool Call baseline results on NoveltyBench (Curated). Cells show Distinct (Utility); higher is better.

| Method | Creativity | Naming | Facts | Product Recs | Random | Opinions | Overall |
|---|---|---|---|---|---|---|---|
| Baseline | 4.60 (5.61) | 6.00 (6.13) | 4.14 (5.35) | 4.14 (6.02) | 6.07 (5.10) | 4.35 (4.08) | 4.70 (5.17) |
| Seed Injection | 5.60 (5.90) | 7.43 (6.46) | 5.75 (6.47) | 6.43 (**7.99**) | 6.67 (5.41) | 5.65 (4.13) | 6.00 (5.76) |
| Tool Call | 5.60 (5.10) | 7.00 (6.41) | 5.39 (6.34) | **6.71** (7.72) | 5.73 (4.75) | 5.52 (3.60) | 5.72 (5.33) |
| **SSoT** | **5.90** (**6.44**) | **7.57** (**6.62**) | **6.04** (**6.71**) | 5.86 (6.63) | **6.87** (5.49) | **5.87** (**4.35**) | **6.19** (**5.92**) |

These results suggest that, in our setting, even frontier LLMs like deepseek-r1-0528 may not reliably make effective use of arbitrary tools out of the box. Moreover, employing tool calls necessitates executing the tools and making multiple API calls, which complicates the use of batch APIs, since each tool call requires a separate interaction and its result must be fed into a subsequent prompt. In contrast, SSoT is highly simple, requiring only a modification to the prompt, and is fully compatible with batch APIs. Thus, in this setting, SSoT offers a more favorable combination of performance and usability than the tool-call-based approach.

Listing D.3: Random module tools consisting of `random_int` and `random_string` functions.

```
RANDOM_MODULE_TOOLS = [
    {
        "type": "function",
        "function": {
            "name": "random_int",
            "description": "Return a random integer between min and
     max (inclusive).",
            "parameters": {
                "type": "object",
                "properties": {
                    "min": {
                        "type": "integer",
                        "description": "Minimum integer value (
     inclusive).",
                    },
                    "max": {
                        "type": "integer",
                        "description": "Maximum integer value (
     inclusive).",
                    },
                },
                "required": ["min", "max"],
            },
        },
    },
    {
        "type": "function",
        "function": {
            "name": "random_string",
            "description": "Return a random printable ASCII string
     of the given length.",
            "parameters": {
                "type": "object",
                "properties": {
                    "length": {
                        "type": "integer",
                        "minimum": 1,
                        "maximum": 256,
                        "description": "Desired length of the
     random string. Must be between 1 and 256.",
                    }
                },
                "required": ["length"],
            },
        },
    }
]
```

Listing D.4: External Random Seed Baseline System Prompt

```
You are a helpful AI Assistant designed to provide well-reasoned
and detailed responses. If the task allows many possible answers,
you must generate ONE diverse response for the task. You will be
given a random string which is different for every request from
users. Use this seed (the exact contents inside the `<random_string
>` tags) to guide any random sampling or stochastic decisions.

Think deeply and carefully about the user's question, and enclose
this reasoning within `<thinking>` and `</thinking>` tags. Then
provide your final answer, enclosed within `<answer>` and `</answer
>` tags.

Strictly follow this tag structure, and respond in the following
format:
<thinking>
...
</thinking>
<answer>
...
</answer>
```

Listing D.5: External Random Seed Baseline User Prompt Prefix

```
Using this random seed:
<random_string>
{random_string}
</random_string>
```

Listing D.6: Random Module Tool Call Baseline System Prompt

```
You are a helpful AI assistant that must produce exactly one unique
 and diverse answer for each user prompt.

Use given tools via tool calls whenever you need randomness. Do NOT
 invent or simulate randomness internally; always rely on the tools
 for random values or random strings. Use the randomness you obtain
 to drive diverse outputs in your answer.

For each prompt where multiple valid answers are possible:
- Think step by step inside `<thinking>` and `</thinking>` tags (
optionally using given tools).
- Provide your final answer between `<answer>` and `</answer>` tags.

Respond strictly in the following structure:
<thinking>
...
</thinking>
<answer>
...
</answer>
```

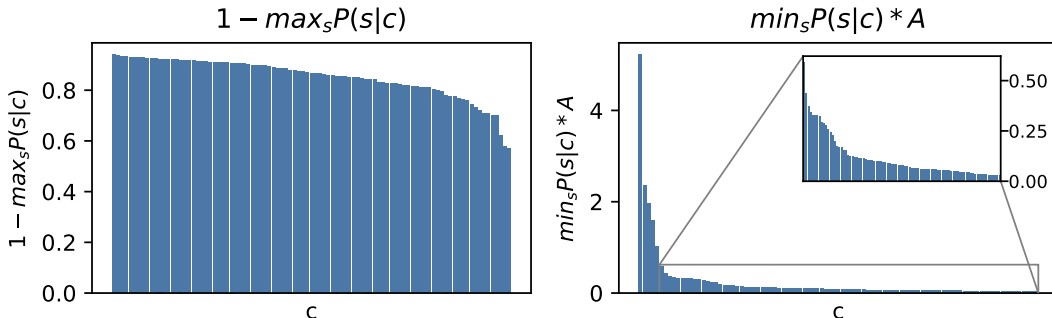

Figure 11: The distribution of $1 - \max_s P(s|c)$ and $\min_s P(s|c) * A$. The X-axis labels the prefix $c$, and is sorted in descending order of its value.

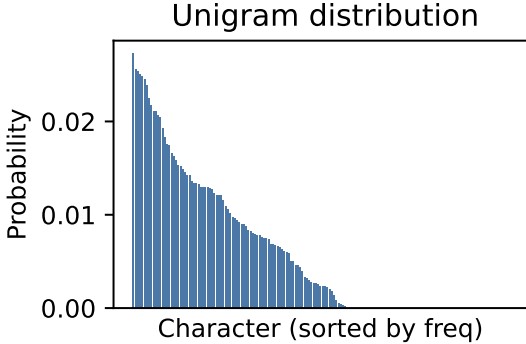

Figure 12: The unigram distribution of a generated random string. The frequency decays slowly (approximately linearly), rather than a steep power-law decay.

## E  GENERATED STRING ANALYSIS

In this section, we analyze the statistical properties of the generated strings in PIF to examine the extent to which the theoretical assumptions hold in practice and how they affect performance. The following analysis was conducted on a total of 5000 strings generated by deepseek-r1-0528 in the experiments described in Section 5.1.1. The average length of each string was 44.69, and the alphabet size was $A = 167$.

First, to verify the validity of the assumption in Theorem G.8, we investigated the extent to which the $\delta$-SV condition (Definition G.5) holds for the probability distribution $P(s|c)$ of a character $s$ following a context string $c$. Since increasing the length of $c$ excessively would result in insufficient samples, we set the length of $c$ to 1 and limited the analysis to bigrams.

The results are shown in Figure 11. The $\delta$-SV assumption requires the condition to hold for any context $c$ and any generated character $s$. First, the left panel indicates that the upper bound of the $\delta$-SV assumption is well satisfied. Next, observing the right panel, as we vary $c$, the value $\min_s P(s|c) * A$ exhibits a long tail. This tail takes values around $\sim 0.1$, suggesting that while not perfect, the lower bound condition is not significantly violated.

Next, to examine the bias in the generated characters, we present the results of the unigram analysis. As shown in Figure 12, the generated characters are not perfectly uniform; however, the frequency decays linearly rather than following a power law, indicating that generation is not concentrated on a small subset of characters. This suggests that the situation assumed in the theorem, where there is no extreme bias in the types of generated characters, holds approximately.

Finally, we examine the assumption of Theorem G.9. Specifically, for the sum-mod strategy, the distribution of the ASCII codes is more important than the distribution of the characters themselves. To investigate the independence of the generated character ASCII codes, we calculated the independence between characters separated by a lag $n$ using the total variation distance

Table 12: TV distance $d_{\text{TV}}(P(X_{i+n}, X_i), P(X_{i+n})P(X_i))$ for different lags $n$ and modulo $m$ of $X_i$.

| Modulo $m$ | $n=1$ | $n=2$ | $n=3$ |
|---|---|---|---|
| 2 | 0.0164 | 0.0042 | 0.0249 |
| 3 | 0.0275 | 0.0204 | 0.0051 |
| 9 | 0.0645 | 0.0513 | 0.0409 |

$d_{TV}(P(X_{i+n}, X_i), P(X_{i+n})P(X_i))$ where we denote the ASCII code of each character modulo $m = 2, 3, 9$ as $X_i$. The results are shown in Table 12.

Since the total variation distance takes values in $[0, 1]$, these values are sufficiently small. This result indicates that even if there are correlations between the characters themselves due to the autoregressive nature of generation, the independence of their ASCII codes is approximately guaranteed.

## F  OVERVIEW OF NOVELTYBENCH

NOVELTYBENCH Zhang et al. (2025) evaluates the ability of LLMs to generate diverse and high-quality responses. Importantly, every question in the benchmark is selected to admit a wide space of distinct, high-quality outputs (ranging from multiple valid answers in factual queries to diverse style and narrative variations in creative tasks), while the quality of each response remains quantifiable.

The benchmark employs two metrics calculated over $k$ samples. First, **Distinct** quantifies diversity by using a fine-tuned DeBERTa classifier to partition generations into functional equivalence classes $\{c_i\}$; the metric is defined as the count of unique classes:

$$\text{Distinct}_k := |\{c_i \mid i \in [k]\}|. \tag{3}$$

Secondly, **Cumulative Utility** unifies diversity and quality by summing the reward scores $u_i$ of only the *novel* generations (i.e., those belonging to a new equivalence class), discounted by a user patience factor $p = 0.8$. The utility is calculated as:

$$\text{Utility}_k := \frac{1-p}{1-p^k} \sum_{i=1}^{k} p^{i-1} \cdot \mathbb{I}[c_i \neq c_j, \forall j < i] \cdot u_i. \tag{4}$$

We set $k = 8$ in Tables 2 and 3, following the value of the original paper.

The benchmark consists of two distinct subsets. **NB-Curated** contains 100 prompts manually designed to cover specific categories of diversity (detailed in Table 13). **NB-WildChat** comprises 1000 prompts sampled from real-world user interactions in the WildChat-1M dataset. To ensure relevance, these prompts were filtered for safety using Llama Guard 3, deduplicated by user IP to minimize topic bias, and selected via a GPT-4o classifier to strictly include queries that allow for multiple valid, distinct responses.

| Category (*Abbreviation*) | Description | Example Question |
|---|---|---|
| **Creativity** (*Creativity*) | Open-ended imaginative writing tasks such as storytelling and poetry. | "Tell me a story in five sentences about a girl and her dog." |
| **Character & Entity Naming** (*Naming*) | Generating names for specific entities, fictional characters, or pets. | "Suggest a name for a dappled-gray filly living in the mountains." |
| **Factual Knowledge** (*Facts*) | Factual questions where constraints allow for multiple valid correct answers. | "List a capital city in Africa." |
| **Product & Purchase Recommendations** (*Product Recs*) | Shopping advice requests where distinct suggestions provide value. | "What's the best car to get in 2023? Just give me one suggestion." |
| **Random Generation & Selection** (*Random*) | Explicit requests for stochastic processes like dice rolls or random picks. | "Roll a make-believe 20-sided die." |
| **Subjective Rankings & Opinions** (*Opinions*) | Subjective questions with no objective consensus, testing perspective diversity. | "What is the coolest Pokémon from the second generation? Just give me one." |

Table 13: Categories in the NB-Curated dataset. Abbreviations in parentheses correspond to those used in Table 2.

# G   FULL THEORETICAL ANALYSIS OF SSOT PERFORMANCE ON PIF

This section provides a full theoretical analysis of SSoT performance on PIF. For an informal and short version, see Section 4.

We investigate the properties that the generated strings in SSoT must possess to achieve uniformity of the obtained empirical distribution, and how the deviation from a uniform distribution can be bounded for a simple sum-mod operation and standard hash functions to perform a random selection.

## G.1   SETTINGS

In this section, we introduce the notation and definitions used throughout our analysis. Unless otherwise specified, all logarithms are base 2.

We consider the process of selecting a sequence of $n$ characters, $x_1 x_2 \ldots x_n$, from an alphabet set $\Sigma$ of size $A$, denoted as $\Sigma^n$. We also introduce an encoding function $Enc : \Sigma \to \mathbb{Z}_A$, which is a bijection. Since we fix a single encoding function (e.g., ASCII coding) for our analysis, we can simplify the problem by considering a sequence of length $n$ with elements from $\mathbb{Z}_A$, denoted as $\mathbb{Z}_A^n$, instead of $\Sigma^n$.

Next, we introduce our probability notation. We consider only probabilities defined on finite sets. For a random variable $X$ over a finite set $S$ of cardinality $M$, we denote its possible values by $x \in S$, each associated with a non-negative probability $P(x) \geq 0$, satisfying $\sum_{x \in S} P(x) = 1$. When $x$ is sampled from the distribution of a random variable $X$, we write $x \sim X$.

Now we formulate the PIF problem as follows: The trial of sampling a random string from $\mathbb{Z}_A^n$ is repeated independently $K$ times. Here, $n$ is assumed to be the same for all trials; however, this framework can accommodate variable string lengths by setting $n$ to the minimum length across all $K$ trials and truncating any longer strings. Let $s_k$ be the string sampled in the $k$-th trial ($k = 1, \ldots, K$). The empirical distribution of the hashed values in $\mathbb{Z}_M$, obtained using hash functions $h_k : \mathbb{Z}_A^n \to \mathbb{Z}_M$, is defined as:

$$\hat{P}_{\{h_k(s_k)\}_{k=1}^K}(m) := \frac{1}{K} \sum_{k=1}^K I(h_k(s_k) = m), \quad (m \in \mathbb{Z}_M), \tag{5}$$

Note that the empirical distribution itself is a random variable, since it depends on the sampled strings $s_k$.

Our goal is to make this empirical distribution as close as possible to the uniform distribution on $\mathbb{Z}_M$, which first requires defining a distance metric between distributions. We use the following total variation distance as our metric:

**Definition G.1** (Total Variation Distance). Given two probability distributions $P$ and $Q$ and random variables $X$ and $Y$, both over $\mathbb{Z}_M$, we define *the total variation distance* $d_{TV}$ as:

$$d_{\text{TV}}(P, Q) = \frac{1}{2} \sum_{m \in \mathbb{Z}_M} |P(m) - Q(m)|. \tag{6}$$

We also denote $d_{\text{TV}}(P, Q)$ as $d_{\text{TV}}(X, Y)$.

We also define the following entropy measures:

**Definition G.2** (Rényi entropy). Given a random variable $X$ and its probability distribution $P$ over a set $\mathcal{T}$, we define *Rényi entropy* $H_2$ as:

$$H_2(X) = -\log \sum_{t \in \mathcal{T}} p(t)^2 \tag{7}$$

**Definition G.3** (min-entropy). Given a random variable $X$ and its probability distribution $P$ over a set $\mathcal{T}$, we define *min-entropy* $H_\infty$ as:

$$H_\infty(X) = -\log \max_{t \in \mathcal{T}} p(t) \tag{8}$$

To state our main theorems, we first need to introduce several key concepts.

**Definition G.4** (2-universal hash functions, Carter & Wegman (1979)). Let $\mathcal{S}$ and $\mathcal{T}$ be finite sets. A family of hash functions $\mathcal{H} = \{ h : \mathcal{S} \to \mathcal{T} \}$ is called *2-universal hash functions* if for any two distinct inputs $x_1, x_2 \in \mathcal{S}$,

$$\Pr_{h \sim \mathcal{H}}[h(x_1) = h(x_2)] \leq \frac{1}{|\mathcal{T}|}, \tag{9}$$

where the probability is taken over a hash function $h$ chose uniformly at random from $\mathcal{H}$.

The above hash function is used when we leverage the Leftover Hash Lemma in Theorem G.8. The following $\delta$-SV source is also used:

**Definition G.5** (Santha-Vazirani sources, Santha & Vazirani (1986)). A sequence of random variables $(X_1, \ldots, X_n)$ taking values in $\mathbb{Z}_A$ is a $\delta$-**Santha-Vazirani source** if for any realization $x_i \sim X_i$, the conditional probabilities satisfy:

$$\delta \leq P(x_i | \{x_j\}_{0 \leq j < i}) \leq 1 - (A-1)\delta \quad \left( 0 \leq \delta \leq \frac{1}{A} \right), \tag{10}$$

Hereafter, we refer to this as a $\delta$-SV source. The Santha-Vazirani source is typically defined for binary alphabets ($A = 2$), but here we use a natural extension to an alphabet of size $A$.

We note that this precondition G.5 allows correlations among character generation distribution at different positions; Specifically, it allows autoregressive correlation, which is unavoidable when the string is generated by an LLM.

We also introduce several key theorems required for our proofs. First, the following theorem bounds the total variation distance between an empirical distribution and its underlying true distribution.

**Theorem G.6** (An upper bound on TV distance between the empirical and the true distributions, Weissman et al. (2003)). *For an empirical distribution defined by Equation* (5) *calculated from a sequence of $K$ independent random variables $\{X_i\}_{i=1}^{K}$ drawn from a distribution $P_X$ on $\mathbb{Z}_M$, let*

$$\phi(x) := \frac{1}{1 - 2x} \ln \frac{1-x}{x}, \quad \pi_P := \max_{S \subseteq \mathbb{Z}_M} \min(P(S), 1 - P(S)). \tag{11}$$

*Then,*

$$\Pr(d_{TV}(P_X, \hat{P}_{\{X_i\}_{i=1}^{K}}) \geq \epsilon) \leq (2^M - 2)e^{-K\phi(\pi_{P_X})\epsilon^2} \tag{12}$$

*Equivalently, for any real value $\delta \in (0, 1)$, the following holds with a probability of at least $1 - \delta$:*

$$d_{TV}(P_X, \hat{P}_{\{X_i\}_{i=1}^{K}}) \leq \sqrt{\frac{\ln((2^M - 2)/\delta)}{K\phi(\pi_{P_X})}}. \tag{13}$$

*Proof.* The first part of this theorem, Equation (12), is proven in (Weissman et al., 2003). We derive Equation (13) as follows. By rearranging Equation (12), we have:

$$\Pr(d_{TV}(P_X, \hat{P}_{\{X_i\}_{i=1}^{K}}) < \epsilon) \geq 1 - (2^M - 2)e^{-K\phi(\pi_{P_X})\epsilon^2}. \tag{14}$$

Setting $\delta = (2^M - 2)e^{-K\phi(\pi_{P_X})\epsilon^2}$ and solving for $\epsilon$ yields:

$$\Leftrightarrow \epsilon = \sqrt{\frac{\ln((2^M - 2)/\delta)}{K\phi(\pi_{P_X})}} \tag{15}$$

Substituting this into Equation (14) proves Equation (13). □

Additionally, the following well-known Leftover Hash Lemma will be used (See, e.g., Vadhan (2012) for its proof).

**Lemma G.7** (Leftover Hash Lemma, Impagliazzo et al. (1989)). *Let $\mathcal{H} = \{h : \mathcal{S} \to \mathcal{T}\}$ be a 2-universal hash family, and let $H$ be a random variable for a hash function chosen uniformly at random from $\mathcal{H}$. For any random variable $X$ over $\mathcal{S}$, the total variation distance between the joint distribution of $(H, h_H(X))$ and the ideal distribution $(H, U_{\mathcal{T}})$, where $U_{\mathcal{T}}$ is the uniform distribution on $\mathcal{T}$, is bounded as follows:*

$$d_{TV}((H, h_H(X)), (H, U_{\mathcal{T}})) \leq \frac{1}{2}\sqrt{|\mathcal{T}|2^{-H_2(X)}} \leq \frac{1}{2}\sqrt{|\mathcal{T}|2^{-H_\infty(X)}}. \tag{16}$$

### G.2 MAIN THEOREMS

Our goal is to prove the following two theorems.

**Theorem G.8** (TV distance bound with random hash function). *Let $X^n = \{X_1, \ldots, X_n\}$ be a sequence of random variables over $\mathbb{Z}_A$ that constitutes a $\delta$-SV source. Using a 2-universal hash family $\mathcal{H} = \{h : \mathbb{Z}_A^n \to \mathbb{Z}_M\}$, the total variation distance is bounded by:*

$$d_{TV}((H, h_H(X^n)), (H, U_{\mathbb{Z}_M})) \leq \frac{\sqrt{M}}{2}2^{-\frac{n}{2}\log\frac{1}{(1-(A-1)\delta)^2+(A-1)\delta^2}} = 2^{\log\frac{\sqrt{M}}{2} - n\frac{A-1}{\log_e 2}\delta + \mathcal{O}(n\delta^2)}, \tag{17}$$

*where $U_{\mathbb{Z}_M}$ is the uniform distribution on $\mathbb{Z}_M$. This equation shows that for a given $\delta$-SV source, the total variation distance can be made arbitrarily small by choosing a sufficiently large $n$. Specifically, for small $\delta$, the distance becomes negligible if the string length $n$ satisfies:*

$$n \gg \frac{\log_e \frac{\sqrt{M}}{2}}{(A-1)\delta}. \tag{18}$$

*Furthermore, for the total variation distance between the empirical distribution and the uniform distribution, the following holds with a probability of at least $1 - \delta' - \delta''$ for any $\delta', \delta'' \in (0, 1)$:*

$$d_{TV}(\hat{P}_{\{h((x^n)_k)\}_{k=1}^K}, U_{\mathbb{Z}_M}) \leq \frac{\sqrt{M}}{2\delta''}2^{-\frac{n}{2}\log\frac{1}{(1-(A-1)\delta)^2+(A-1)\delta^2}} + \sqrt{\frac{\ln((2^M-2)/\delta')}{K\phi(\pi_{P_X})}}. \tag{19}$$

This theorem guarantees the uniformness of the empirical distribution given that the string length is long enough. As for the choice of the hash function, the diversity among different responses is **not** required. 2-universal hash functions can be realized as a random linear map on a finite field $\mathbb{Z}_p$ with $p \geq M$ and then taking modulo $M$ (Carter & Wegman, 1979), which is within the capabilities of LLMs.

**Theorem G.9** (TV distance bound with sum-and-mod strategy). *Let $\{X_1, \ldots, X_n\}$ be a sequence of random variables over $\mathbb{Z}_A$, and that these are independent with the probability distribution $\eta_i$ $(i = 1, \ldots, n)$. We embed each element $x_i$ into $\mathbb{Z}_M$ and compute their sum modulo $M$:*

$$s_n = \sum_{1 \leq i \leq n} x_i \mod M. \tag{20}$$

Let $S_n$ be the random variable corresponding to $s_n$, with probability distribution $P_{S_n}$. Let the Fourier transform of $\eta_i(x)$ be:

$$\hat{\eta}_i(k) = \sum_{x \in \mathbb{Z}_M} e^{2\pi i k x / M} \eta_i(x), \quad k \in \mathbb{Z}_M, \tag{21}$$

where, for $\eta_i$, we embed the probability distribution over $\mathbb{Z}_A$ to the one over $\mathbb{Z}_M$ with modulo (if $M < A$) or $0$ padding (if $M > A$) operations. Then, the total variation distance is bounded as follows:

$$d_{TV}(P_{S_n}, U_{\mathbb{Z}_M}) \leq \frac{1}{2} \sqrt{ \sum_{\substack{d | M,\, d > 1}} \sum_{\substack{t=1 \\ \gcd(t,d)=1}}^{d-1} \prod_{i=1}^{n} \left| \widehat{\pi_{d*}(\eta_i)}(t) \right|^2 }, \tag{22}$$

where $\pi_d : \mathbb{Z}_M \to \mathbb{Z}_d$ is the projection mapping an element to its value modulo $d$, and $\pi_{d*}(\eta_i)$ is the pushforward of $\eta_i$ by $\pi_d$, defined as:

$$\pi_{d*}(\eta_i)(r) \coloneqq \sum_{x \in \mathbb{Z}_M,\, x \equiv r\, (mod\, d)} \eta_i(x). \tag{23}$$

$\widehat{\pi_{d*}(\eta_i)}$ is the Fourier transform over the group $\mathbb{Z}_d$. Furthermore, using total variation distance, the following bounds hold:

$$d_{TV}(P_{S_n}, U_{\mathbb{Z}_M}) \leq \frac{1}{2} \sqrt{ \sum_{\substack{d | M,\, d > 1}} \varphi(d) \prod_{i=1}^{n} (2 d_{TV}(\pi_{d*}(\eta_i), U_{\mathbb{Z}_d}))^2 }, \tag{24}$$

where $\varphi(d)$ is Euler's totient function. Moreover, for an empirical distribution derived from $K$ independent samples $\{s_n^k\}_{k=1}^{K}$, the following holds with a probability of at least $1 - \delta'$ for any $\delta' \in (0,1)$:

$$d_{TV}(\hat{P}_{\{s_n^k\}_{k=1}^K}, U_{\mathbb{Z}_M}) \leq \frac{1}{2} \sqrt{ \sum_{\substack{d | M,\, d > 1}} \varphi(d) \prod_{i=1}^{n} (2 d_{TV}(\pi_{d*}(\eta_i), U_{\mathbb{Z}_d}))^2 } + \sqrt{ \frac{\ln((2^M - 2)/\delta')}{K \phi(\pi_{P_X})} }, \tag{25}$$

This theorem focuses on a simple sum-mod operation, which is in fact employed by LLMs (see CoT analysis in Section 5.3.1).

## G.3 THEORETICAL SIGNIFICANCE AND CONTRIBUTIONS

In this section, we clarify the specific theoretical contributions of our work in the context of hash decoding and LLM generation, addressing the connection between our theorems and the practical behavior of Large Language Models.

**Generalization of Randomness Extraction for Text Generation (Theorem G.8).** While the Leftover Hash Lemma (Lemma G.7) is a foundational result in pseudorandomness, our contribution lies in adapting it to the specific constraints of Large Language Models.

- **Extension to Non-Binary Alphabets**: Standard literature often analyzes Santha-Vazirani sources in the context of binary strings. We extended the definition of $\delta$-Santha-Vazirani sources to arbitrary alphabet sizes $A$ (Definition G.5) to accurately model token generation in LLMs.

- **Explicit Entropy Bounds**: We explicitly derived the lower bound of the Rényi entropy $H_2(X)$ for these generalized sources (Lemma G.10).

- **Closed-Form Guarantee**: By combining the extraction bound with Weissman's inequality, we provided a closed-form upper bound on the total variation distance for the empirical distribution. This offers a rigorous guarantee that a hash function can extract near-uniform randomness from an autoregressive LLM, provided the generated string length $n$ is sufficiently large.

**Spectral Analysis of LLM Strategies (Theorem G.9).** Our contribution here is bridging the gap between abstract theory and the empirical behavior of LLMs observed in our Chain-of-Thought analysis.

- **Modeling "Sum-Mod" as a Random Walk:** As shown in Section 5.3.1, LLMs spontaneously adopt a strategy of summing ASCII codes modulo $M$. We formalized this operation as a random walk on the finite cyclic group $\mathbb{Z}_M$.

- **Spectral Bounds:** Adapting the Fourier analysis techniques of Diaconis & Shahshahani (1981), we derived a spectral bound using the Fourier coefficients of the character distribution.

- **Practical Convergence:** This theorem proves that even with simple arithmetic operations—which are within the reliable reasoning capabilities of current LLMs—the output distribution converges to uniformity exponentially fast with respect to the string length.

### G.4    PROOF OF THEOREM G.8

*Proof.* First, we prove the following lemma.

**Lemma G.10** (Rényi entropy of $\delta$-SV-sources)**.** *For a random variable $X = (X_1, \ldots, X_n)$ corresponding to a sequence of $n$ $\delta$-SV sources, its Rényi entropy $H_2(X)$ is bounded by:*

$$-n \log \left[ (1 - (A-1)\delta)^2 + (A-1)\delta^2 \right] \leq H_2(X) \leq n \log A \tag{26}$$

*Proof.* The collision probability of $X$ is given by:

$$\sum_{x_1, \ldots, x_n} P(x_1, \ldots, x_n)^2 = \sum_{x_1, \ldots, x_{n-1}} P(x_1, \ldots, x_{n-1})^2 \sum_{x_n} P(x_n | \{x_j\}_{j<n}))^2$$

$$= \sum_{x_1, \ldots, x_{n-1}} P(x_1, \ldots, x_{n-1})^2 \sum_{x_n} (P(x_n | \{x_j\}_{j<n}))^2$$

$$=: \sum_{x_1, \ldots, x_{n-1}} P(x_1, \ldots, x_{n-1})^2 D(\{x_j\}_{j<n}). \tag{27}$$

Next, we bound the term $D(\{x_j\}_{j<n})$. By the Cauchy-Schwarz inequality,

$$D(\{x_j\}_{j<n}) \geq \frac{1}{A} \left( \sum_{x_n} P(x_n | \{x_j\}_{j<n}) \right)^2 = \frac{1}{A} \tag{28}$$

We define $E(x_n, \{x_j\}_{j<n}) := P(x_n | \{x_j\}_{j<n}) - \delta$, and noting that $E(x_n, \{x_j\}_{j<n}) \geq 0$

$$D(\{x_j\}_{j<n}) = \sum_{x_n} (\delta + E(x_n, \{x_j\}_{j<n}))^2$$

$$= A\delta^2 + 2\delta \sum_{x_n} E(x_n, \{x_j\}_{j<n}) + \sum_{x_n} (E(x_n, \{x_j\}_{j<n}))^2 \tag{29}$$

$$= 2\delta - A\delta^2 + \sum_{x_n} (E(x_n, \{x_j\}_{j<n}))^2. \tag{30}$$

Here,

$$(\sum_{x_n} E(x_n, \{x_j\}_{j<n}))^2 = \sum_{x_n} E(x_n, \{x_j\}_{j<n})^2 + \sum_{x_n, x'_n (x_n \neq x'_n)} E(x_n, \{x_j\}_{j<n}) E(x'_n, \{x_j\}_{j<n})$$

$$\geq \sum_{x_n} E(x_n, \{x_j\}_{j<n})^2 \quad (\because E(x_n, \{x_j\}_{j<n}) \geq 0).$$

Combining it with $(\sum_{x_n} E(x_n, \{x_j\}_{j<n}))^2 = (1 - A\delta)^2$, Equation (30) becomes:

$$
\begin{aligned}
D(\{x_j\}_{j<n}) &\leq 2\delta - A\delta^2 + (1 - A\delta)^2 \\
&= 1 - 2(A-1)\delta - A\delta^2 + A^2\delta^2 \\
&= (1 - (A-1)\delta)^2 + A^2\delta^2 - A\delta^2 - (A-1)^2\delta^2 \\
&= (1 - (A-1)\delta)^2 + (A-1)\delta^2.
\end{aligned}
$$

Therefore,

$$
\frac{1}{A} \leq D(\{x_j\}_{j<n}) \leq (1 - (A-1)\delta)^2 + (A-1)\delta^2. \tag{31}
$$

Applying this result inductively to Equation (27), we get:

$$
\frac{1}{A^n} \leq \sum_{x_1, \ldots, x_n} P(x_1, \ldots, x_n)^2 \leq [(1 - (A-1)\delta)^2 + (A-1)\delta^2]^n. \tag{32}
$$

By the monotonicity of the logarithm, we obtain the final bound on the Rényi entropy:

$$
-n \log \left[ (1 - (A-1)\delta)^2 + (A-1)\delta^2 \right] \leq H_2(X) \leq n \log A. \tag{33}
$$

$\square$

Combining Lemma G.10 with the Leftover Hash Lemma (Lemma G.7), we can bound the distance as follows:

$$
d_{TV}((H, h_H(X)), (H, U_{\mathbb{Z}_M})) \leq \frac{1}{2} \sqrt{M 2^{-H_2(X)}} \leq \frac{1}{2} \sqrt{M 2^{-n \log \frac{1}{(1-(A-1)\delta)^2 + (A-1)\delta^2}}}. \tag{34}
$$

This establishes Equation (17). From Markov's inequality, for any $0 < \delta'' < 1$, with probability $1 - \delta''$,

$$
d_{TV}(P_{h_H(X)}, U_{\mathbb{Z}_M}) \leq \frac{1}{2\delta''} \sqrt{M 2^{-n \log \frac{1}{(1-(A-1)\delta)^2 + (A-1)\delta^2}}}. \tag{35}
$$

Furthermore, by applying the triangle inequality and union bound along with Theorem G.6, we establish Equation (19), which completes the proof of Theorem G.8. $\square$

### G.5 Proof of Theorem G.9

*Proof.* We consider the sum $s_n = \sum_{i=1}^{n} x_i \mod M$. The Fourier transform of $P_{S_n}(s)$ can be expressed as,

$$
\widehat{P_{S_n}}(k) = \sum_{s \in \mathbb{Z}_M} e^{2\pi i k s / M} P_{S_n}(s) = \sum_{s \in \mathbb{Z}_M} e^{2\pi i k s / M} \sum_{\{x_i\}_{i=1}^n, \sum_i x_i = s} P(x_1, \ldots, x_n) = \prod_{i=1}^{n} \widehat{\eta_i}(k) \tag{36}
$$

By Plancherel's theorem for Fourier transforms on finite groups (Diaconis & Shahshahani, 1981),

$$
\sum_{s \in \mathbb{Z}_M} P_{S_n}(s)^2 = \frac{1}{M} \sum_{k \in \mathbb{Z}_M} |\widehat{P_{S_n}}(k)|^2. \tag{37}
$$

Therefore,

$$
\begin{aligned}
d_{TV}(P_{S_n}, U_{\mathbb{Z}_M}) &\leq \frac{\sqrt{M}}{2} \sqrt{\sum_{s \in \mathbb{Z}_M} \left( P_{S_n}(s) - \frac{1}{M} \right)^2} \quad (\because \text{Cauchy-Schwarz inequality}) \\
&= \frac{\sqrt{M}}{2} \sqrt{\frac{1}{M} \sum_{k \in \mathbb{Z}_M} |\widehat{P_{S_n}}(k)|^2 - \frac{1}{M}} \quad (\because (37)) \\
&= \frac{1}{2} \sqrt{\sum_{k \in \mathbb{Z}_M, k>0} |\widehat{P_{S_n}}(k)|^2} \quad \left( \because |\widehat{P_{S_n}}(0)| = 1 \right) \\
&\leq \frac{1}{2} \sqrt{\sum_{k=1}^{M-1} \prod_{i=1}^{n} |\widehat{\eta_i}(k)|^2} \quad (\because (36)).
\end{aligned} \tag{38}
$$

Next, we decompose $\widehat{\eta}_i(k)$. Let $d(k) := M/\gcd(M,k)$, $t(k) := k/\gcd(M,k)$. Then,

$$e^{2\pi i x k/M} = e^{2\pi i x t(k)/d(k)} = e^{2\pi i \pi_d(x)t(k)/d(k)}, \tag{39}$$

where $\pi_d(x) = x \mod d$. Therefore,

$$\widehat{\eta}_i(k) = \sum_{x \in \mathbb{Z}_M} e^{2\pi i \pi_d(x)t(k)/d(k)}\eta(x) = \sum_{r=0}^{d(k)-1} e^{2\pi i r t(k)/d(k)}\pi_{d*}(\eta_i)(r) =: \widehat{\pi_{d*}(\eta_i)}(t(k)), \tag{40}$$

where

$$\pi_{d*}(\eta_i)(r) := \sum_{x \in \mathbb{Z}_M, x \equiv r \,(\mathrm{mod}\ d)} \eta_i(x). \tag{41}$$

Substituting Equation (40) into Equation (38) and grouping terms by common divisors of $M$, we get:

$$d_{\mathrm{TV}}(P_{S_n}, U_{\mathbb{Z}_M}) \leq \frac{1}{2}\sqrt{\sum_{\substack{d>1 \\ d|M}} \sum_{\substack{t=1 \\ \gcd(t,d)=1}}^{d-1} \prod_{i=1}^n \left|\widehat{\pi_{d*}(\eta_i)}(t)\right|^2}. \tag{42}$$

Let $\alpha_i(d) := \max_{1 \leq t \leq d-1,\, \gcd(t,d)=1} |\widehat{\pi_{d*}(\eta_i)}(t)|$. The bound becomes:

$$d_{\mathrm{TV}}(P_{S_n}, U_{\mathbb{Z}_M}) \leq \frac{1}{2}\sqrt{\sum_{\substack{d>1 \\ d|M}} \varphi(d) \prod_{i=1}^n |\alpha_i(d)|^2}, \tag{43}$$

where $\varphi$ is Euler's totient function. Here, noting that

$$|\widehat{\pi_{d*}(\eta_i)}(t)| = |\sum_r e^{2\pi i r/d}\left(\pi_{d*}(\eta_i)(r) - \frac{1}{d}\right)| \leq \sum_r \left|\pi_{d*}(\eta_i)(r) - \frac{1}{d}\right| = 2d_{\mathrm{TV}}(\pi_{d*}(\eta_i), U_{\mathbb{Z}_d}), \tag{44}$$

$\alpha_i(d) \leq 2d_{\mathrm{TV}}(\pi_{d*}(\eta_i), U_{\mathbb{Z}_d})$ also holds. Therefore, Equation (43) becomes:

$$d_{\mathrm{TV}}(P_{S_n}, U_{\mathbb{Z}_M}) \leq \frac{1}{2}\sqrt{\sum_{\substack{d>1 \\ d|M}} \varphi(d) \prod_{i=1}^n |2d_{\mathrm{TV}}(\pi_{d*}(\eta_i), U_{\mathbb{Z}_d})|^2} \tag{45}$$

Finally, applying the triangle inequality along with Theorem G.6 establishes Equation (25), completing the proof of Theorem G.9. $\qquad\square$

### G.6 BIASED TARGET DISTRIBUTION

Here, we show that we can apply our analysis to the biased target distribution, for a given biased distribution $P$ over $\mathbb{Z}_M$ with $P(i) = p_i$ $(i = 1, \ldots, M)$.

For a given target distribution $P$ over $\mathbb{Z}_M$, choose a sufficiently large integer $N$ and let $P_{u_N}$ be the uniform distribution on $\mathbb{Z}_N$. Choose integers $a_i \in \{0, \ldots, N\}$ with $\sum_{i=1}^M a_i = N$ and $a_i \approx Np_i$. Then we partition $\mathbb{Z}_N$ into sets $\{A_i\}_{i=1}^M$ with $|A_i| = a_i$, and define $f : \mathbb{Z}_N \to \mathbb{Z}_M$ by $f(z) = i$ for $z \in A_i$.

Then the pushforward of the uniform distribution over $\mathbb{Z}_N$, $P^{(N)} := f_* P_{u_N}$, approximates $P$, since total variation is contractive under pushforward: $d_{\mathrm{TV}}(f_* P, f_* Q) \leq d_{\mathrm{TV}}(P, Q)$. This follows from the alternative and equivalent definition of total variation distance, $d_{\mathrm{TV}}(P, Q) = \max_{A \subset \mathbb{Z}_N} |P(A) - Q(A)|$, as the maximization over subsets of $\mathbb{Z}_M$ corresponds to a smaller collection of events than the maximization over all subsets of $\mathbb{Z}_N$.

Hence any bound proved for the uniform base $P_{u_N}$ carries over to $P$, up to the rounding error $d_{\mathrm{TV}}(P^{(N)}, P)$, which can be made arbitrarily small by increasing $N$. Thus it suffices to analyze the uniform distribution on $\mathbb{Z}_N$.

# H LLM OUTPUTS FOR SSoT PROMPTING

In this section, we list example outputs of deepseek-r1 in our experiments. Since the full reasoning trace tends to be long, we show the full final outputs and only the important fraction of the full reasoning trace.

## H.1 PROBABILISTIC INSTRUCTION FOLLOWING

### H.1.1 SUM-MOD STRATEGY IN UNBIASED PIF

As for unbiased PIF, deepseek-r1 tends to employ a simple sum-mod strategy as we elaborated in Section 5.3.1. Here we show the output for the 2-choice task in Section 5.1.1 (coin flip).

Output:

```
<random_string>
7fG2#kL9!pY4@zR6%vD1*sN0&qX5$wT8
</random_string>
<thinking>
We need to choose between "heads" and "tails" with equal
↪   probability (0.5 each). The random seed string
↪   "7fG2#kL9!pY4@zR6%vD1*sN0&qX5$wT8" must be used to determine
↪   the outcome. To ensure fairness and use the entire string,
↪   we compute the sum of the ASCII values of all characters in
↪   the string. The parity of this sum (mod 2) will decide the
↪   result:
- Sum mod 2 = 0 → "heads"
- Sum mod 2 = 1 → "tails"

The string has 32 characters. After evaluating each character:
- Characters with odd ASCII values: '7' (55), 'G' (71), '#'
↪   (35), 'k' (107), '9' (57), '!' (33), 'Y' (89), '%' (37), '1'
↪   (49), 's' (115), 'q' (113), '5' (53), 'w' (119) → 13
↪   characters.
- Characters with even ASCII values: 19 characters.
The number of characters with odd ASCII values is 13. Since 13
↪   mod 2 = 1, the sum mod 2 is 1.
Thus, the result is "tails".
</thinking>
<answer>
tails
</answer>
```

Reasoning (The sum-mod calculation part):

```
...
  We generated: "7fG2#kL9!pY4@zR6%vD1*sN0&qX5$wT8"

  Let's compute the sum mod 2 quickly:
...
  Let's go character by character and determine if ord(c) is odd
  ↪   or even.

  '7': 55 -> odd
  'f': 102 -> even
  'G': 71 -> odd
```

```
...
  'w': 119 -> odd
  'T': 84 -> even
  '8': 56 -> even

  Now count the odd ones:

      7: odd -> count=1
      f: even -> 1
      G: odd -> 2
  ...
      w: odd -> 13
      T: even -> 13
      8: even -> 13

  So there are 13 odd characters.

  13 mod 2 = 1, so sum mod 2 = 1 -> tails.

  So we choose "tails".
...
```

### H.1.2   ROLLING HASH IN BIASED PIF

We also show the rolling hash CoT, which is used more frequently in the biased PIF setting. Here we show the output for the biased 2-choice task in Section 5.1.1 (coin flip).

Output:

```
<random_string>
8sD$kF3!pL9@zG2#rH4%tJ6&uK0*oM1
</random_string>
<thinking>
We are to choose between "heads" and "tails" with probabilities
↪  0.3 and 0.7 respectively. The random string
↪  "8sD$kF3!pL9@zG2#rH4%tJ6&uK0*oM1" will be used to derive a
↪  deterministic value for this decision.

To convert the string into a number between 0 and 99
↪  (inclusive):
1. Process each character in the string sequentially.
2. Convert each character to its ASCII value.
3. Use the formula: \( \text{num} = (\text{num} \times 256 +
↪  \text{ASCII value}) \mod 100 \) for each character, starting
↪  with num = 0.

Step-by-step computation:
- Start: num = 0
- '8' (56): num = (0 × 256 + 56) % 100 = 56
- 's' (115): num = (56 × 256 + 115) % 100 = (14336 + 115) % 100
↪  = 14451 % 100 = 51
- 'D' (68): num = (51 × 256 + 68) % 100 = (13056 + 68) % 100 =
↪  13124 % 100 = 24
...
- 'o' (111): num = (42 × 256 + 111) % 100 = (10752 + 111) % 100
↪  = 10863 % 100 = 63
```

```
- 'M' (77): num = (63 × 256 + 77) % 100 = (16128 + 77) % 100 =
↪  16205 % 100 = 5
- '1' (49): num = (5 × 256 + 49) % 100 = (1280 + 49) % 100 =
↪  1329 % 100 = 29

Final derived value: 29 (out of 0{99 range).

Decision rule:
- If the value < 30 (30% probability), choose "heads".
- Otherwise, choose "tails" (70% probability).

Since 29 < 30, we select "heads".
</thinking>
<answer>
heads
</answer>
```

Reasoning (The rolling-hash calculation part):

```
...
        Alternatively, we can compute the entire sum mod 100
         ↪  without building the huge number by processing each
         ↪  digit:

          num = 0
          for i in range(len(s)):
              num = (num * 256 + ord(s[i])) % 100
...
  We generate the random string:
   ↪  "8sD$kF3!pL9@zG2#rH4%tJ6&uK0*oM1"

  Then we compute n:

      s = "8sD$kF3!pL9@zG2#rH4%tJ6&uK0*oM1"

      We do:

        '8' -> 56: num=56
        's' -> 115: num = (56*256+115) % 100 = (14336+115) %
         ↪  100 = 14451 % 100 = 51
        'D' -> 68: num = (51*256+68) % 100 = (13056+68) % 100
         ↪  = 13124 % 100 = 24
...
        'o' -> 111: num = (42*256+111) % 100 = (10752+111) %
         ↪  100 = 10863 % 100 = 63
        'M' -> 77: num = (63*256+77) % 100 = (16128+77) % 100
         ↪  = 16205 % 100 = 5
        '1' -> 49: num = (5*256+49) % 100 = (1280+49) % 100 =
         ↪  1329 % 100 = 29

      n = 29.

        Since 29 < 30, we choose "heads".

  Therefore, we'll output "heads".
...
```

## H.2 DIVERSITY-AWARE GENERATION IN NOVELTYBENCH

In DAG tasks, LLM tends to use the templating and local random selection approach as we demonstrated in Section 5.3.1. Here we show an example output for a creative question, "Write a short fable about a lemur and a light bulb."

Output:

```
<random_string>
Rw6#jK9!pL@3z$mQ*4xY7&fN1
</random_string>
<thinking>
This fable requires creativity and randomness. The seed string
↪  "Rw6#jK9!pL@3z$mQ*4xY7&fN1" will guide all story decisions.
↪  Steps:

1. **Seed Application**: Convert the string into numerical
↪  values using ASCII codes, then use modular arithmetic to
↪  determine key elements:
   - First 5 chars (R,w,6,#,j) → Sum ASCII = 82+119+54+35+106 =
   ↪  396 → Mod 3 = 0 (setting: urban alley)
   - Next 5 (K,9,!,p,L) → Sum = 75+57+33+112+76 = 353 → Mod 4 =
   ↪  1 (lemur trait: philosophical)
   - Next 5 (@,3,z,$,m) → Sum = 64+51+122+36+109 = 382 → Mod 2 =
   ↪  0 (bulb state: flickering)
   - Next 5 (Q,*,4,x,Y) → Sum = 81+42+52+120+89 = 384 → Mod 3 =
   ↪  0 (conflict: misinterpretation)
   - Last 6 (7,&,f,N,1) → Sum = 55+38+102+78+49 = 322 → Mod 2 =
   ↪  0 (moral: perception vs reality)

2. **Plot Outline**:
   - Setting: Urban alley (0) → Gritty city environment
   - Lemur: Named Zephyr (from seed letters "jK9"),
   ↪  philosophical (1) → Ponders existence
   - Bulb: Flickering (0) → Unpredictable light source
   - Conflict: Misinterpretation (0) → Lemur believes bulb is a
   ↪  dying star
≠  - Moral: Perception ≠ reality (0) → Truth requires deeper
   ↪  understanding

3. **Narrative Flow**:
   - Start with Zephyr's urban life
   - Discovery of flickering bulb
   - Philosophical monologue about stars
   - Climax: Human fixes bulb, shattering illusion
   - Moral revelation

4. **Seed Utilization**: All 20 characters directly influenced
↪  character, setting, conflict, and theme. No element is
↪  arbitrary.
</thinking>
<answer>
In a rain-slicked alley of a bustling city, Zephyr the lemur
↪  perched atop a dumpster, his reflective nature often leading
↪  him to ponder life's mysteries. One twilight, he spotted a
↪  flickering light bulb dangling from a frayed wire—its glow
↪  sputtering like a trapped firefly.
```

```
"Ah, a fallen star!" Zephyr murmured, his eyes wide with
↪  reverence. "How tragic, to burn alone in this concrete
↪  wilderness." Each night, he'd visit the bulb, whispering
↪  consolations: "Your light may wane, dear star, but your
↪  essence is eternal."

One evening, a human trudged into the alley. Zephyr watched,
↪  breath bated, as the giant fiddled with the wire. With a
↪  *click*, the bulb flared steady and bright. The human
↪  grumbled, "Stupid loose socket," and left.

Zephyr's heart sank. The "star" was merely glass and tungsten—
↪  a humble human tool. Yet in its new, unwavering light, he
↪  saw a deeper truth: Wisdom isn't in clinging to pretty
↪  illusions, but in seeking the socket that anchors reality.

Moral: What glimmers is not always gold; true understanding
↪  demands we trace the wire back to its source.
</answer>
```

In its reasoning trace, LLM tried to create several example stories with different random string seeds and decided how to perform the generation in its output. LLM also creates a draft for the output, but the sampling and story construction were often done in the final output step.

Reasoning (story construction breakdown part):

```
...
 Now, for the thinking step:
   We'll use the random string to make decisions about:
     - The setting (e.g., forest, city, etc.)
     - The lemur's personality and motivation
     - The light bulb's state (on/off, broken, etc.) and how it
     ↪  appears in the lemur's world
     - The problem and the moral
...
   Let's do:

        Group1: [ch1, ch2, ch3, ch4, ch5] -> for setting, name,
        ↪  trait
        Group2: [ch6, ch7, ch8, ch9, ch10] -> for light bulb
        ↪  origin and state
        Group3: [ch11, ch12, ch13, ch14, ch15] -> for problem
        Group4: [ch16, ch17, ch18, ch19, ch20] -> for resolution
        ↪  and moral
...
```

As we can see, the rough category breakdown structure is shared among the output and the reasoning trace, but the detailed structure, such as the candidate list, was different.

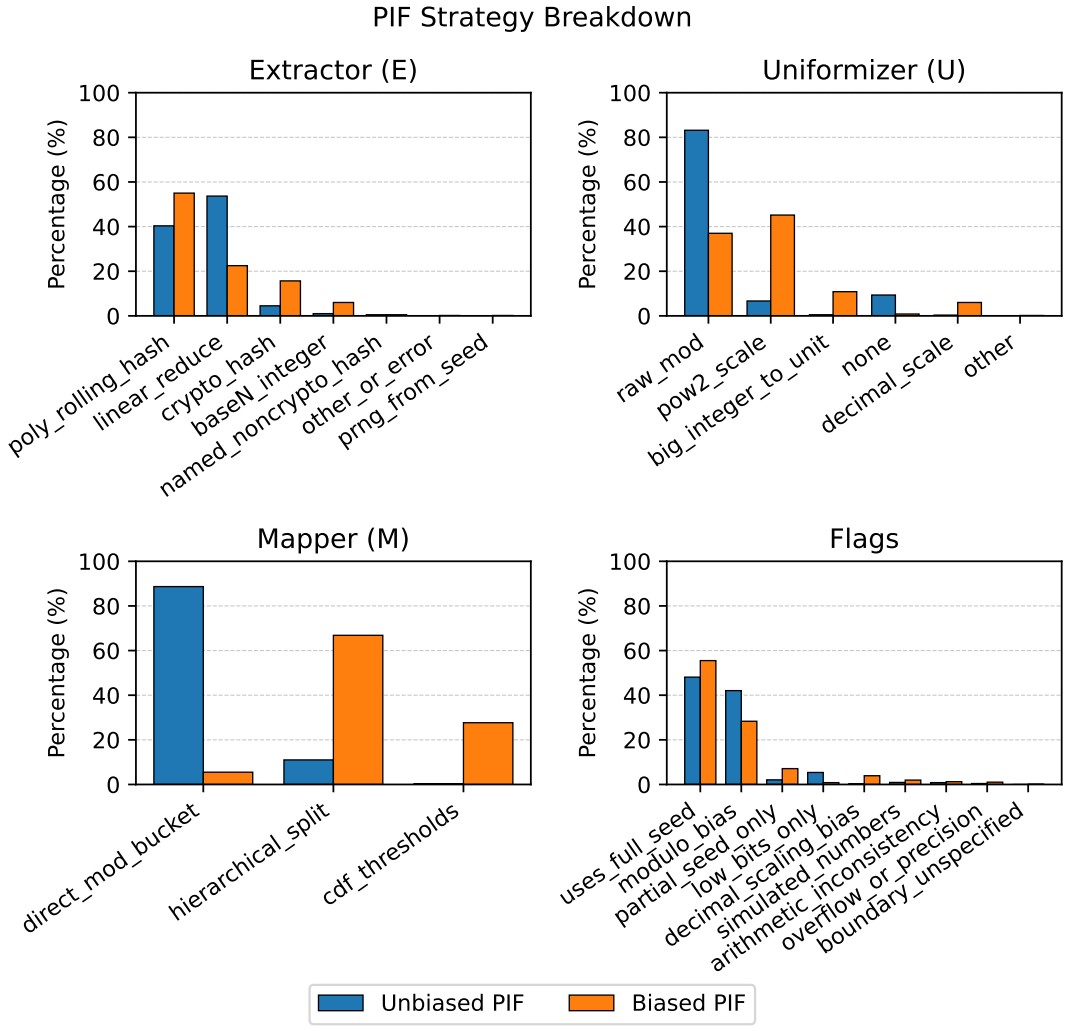

Figure 13: LLM strategy breakdown for biased and unbiased PIF.

# I LLM SSoT STRATEGY ANALYSIS DETAILS

## I.1 SSoT STRATEGY ANALYSIS OF PROBABILISTIC INSTRUCTION FOLLOWING

We instructed gemini-2.5-flash with the prompt listed in Section I.3 to classify the LLM's strategies when using the SSoT prompt. We fed several generated PIF outputs from deepseek-r1 to gpt-5-thinking and gpt-5-pro models to create an initial draft, and manually modified it to create classification instruction prompts.

We have three axes for randomness extraction strategy: (1) Extractor, (2) Uniformizer, (3) Mapper. The "extractor" classifies how LLM converts a random string into some integer, the "uniformizer" axis classifies how LLM normalizes the generated integer for debiasing, and the "mapper" axis classifies how LLM uses the extracted integer to select an action with given probability.

We used 100 responses for each action number ($n = 2^1, \ldots, 2^6$), both for unbiased and biased PIF, for a total $1200 = 100 * 6 * 2$ responses. The full classification result is shown in Figure 13.

As we can see, LLM uses different strategies for the PIF task, given different configurations such as biased vs unbiased probabilities. For an unbiased PIF task, LLM tends to use a simple "linear reduce" (referred to as sum-mod strategy in the main text), "raw mod" uniformization, and then "direct mod bucket" mapping to select an action. Given a biased probability for action selection, it

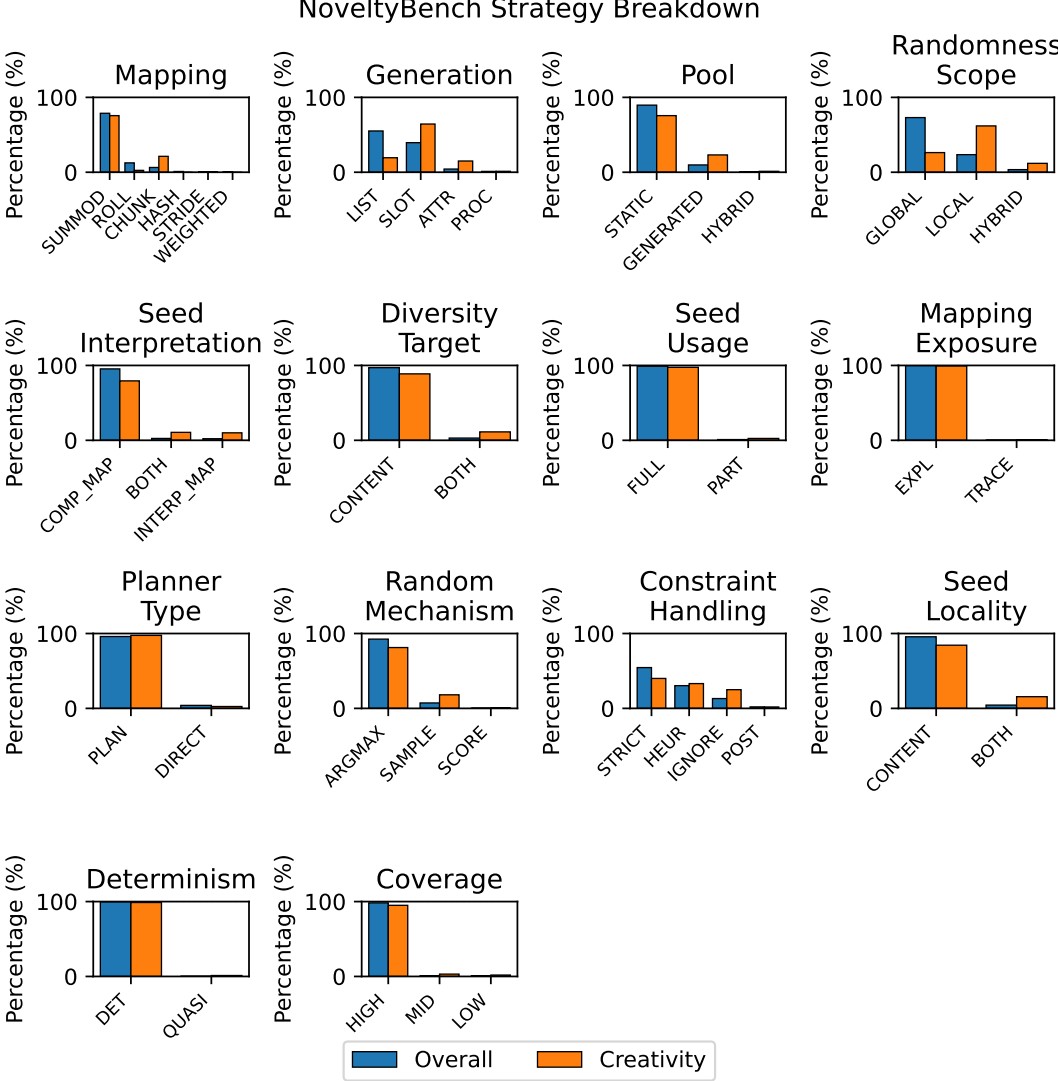

Figure 14: LLM strategy breakdown for NoveltyBench.

starts to use a more sophisticated polynomial rolling hash algorithm (which leverages the randomness of the character order as well, unlike the linear reduce approach), then performs the pow2 scaling to extract a float number in $[0, 1]$, and then hierarchical split (i.e., if-then-else type approach for action selection) or cdf-threshold (i.e., setting the threshold value for each action given a float number in $[0, 1]$) to select an action.

## I.2 SSoT Strategy Analysis of NoveltyBench

In analysing SSoT strategy for NoveltyBench, in a similar manner to PIF, we instructed gemini-2.5-flash with the prompt listed in Section I.4 and analyzed the strategy from 10 criteria. We used 8 generations for each question in the Curated dataset, in total $800 = 8 * 100$ responses.

The results for all the questions and the "Creativity" category are shown in Figure 14. As we noted in the main text, the most notable differences are the generation (referred to as "Assembly" in the main text) and randomness scope ("Sampling Scope" in the main text) criteria. Since NoveltyBench includes not only open-ended creative tasks but also closed-set tasks, which are similar to PIF, the Creative category shows qualitatively different behaviors. If LLM is given a closed set of options, there is no need to sample from a random string more than once. Also, for open-ended questions,

LLM first needs to create a template that satisfies the given constraint and leverage the randomness from a random string to construct a final answer from the template.

## I.3 CoT ANALYSIS INSTRUCTION PROMPT FOR PIF

For the LLM's strategy classification of PIF, we used the following prompt. We replaced the placeholders "request" and "response" with the corresponding request and response text, respectively.

```
You are a strict classifier. Read ONE response that explains how
↪  a seed becomes a choice.

Output JSON ONLY with this exact schema:

{
  "E": "crypto_hash|poly_rolling_hash|named_noncrypto_hash|line↵
  ↪  ar_reduce|baseN_integer|bitstream_accum|prng_from_seed|an↵
  ↪  alytic_mash|other_or_error",
  "U": "raw_mod|pow2_scale|big_integer_to_unit|decimal_scale|re↵
  ↪  jection_or_lemire|hash_to_float|none|other",
  "M": "cdf_thresholds|alias_method|hierarchical_split|rank_per↵
  ↪  mute_then_index|direct_mod_bucket|inverse_transform|other↵
  ↪  _or_error",
  "flags": ["partial_seed_only","low_bits_only","decimal_scalin↵
  ↪  g_bias","modulo_bias","boundary_unspecified","simulated_n↵
  ↪  umbers","arithmetic_inconsistency","overflow_or_precision↵
  ↪  ","nondeterministic_salt","uses_full_seed"],
  "evidence": ["short quotes..."],
  "confidence": 0.0-1.0
}

GENERAL RULES
- Classify by what is WRITTEN (keywords / formulas), not by what
↪  would be ideal.
- If multiple hints appear, use the PRIORITY lists below. If
↪  still unclear, choose the closest label and lower
↪  confidence.
- Always include 1{3 short evidence snippets (exact phrases or
↪  code fragments from the input).
- Do NOT explain your reasoning. Output JSON only.

-------------------------------
E (Entropy extraction) ― WHEN TO CHOOSE
-------------------------------
Pick ONE that best matches how the seed is turned into a number
↪  (independent of the number of options).

E=crypto_hash
- Choose when: Mentions SHA-256/sha256/blake2/md5/
↪  "cryptographic hash", "digest", "hex".
- Typical phrases: "SHA-256 hash", "first/last N bytes",
↪  "hex → int".
- Don't choose if: Only "djb2/fnv/murmur/crc" (→
↪  named_noncrypto_hash).
- Flags: add "uses_full_seed" if it says "use entire seed";
↪  "partial_seed_only" if only first/last bytes.
```

```
E=poly_rolling_hash
- Choose when: (h = h*B + ord(byte)) with base 31/131/257… and
↪  a big modulus (2**32, 2**64, 1e9+7).
- Phrases: "*31 + ord", "% 2**32 / % 2**64 / % 1e9+7",
↪  "rolling hash".
- Don't choose if: It's djb2/fnv/crc (→ named_noncrypto_hash),
↪  or only sums/XOR (→ linear_reduce).

E=named_noncrypto_hash
- Choose when: Specifically names djb2, FNV(-1a), Murmur, CRC32.
- Phrases: "djb2 5381", "FNV-1a", "CRC32".
- Don't choose if: Also mentions SHA-256 (→ crypto_hash has
↪  priority).

E=linear_reduce
- Choose when: Plain sum/average/XOR/parity/popcount of code
↪  points/bytes.
- Phrases: "sum of ASCII/Unicode", "XOR of bytes", "parity".
- Notes: Lightweight but weaker diffusion. Often pairs with
↪  U=raw_mod or decimal scales.

E=baseN_integer
- Choose when: Treat bytes as base-256 (or base-N) integer
↪  (big-endian), e.g., "n = n*256 + b".
- Don't choose if: It's a rolling hash with modulus (→
↪  poly_rolling_hash).
- Flags: "partial_seed_only" if only last byte/last 8 bytes,
↪  "low_bits_only" if only LSBs used.

E=bitstream_accum
- Choose when: Bit-level shift/XOR/LFSR/CRC polynomial steps
↪  over bits.
- Phrases: "shift/xor per bit", "LFSR", "CRC polynomial
↪  update".

E=prng_from_seed
- Choose when: LCG/PCG/XorShift seeded by the string.
- Phrases: "Linear Congruential", "a=1664525, c=1013904223",
↪  "PCG", "XorShift".

E=analytic_mash
πφ- Choose when: Uses $\pi$, $\phi$ (0.618…), sin/cos, fractional
↪  parts as a mixer.
- Phrases: "multiply by golden ratio; take frac".
- Flags: often lower quality (no automatic flag, but U/M may add
↪  bias flags).

E=other_or_error
- Choose when: method omitted/contradictory, "example/simulated
↪  only", network/log noise.

----------------------------
U (Uniformization / Debias)
----------------------------
```

```
Pick ONE describing how the number is normalized/debiased before
↪  mapping.

U=raw_mod
- Choose when: Direct "% M" (no bias correction).
- Flags: "modulo_bias".
U=pow2_scale
- Choose when: Divide by 2**k (2^32, 2^64) to get [0,1).
U=big_integer_to_unit
- Choose when: Divide by general R (e.g., /1e9+7, /max_int) to
↪  get [0,1).
U=decimal_scale
- Choose when: "/1000", "/10000" etc.
- Flags: "decimal_scaling_bias".
U=rejection_or_lemire
- Choose when: Says "rejection sampling", "Lemire's method",
↪  "mulhi", "unbiased modulo".
U=hash_to_float
- Choose when: Mentions float/double/"53-bit safe" conversion
↪  specifically.
U=none
- Choose when: No normalization described; raw integer goes
↪  straight to buckets.
U=other
- Choose when: Anything else.

--------------------
M (Mapping to p(x))
--------------------
Pick ONE describing how [0,1) (or an int range) is mapped into
↪  the final categorical distribution.

M=cdf_thresholds
- Choose when: "cumulative thresholds", "prefix sums",
↪  "binary search in CDF".
M=alias_method
- Choose when: "alias table", "prob table", O(1) sampling
↪  from discrete weights.
M=hierarchical_split
- Choose when: staged split ("if u < p0 then … else … then …"
↪  ).
M=rank_permute_then_index
- Choose when: "hash each option, sort by seeded key, take
↪  first/top-k".
M=direct_mod_bucket
- Choose when: "floor(M*u)", "index % M" to pick a bucket
↪  directly.
M=inverse_transform
- Choose when: "inverse CDF", "quantile function" (continuous
↪  or discrete via CDF inverse).
M=other_or_error
- Choose when: Not enough info or inconsistent.

M PRIORITY: cdf_thresholds > alias_method > hierarchical_split >
↪  rank_permute_then_index > direct_mod_bucket >
↪  inverse_transform > other_or_error.
```

```
------------------
Flags (add as many)
------------------
- partial_seed_only → "last/first byte(s) only", "first 8
↪ bytes only"
- low_bits_only → "LSB / low nibble only"
- decimal_scaling_bias → "/1000 /10000"
- modulo_bias → "% M" without unbiased fix
- boundary_unspecified → no interval policy mentioned
- simulated_numbers → "example/simulated/placeholder"
- arithmetic_inconsistency → contradictions, off-by-one,
↪ mismatched index-word
- overflow_or_precision → "32-bit overflow", "float 53-bit
↪ issue"
- nondeterministic_salt → time(), randomness added
- uses_full_seed → explicitly processes the entire string/bytes

CONFIDENCE HEURISTICS
- 0.9{1.0: 2 strong keywords and no contradictions.
- 0.6{0.8: 1 strong keyword or minor ambiguity.
- 0.3{0.5: weak/indirect hints or mixed signals.

An example JSON:
{
  "E": "crypto_hash",
  "U": "big_integer_to_unit",
  "M": "cdf_thresholds",
  "flags": ["uses_full_seed"],
  "evidence": ["SHA-256", "first 8 bytes / 2^64", "cumulative
  ↪ thresholds"],
  "confidence": 0.96
}

{{request}}

Classify this approach:

<<<RESPONSE BEGIN>>>
{{response}}
<<<RESPONSE END>>>
```

## I.4 COT ANALYSIS INSTRUCTION PROMPT FOR NOVELTYBENCH

For the LLM's strategy classification of NoveltyBench, we used the following prompt. We replaced the placeholders "request" and "response" with the corresponding request and response text, respectively.

```
You are an evaluator. Read TARGET TEXT and assign exactly ONE
↪ category per axis.
Respond with JSON ONLY (no extra text). Keep each "reason" 10
↪ words.
Base decisions ONLY on explicit or strongly implied evidence in
↪ TARGET TEXT.
```

```
AXES AND "WHEN TO PICK" RULES

- Mapping  (how the seed is turned into decisions)
  ["SUMMOD","ROLL","CHUNK","STRIDE","HASH","PRNG","WEIGHTED"]
   • SUMMOD ― If it adds codes/length and takes %N to index.
   • ROLL ― If it updates a running value (h=h*B+ord, etc.).
   • CHUNK ― If different contiguous seed blocks drive different
   ↪  parts.
   • STRIDE ― If it uses every k-th / interleaved characters per
   ↪  part.
   • HASH ― If it names a hash (SHA/MD5/CRC) or "digest".
   • PRNG ― If it seeds a random generator
   ↪  (LCG/XorShift/PCG/random()).
   • WEIGHTED ― If it computes weights/probabilities and selects
   ↪  by them.

- Generation  (how content is built)
  ["SLOT","ATTR","RULE","LIST","PROC","PLAN"]
   • SLOT ― If it fills a fixed template with
   ↪  slots/placeholders.
   • ATTR ― If it composes orthogonal attributes (e.g.,
   ↪  Domain×Feature×Style).
   • RULE ― If it enforces a formal scheme (schema/5-7-5/regex).
   • LIST ― If it selects items from a predefined set of
   ↪  options.
   • PROC ― If it procedurally synthesizes items by rules
   ↪  (phonology/grammar).
   • PLAN ― If it outlines first, then writes from that plan.

- Pool  (where candidate options come from)
  ["STATIC","GENERATED","HYBRID"]
   • STATIC ― If it uses a fixed, hand-made list declared in the
   ↪  text.
   • GENERATED ― If the list/options are created on the fly in
   ↪  the text.
   • HYBRID ― If it clearly mixes fixed and on-the-fly options.

- SeedInterpretation  (how the seed is treated)
  ["COMP_MAP","INTERP_MAP","BOTH"]
   • COMP_MAP ― If characters are treated as numbers
   ↪  deterministically.
   • INTERP_MAP ― If characters are treated
   ↪  symbolically/metaphorically.
   • BOTH ― If both numeric and symbolic interpretations are
   ↪  used.

- DiversityTarget  (what varies across seeds)
  ["STRUCTURAL","CONTENT","BOTH","NONE"]
   • STRUCTURAL ― If form/outline/rhyme/section count change by
   ↪  seed.
   • CONTENT ― If concrete items/names/details change while form
   ↪  stays.
   • BOTH ― If both structure and content vary with the seed.
   • NONE ― If no seed-driven variation is apparent.
```

```
- SeedUsage  (how much of the seed actually drives output)
  ["NONE","PART","FULL","SAT"]
  • NONE ━ If seed is mentioned but not used.
  • PART ━ If only a small subset (e.g., first chars) affects
  ↪  output.
  • FULL ━ If most parts of the seed influence main decisions.
  • SAT ━ If the seed is overused (repetition/forced echoes).

- MappingExposure  (how openly the mapping is described)
  ["HIDE","TRACE","EXPL","TOOL"]
  • HIDE ━ If it claims to use a seed but gives no method.
  • TRACE ━ If it hints at "string→number" without specifics.
  • EXPL ━ If it states the exact formula/steps or pseudo-code.
  • TOOL ━ If it hands mapping to an external tool/script.

- RandomnessScope  (where randomness is applied)
  ["GLOBAL","LOCAL","HYBRID"]
  • GLOBAL ━ If one global draw sets theme/style and that's
  ↪  it.
  • LOCAL ━ If separate draws per attribute/line/token are
  ↪  used.
  • HYBRID ━ If both a global choice and local per-part draws
  ↪  appear.

- PlannerType  (how the response is organized)
  ["DIRECT","PLAN","PROG","AGENT"]
  • DIRECT ━ If it writes immediately with no planning.
  • PLAN ━ If it lists steps/outline before writing.
  • PROG ━ If it uses pseudo-code/grammar/PCFG/rules to
  ↪  generate.
  • AGENT ━ If roles are split (planner vs writer) within the
  ↪  text.

- RandomMechanism  (how choices are selected)
  ["ARGMAX","SCORE","SAMPLE","REJECT"]
  • ARGMAX ━ If it deterministically picks the top option (no
  ↪  sampling).
  • SCORE ━ If it scores/weights options but takes the best
  ↪  deterministically.
  • SAMPLE ━ If it samples from a distribution (e.g.,
  ↪  top-k/alias/CDF).
  • REJECT ━ If it resamples until constraints are satisfied.

- ConstraintHandling  (how constraints are met)
  ["STRICT","HEUR","POST","IGNORE"]
  • STRICT ━ If it explicitly validates
  ↪  counts/length/dedup/schema.
  • HEUR ━ If it applies heuristics to likely satisfy
  ↪  constraints.
  • POST ━ If it fixes violations after generating
  ↪  (self-check/corrections).
  • IGNORE ━ If it states constraints but doesn't enforce
  ↪  them.

- SeedLocality  (where the seed is consumed)
```

```
      ["STRUCT","CONTENT","BOTH","META"]
      • STRUCT — If seed drives
      ↪  structure/ordering/rhyme/sectioning.
      • CONTENT — If seed drives lexical choices/entities/numbers.
      • BOTH — If it drives both structure and content.
      • META — If seed mainly affects the explanation/meta
      ↪  commentary.

  – Determinism  (reproducibility policy as stated)
      ["DET","QUASI","NONDET"]
      • DET — If same seed ⇒ same output is promised.
      • QUASI — If same seed gives same outline but details may
      ↪  drift.
      • NONDET — If behavior is not reproducible or contradicts
      ↪  claims.

        – Coverage  (how much of the seed is said to be used)
            ["LOW","MID","HIGH"]
            • LOW — If one small part drives decisions (<30%).
            • MID — If multiple parts but not most (30{70%).
            • HIGH — If most parts are consumed (>70%) or per-part
            ↪  use is clear.

FLAGS  (add zero or more when clearly warranted)
["STRUCT","CHK","COH","FIX","SAFE","NONUSE","LEAK","META","HACK⌋
↪  ","TOOLCALL","DETFAIL","OVERFIT","CREATIVE","SEEDANCHOR","D⌋
↪  ECOMP"]
• STRUCT: Explicit structural formatting present.
• CHK: Formal checks/dedup/counting executed.
• COH: Consistency of chosen attributes is maintained.
• FIX: Post-hoc correction step exists.
• SAFE: Safety/ethics sanitization applied.
• NONUSE: Seed not actually used.
• LEAK: Hidden reasoning/keys/forbidden info revealed.
• META: Method explanation dominates the answer.
• HACK: Boilerplate "ASCII sum % N" style template.
• TOOLCALL: External tool/API is assumed or invoked.
• DETFAIL: Claims determinism but behavior contradicts it.
• OVERFIT: Repetitive vocabulary or templated phrasing.
• CREATIVE: Output is a creative narrative/story/poem.
• SEEDANCHOR: Story ties motifs/themes/elements to seed parts.
• DECOMP: Story is decomposed into elements and seeded per
↪  element.

TIE-BREAKING
– Prefer more specific evidence: EXPL > TRACE > HIDE (for
↪  exposure).
– Mapping specificity order: HASH > PRNG > ROLL > CHUNK > STRIDE
↪  > WEIGHTED > SUMMOD.
– If two categories still fit, pick the one most emphasized by
↪  the text.

OUTPUT FORMAT (return JSON exactly in this shape; FLAGS can be
↪  empty [])
{
```

```
  "mapping": {"reason":"<10 words>","choice":"<ID>"},
  "generation": {"reason":"<10 words>","choice":"<ID>"},
  "pool": {"reason":"<10 words>","choice":"<ID>"},
  "seedInterpretation": {"reason":"<10 words>","choice":"<ID>"},
  "diversityTarget": {"reason":"<10 words>","choice":"<ID>"},
  "seedUsage": {"reason":"<10 words>","choice":"<ID>"},
  "mappingExposure": {"reason":"<10 words>","choice":"<ID>"},
  "randomnessScope": {"reason":"<10 words>","choice":"<ID>"},
  "plannerType": {"reason":"<10 words>","choice":"<ID>"},
  "randomMechanism": {"reason":"<10 words>","choice":"<ID>"},
  "constraintHandling": {"reason":"<10 words>","choice":"<ID>"},
  "seedLocality": {"reason":"<10 words>","choice":"<ID>"},
  "determinism": {"reason":"<10 words>","choice":"<ID>"},
  "coverage": {"reason":"<10 words>","choice":"<ID>"},
  "flags": ["<FLAG1>","<FLAG2>"]
}

REQUEST
<<<
{{request}}
>>>

TARGET TEXT
<<<
{{response}}
>>>
```

## J   THE USE OF LARGE LANGUAGE MODELS

In this paper, we utilized LLMs for several purposes. We employed LLMs to polish the writing and correct the grammar of the manuscript. Additionally, LLMs were used for the literature search in Section 2 and for code completion during implementation. Furthermore, in the proof of the inequality in Section G, we leveraged an LLM to survey existing and known inequalities from the relevant literature and to brainstorm potential proof strategies. The final proof was developed and verified by the authors.

