# OpenReview forum: "String Seed of Thought: Prompting LLMs for Distribution-Faithful and Diverse Generation"
_ICLR.cc/2026/Conference — ICLR 2026 Poster_

### Official Review · Reviewer_KwFB · 2025-10-18

**Soundness:** 2
**Presentation:** 2
**Contribution:** 2
**Rating:** 2
**Confidence:** 3

**Summary:**

The authors of this paper introduced Seed of Thought (SSoT), a prompting method for improving LLMs' probabilistic instruction following. For example, LLMs can approximately draw 50/50 for heads or tails if the prompt is to "flip an unbiased coin". The probabilistic instruction following method proposed in this paper is done by crafting prompts: LLMs need to first generate a random sequence and then derive an answer based on the generated sequence.

**Strengths:**

1. Two theorems provide the lower bound of the total variation distance between the sample and the required distributions.
2. The experiment section is well presented overall. Figures and tables convinced me that the proposed method indeed improves the LLM's probabilistic instruction-following capabilities.
3. In addition to 2, the rich details in the appendix and the uploaded supporting material benefit the reproduction of numerical studies.

**Weaknesses:**

1. The assumptions that "each character is randomly drawn from a distribution" of the proved theorems are hard to meet for LLMs. They are pre-trained on text corpora. Every individual token generated by them is based on its previous content (Causal Language Modeling), thus never random. I can be proved wrong, at least empirically, by an analysis of the distribution/duplication of LLM-generated random strings.

2. Personally, I am not fully motivated to tune LLM to perform PIF tasks, which could be done precisely with `random` modules if the option set is predefined. This point is the major driver of my evaluation score. The authors can further elaborate on this in the motivation section, or add results for PRNG in Section 5.2, where I cannot link why PRNG is inadequate for diversity-aware generation.

3. The SSoT System prompts in Appendix A are different for each case, which weakened the generalization capability of the proposed method (thinking `please think step-by-step` for CoT). An explanation of why we need different descriptions, or how to design SSoT prompts, alleviates this weakness.

**Presentation-related Suggestions:**:
1. Several paragraphs need to be rewritten into formal academic writing (e.g. line 146, `the only thing you need to do` uses second-person narration)
2. The last panel of Figure 2 (line 186) is hard to read, as the Ideal threshold shape overlapped with the tiny bars

**Questions:**

1. I am not familiar with the TV distance bound with random hash function theories, nor did I check every step of the derivatives. Can the author briefly describe the contribution of these theorems in terms of deterministic hash decoding (or a more specific/related field), regardless of whether the LLM-instruction following can meet the assumptions or not?

---

> ### Author Response · Authors · 2025-11-21
> **Response to Reviewer KwFB (1/3)**
>
> ---
>
> We thank the reviewer for their constructive feedback and questions. We have revised the manuscript accordingly in response.
>
> ---
>
> ## Response to the reviewer's summary
> > The authors of this paper introduced Seed of Thought (SSoT), a prompting method for improving LLMs' probabilistic instruction following. For example, LLMs can approximately draw 50/50 for heads or tails if the prompt is to "flip an unbiased coin". The probabilistic instruction following method proposed in this paper is done by crafting prompts: LLMs need to first generate a random sequence and then derive an answer based on the generated sequence.
>
> We thank the reviewer for the great summary of our proposed mechanism. While the "unbiased coin flip" serves as an intuitive introductory example, we would like to take this opportunity to emphasize the broader applicability of SSoT demonstrated in our work.
>
> Beyond simple binary choices, our experiments show that SSoT is highly robust for:
> 1. **Complex PIF**: Handling biased target distributions and larger action spaces (up to 64 options), a setting where standard prompting fails.
> 1. **Diversity-Aware Generation (DAG)**: Crucially, we demonstrated that SSoT significantly improves performance in open-ended, free-form creative writing tasks via NoveltyBench, where predefined option sets do not exist.
>
> Furthermore, in this rebuttal, we have provided new evidence that SSoT improves sampling performance even for continuous distributions (Uniform, Normal, and Beta). We hope these extensive experiments demonstrate the generality of the SSoT approach. If you have any additional concerns regarding the generality, the performance, or practical utility of SSoT, please do not hesitate to let us know.
>
> ---
> ## Weaknesses
>
> ---
> ### W1: Theoretical assumptions (Randomness of tokens).
> > The assumptions that "each character is randomly drawn from a distribution" of the proved theorems are hard to meet for LLMs. They are pre-trained on text corpora. Every individual token generated by them is based on its previous content (Causal Language Modeling), thus never random. I can be proved wrong, at least empirically, by an analysis of the distribution/duplication of LLM-generated random strings.
>
> We thank the reviewer for this rigorous critique regarding the theoretical assumptions. As we mentioned in the main text, we fully agree that the autoregressive nature of LLMs introduces correlations. However, we would like to clarify how our theoretical framework specifically accounts for this and provide new empirical evidence to justify our assumptions.
>
> 1. **Theorem 4.1 (Hash Strategy)**: This theorem does not assume perfect independence. Instead, it is explicitly formulated using $\delta$-Santha-Vazirani sources, which mathematically model "semi-random" sources with weak correlations. This design choice was made precisely to accommodate the autoregressive nature of LLM generation. Furthermore, we empirically tested the validity of the $\delta$-SV assumption in Appendix E and confirmed that the generated strings approximately satisfy this condition.
> 1. **Theorem 4.2 (Sum-Mod Strategy)**: While this theorem assumes independence for analytical tractability, we empirically verified whether this assumption holds in practice. In Appendix E, we analyzed the ASCII codes of the generated strings (which are the actual inputs for the Sum-Mod operation). We found that the total variation distance between the joint distribution and the product of marginals is negligible (all $< 0.07$ for lags 1-3). Therefore, we empirically confirmed that the independence assumption holds approximately in the numerical domain used for calculation.
> 3. **Consistency with Performance**: The fact that LLMs naturally adopting the Sum-Mod strategy (e.g., deepseek-r1 in Unbiased PIF) achieve near-ideal performance (Figure 3) strongly supports our finding that the deviation from independence is small enough to be practically ignored.
>
> We believe these clarifications, combined with the new quantitative analysis in Appendix E, demonstrate that our theoretical bounds provide a valid and valuable explanation for the method's effectiveness.
>
> ---

---

> > ### Author Response · Authors · 2025-11-21
> > **Response to Reviewer KwFB (2/3)**
> >
> > ---
> > ### W2: Why not use a random module (PRNG)? (The Practicality Concern)
> >
> > > Personally, I am not fully motivated to tune LLM to perform PIF tasks, which could be done precisely with random modules if the option set is predefined. This point is the major driver of my evaluation score. The authors can further elaborate on this in the motivation section, or add results for PRNG in Section 5.2, where I cannot link why PRNG is inadequate for diversity-aware generation.
> >
> > We thank the reviewer for raising concerns regarding the practical utility of SSoT. We would like to clarify the nature of Diversity-Aware Generation (DAG) in our work, where external PRNGs are generally not applicable.
> >
> > To illustrate this, consider the open-ended tasks in NoveltyBench, such as "Write a haiku about a whale and a walnut tree." In such free-form creative writing, there is no fixed set of candidates for an external PRNG to sample from. Consequently, an external random module cannot be used to diversify the output because the options do not exist prior to generation.
> >
> > While we agree that for simple PIF tasks with a predefined option set, external random modules can serve as a substitute, SSoT addresses this critical gap in DAG. By simply modifying the prompt, SSoT generates randomness inside LLM's internal reasoning process, effectively improving diversity not only for closed-set questions but also for open-ended queries (e.g., "What's a cool fact about Pittsburgh?") and creative writing tasks where external tools fail.
> >
> > To further clarify this distinction, we have added detailed supplementary information about NoveltyBench to Appendix F.
> >
> > ---
> > ### W3 Prompt Complexity
> >
> > > The SSoT System prompts in Appendix A are different for each case, which weakened the generalization capability of the proposed method (thinking please think step-by-step for CoT). An explanation of why we need different descriptions, or how to design SSoT prompts, alleviates this weakness.
> >
> > We thank the reviewer for raising this point regarding prompt generalization.
> >
> > The variation in prompts across tasks stems from our experimental design, which was strictly intended to avoid overestimating the effect of SSoT. To ensure a fair comparison, we first constructed strong baselines tailored to each task (e.g., including specific instructions like "think step-by-step" or task-specific context). We then created the SSoT prompts by adding the random string instruction to these baselines with minimal modifications. This approach allowed us to isolate the pure contribution of the random string mechanism.
> >
> > However, to address your concern and demonstrate the generalization capability of our method, we verified that SSoT performs effectively even with a simplified, unified prompt.
> > We evaluated PIF performance using a minimal prompt that simply instructs the model to perform SSoT. The JS divergence results are shown below:
> >
> > |Model|Method|2-choice|biased 2-choice|3-choice|biased 3-choice|biased 9-choice|
> > |-|-|-|-|-|-|-|
> > | deepseek-r1| Baseline| 36.09 ± 13.60| 69.58 ± 13.42| 106.30 ± 19.45| 49.53 ± 11.24| 138.21 ± 26.64|
> > | deepseek-r1 | SSoT| 3.03 ± 3.43 (↓92%)| 1.51 ± 1.55 (↓98%)| 4.98 ± 3.84 (↓95%)| 4.30 ± 4.92 (↓91%)| **18.06 ± 11.35 (↓87%)**|
> > |deepseek-r1|**SSoT (Simple)**|**2.37 ± 2.63 (↓93%)**|**1.19 ± 1.54 (↓98%)**|**1.63 ± 1.57 (↓98%)**|**3.72 ± 3.27 (↓92%)**| 25.17 ± 9.84 (↓82%)|
> > | PRNG|-|1.85 ± 2.58|1.93 ± 2.80|3.36 ± 2.48|2.85 ± 3.15|13.72 ± 4.21|
> >
> > Furthermore, for DAG tasks, we confirmed that simply adding the objective (i.e., generate a diverse response) to the unified prompt is sufficient to improve performance. We note that this instruction is a task objective that should be included regardless of the use of SSoT.
> >
> > (Each cell shows Distinct (Utility) metrics.)
> > |Method|Creativity|Naming|Facts|Product Recs|Random|Opinions|Overall|
> > |-|-|-|-|-|-|-|-|
> > | Baseline|4.60 (5.61)|6.00 (6.13)|4.14 (5.35)|4.14 (6.02)|6.07 (5.10)|4.35 (4.08)|4.70 (5.17)|
> > |SSoT|5.90 (**6.44**)|**7.57** (**6.62**)|**6.04** (**6.71**)|**5.86** (6.63)|**6.87** (**5.49**)|5.87 (**4.35**)|6.19 (**5.92**)|
> > |**SSoT (Simple)**|**6.20** (6.38)|6.86 (5.66)|5.96 (6.49)|**5.86** (**7.32**)|6.60 (5.30)|**6.39** (4.02)|**6.26** (5.72)|
> >
> > These results demonstrate that complex, task-specific descriptions are not a requirement for SSoT; a simple, unified structure is sufficient to achieve significant improvements. We have added these additional experiments and the simplified prompts to Appendix D.4.
> >
> > ---

---

> > > ### Author Response · Authors · 2025-11-21
> > > **Response to Reviewer KwFB (3/3)**
> > >
> > > ---
> > > ## Presentation Issues
> > >
> > > > Several paragraphs need to be rewritten into formal academic writing (e.g. line 146, the only thing you need to do uses second-person narration)
> > >
> > > > The last panel of Figure 2 (line 186) is hard to read, as the Ideal threshold shape overlapped with the tiny bars
> > >
> > > We thank the reviewer for careful reading and pointing out the presentation issues. We fixed both issues in the revised manuscript as follows:
> > > 1. We fixed the paragraph in line 146 to a more formal academic style.
> > > 1. We enlarged the biased 9-choice panel of Figure 2.
> > >
> > > ---
> > > ## Questions
> > >
> > > ---
> > > ### Q1: Contribution of the Theorems
> > >
> > > > I am not familiar with the TV distance bound with random hash function theories, nor did I check every step of the derivatives. Can the author briefly describe the contribution of these theorems in terms of deterministic hash decoding (or a more specific/related field), regardless of whether the LLM-instruction following can meet the assumptions or not?
> > >
> > > We thank the reviewer for this opportunity to clarify the theoretical contributions of our work. While our proofs build upon foundational results, our specific contributions lie in adapting and extending these theories to the novel context of LLMs to provide rigorous guarantees for SSoT.
> > >
> > > **Theorem 4.1 (Hash Strategy)**
> > >
> > > Our contribution is establishing the connection between randomness extraction theory and autoregressive text generation.
> > > * **Extension to Non-Binary Alphabets**: We explicitly formulated the LLM token generation process as a $\delta$-Santha-Vazirani source over an arbitrary alphabet. This adapts the classical binary SV source model to the realistic vocabulary size of LLMs.
> > > * **Explicit Entropy and TV Bounds**: We derived the lower bound of the Rényi entropy for these sources. By applying the Leftover Hash Lemma and Weissman’s inequality, we provided a closed-form upper bound on the total variation distance. This theoretically guarantees that a hash function can extract near-uniform randomness from an autoregressive LLM, provided the sequence length $n$ is sufficient.
> > >
> > > **Theorem 4.2 (Sum-Mod Strategy)**
> > >
> > > Our contribution here is bridging the gap between theoretical mixing times and the actual arithmetic behavior of LLMs.
> > >
> > > * **Modeling "Sum-Mod" as a Random Walk**: Based on our CoT analysis, we modeled the LLM's "sum of ASCII codes modulo $M$" strategy as a random walk on the finite group $\mathbb{Z}_M$ (a cycle graph).
> > > * **Spectral Bounds for LLM Strategies**: By adapting Fourier analysis techniques (specifically the Upper Bound Lemma by Diaconis & Shahshahani), we derived a spectral bound using the Fourier coefficients of the token character distribution.
> > > * **Exponential Convergence**: We proved that even with simple arithmetic operations, the output distribution converges to uniformity exponentially fast with respect to the string length $n$, providing a solid theoretical basis for the strategy's effectiveness.
> > >
> > > We have added this discussion on the theoretical positioning of our work to Appendix G.
> > >
> > > ---

---

> > > > ### Comment · Reviewer_KwFB · 2025-11-21
> > > >
> > > > I appreciate the authors’ revisions and thank the reviewers for addressing many of my earlier concerns. The manuscript has clearly improved, thus I improved my evaluation point, but I would like to clarify my remaining hesitation.
> > > >
> > > > My primary concern continues to center on the motivation. The proposed SSoT method follows three steps:
> > > > 1. The LLM generates a random string.
> > > > 2. The LLM performs ruled transformations on this string (e.g., hashing or sum modulo operations).
> > > > 3. The LLM then generates diverse samples conditioned on steps (1) and (2).
> > > >
> > > > From the manuscript, it is empirically evident, and I believe most readers would agree, that step (3) demonstrates the LLM’s ability to perform PIF when given additional instructions and context. However, I remain unconvinced about the necessity of having the LLM itself perform steps (1) and (2). These operations can be trivially implemented using standard pseudo-random number generators and lightweight heuristic code outside the model, then inserted into the prompt at inference time.

---

> > > > > ### Author Response · Authors · 2025-11-25
> > > > > **Response to Reviewer KwFB**
> > > > >
> > > > > > The manuscript has clearly improved, thus I improved my evaluation point,
> > > > >
> > > > > We thank the reviewer for acknowledging the improvements and raising the score.
> > > > >
> > > > > > However, I remain unconvinced about the necessity of having the LLM itself perform steps (1) and (2). These operations can be trivially implemented using standard pseudo-random number generators and lightweight heuristic code outside the model, then inserted into the prompt at inference time.
> > > > >
> > > > > We appreciate the reviewer's concern that steps (1) and (2) could be delegated to external modules, which we view as a natural comparison point for SSoT. We address this concern from two perspectives: (I) empirical comparison with seed injection and external tool calls, and (II) practical accessibility and engineering implications.
> > > > >
> > > > > ---
> > > > > ### Empirical comparison with seed injection and external tool calls
> > > > >
> > > > > To compare SSoT with such alternatives, we conducted experiments on the NoveltyBench (Curated) dataset using deepseek-r1-0528 and the following two baselines, which implement steps (1) and (2) outside the LLM while leaving step (3) to the model:
> > > > >
> > > > > 1. **Seed Injection**: For each query, we sample a fresh random string from an external PRNG, insert it into the user prompt, and instruct the LLM to use it when generating a diverse response.
> > > > > 1. **Tool Call**: We provide the LLM with two tools, `random_int` (to generate a random integer within a specified range) and `random_string` (to generate a random string of a specified length), and explicitly instruct the model to call these tools whenever it needs randomness.
> > > > >
> > > > > We then evaluated Distinct and Utility on NoveltyBench. The results are summarized below (full details in Appendix D.6):
> > > > >
> > > > > (Cells show Distinct (Utility); higher is better.)
> > > > > |Method|Creativity|Naming|Facts|Product Recs|Random|Opinions|Overall|
> > > > > |-|-|-|-|-|-|-|-|
> > > > > |Baseline|4.60 (5.61)|6.00 (6.13)|4.14 (5.35)|4.14 (6.02)|6.07 (5.10)|4.35 (4.08)|4.70 (5.17)|
> > > > > |Seed Injection|5.60 (5.90)|7.43 (6.46)|5.75 (6.47)|6.43 (**7.99**)|6.67 (5.41)|5.65 (4.13)|6.00 (5.76)|
> > > > > |Tool Call|5.60 (5.10)|7.00 (6.41)|5.39 (6.34)|**6.71** (7.72)|5.73 (4.75)|5.52 (3.60)|5.72 (5.33)|
> > > > > |**SSoT**|**5.90 (6.44)**|**7.57 (6.62)**|**6.04 (6.71)**|5.86 (6.63)|**6.87 (5.49)**|**5.87 (4.35)**|**6.19 (5.92)**|
> > > > >
> > > > > As shown, SSoT consistently outperforms Seed Injection and Tool Call in Distinct and Utility across most categories. Notably, on the "Creativity" task, Seed Injection yields only a modest Utility gain and Tool Call even lowers Utility, whereas SSoT achieves high scores in both metrics, demonstrating its ability to enhance diversity without compromising response quality.
> > > > >
> > > > > We attribute this performance gap to several factors:
> > > > > * **Limitations of Seed Injection**: Providing a single external string (one fresh seed per query) limits the model's ability to employ strategies that leverage randomness multiple times or for local decisions, as observed in our CoT analysis in Section 5.3.1. Furthermore, as shown in the rightmost panel of Figure 3 (Section 5.1.2), PIF performance with an injected seed remains consistently below that of SSoT, where the model internally generates and manipulates its own random string.
> > > > > * **Instability of Tool Calls**: While the Tool Call baseline, in principle, allows multiple requests for randomness, its performance proved unstable in our setting. The use of custom tools defined at inference time led to frequent tool-calling failures (e.g., incorrect function names or arguments), which degraded overall effectiveness.
> > > > >
> > > > > ---
> > > > > ### Practical accessibility and engineering implications
> > > > >
> > > > > Beyond these empirical advantages, where SSoT achieves strong quality and diversity without additional tuning, we also emphasize the practical accessibility and engineering implications.
> > > > >
> > > > > Since SSoT is implemented purely at the prompt level, it can be used directly from standard chat interfaces or instruction fields without any programming or infrastructure changes, and is therefore available to anyone who can issue prompts to the model. In contrast, approaches based on external modules are only available to users who can set up custom tools, manage an execution environment, and automate multi-turn interactions with the LLM.
> > > > >
> > > > > Even for such users with sufficient programming expertise, integrating tool calls into a pipeline introduces additional overhead: it requires maintaining the execution environment for the tools and necessitates multiple API calls (multi-turn) to the LLM, which complicates efficient use of batch APIs. By comparison, SSoT requires only a single prompt modification, works seamlessly with batch APIs, and offers the flexibility and ease of use required for immediate deployment.
> > > > >
> > > > > ---
> > > > > In conclusion, given its strong empirical performance and practicality as a pure prompting method, we believe SSoT offers distinct advantages over relying on external modules or tool calls for steps (1) and (2), improving both performance and usability compared to these alternatives.
> > > > >
> > > > > ---

---

> > > > > > ### Comment · Reviewer_KwFB · 2025-11-25
> > > > > >
> > > > > > Thank you to the authors for the considerable effort in expanding the experimental section. Appendix D.6 resolved my concerns, and I have no further questions.

---

> > > > > > > ### Author Response · Authors · 2025-11-29
> > > > > > > **Response to Reviewer KwFB**
> > > > > > >
> > > > > > > Dear Reviewer KwFB,
> > > > > > >
> > > > > > > We sincerely appreciate the additional time and effort you devoted to the discussion. We are grateful for your acknowledgement that your concerns have been addressed, as well as for your decision to raise the score above the acceptance threshold. Your insightful comments and the constructive exchange have significantly strengthened the manuscript.

---

### Official Review · Reviewer_kxrX · 2025-10-25

**Soundness:** 3
**Presentation:** 3
**Contribution:** 2
**Rating:** 4
**Confidence:** 3

**Summary:**

The paper proposes String Seed of Thought (SSoT), a prompt-only method for probability instruction following and open-ended diversity. The model is first prompted to generate a pseudo-random character string, and then leverages its reasoning ability to map that seed to the required output via simple rules (e.g., sum-mod/rolling hash). The authors provide theoretical guarantees of how self-correlated strings can be mapped to a near-uniform distribution with a deterministic hash mapping, and further show the distribution gap for the sum-mode strategy that LLM typically uses. Empirically, SSoT improves distribution faithfulness on coin-flip/k-of-n/RPS tasks and increases diversity on NoveltyBench. It contributes to the LLM applications that requires controllable stochasticity and diversity.

**Strengths:**

1. This method is simple yet effective. It works well across different LLMs and task types without tuning, demonstrating strong engineering practicality.
2.  This method is theoretically sound. It provides a rigorous bound showing that the TV distance between the empirical and target distribution decreases as the generated string length increases, even when the string exhibits autocorrelation. This result gives the method solid mathematical grounding. Moreover, the authors derived the bound for the sum-mod strategy that the model empirically adopts, offering a clear explanation of why the approach works.
3. The experiments are comprehensive and convincing, covering multiple application scenarios: closed-set probabilistic sampling and open-ended diverse generation. Multiple strong baselines are compared, and the method is tested on 5 advanced LLMs, all showing substantial improvements. This demonstrates its robustness across architectures and tasks. The paper also provides detailed supplementary materials for reproducibility.
4. Beyond reporting quantitative results, the authors perform an in-depth analysis of LLM’s reasoning behavior during generation. They discover that the models spontaneously develop strategies, such as sum-mod and rolling hash mappings, adaptively according to the task, to extract random actions from the generated strings. This is not only an interesting empirical observation but also deepens our understanding of LLM reasoning behaviors.

**Weaknesses:**

1. It would be better if the authors could provide some failure case analysis. In Table 1, the performance of QwQ-32B on the 2-choice task is even worse than the baseline. Is this degradation due to autocorrelation in the generated random strings, inappropriate mapping, or possible execution/calculation errors?
2. The models used in the experiments are all quite large. Considering that SSoT relies on the model itself to decide the mapping strategy and execute it, this raises doubts that the performance may be strongly limited by the capability (especially in reasoning) of the models. It also raises concerns about scalability to more complex PIF tasks, as it remains unclear whether LLMs can reliably come up with and implement more complex operations.
3. The intrinsic reasoning process of generating random strings and mapping is not transparent. Although the authors provide analysis of the models’ reasoning behavior, it remains unclear whether future versions of these models will change their strategies, or whether different models would adopt different strategies, which makes the algorithm unstable and difficult for users to control. This lack of transparency also weakens the connection between the authors’ theoretical analysis and practical implementation.
4. Considering the assumptions on the conditional distribution of the string in Theorems 4.1 and 4.2, it would be better to verify them quantitatively.

**Questions:**

1. I hope the authors could provide a more detailed introduction to the datasets for the DAG task and the related metrics. I checked the reference to understand how “distinct” and “utility” are calculated. It seems that distinct evaluates pure diversity, while utility combines both diversity and quality. However, I'm still unclear what the columns in Table 2, such as "creativity" and “naming”, represent. If they are subsets of the dataset, improvements in some tasks, including "creativity", would be valuable. However, I feel we shouldn't expect diversity increases in subsets like "facts", especially when the utility metric can't decompose the changes in quality.

---

> ### Author Response · Authors · 2025-11-21
> **Response to Reviewer kxrX (1/3)**
>
> ---
>
> We thank the reviewer for the positive assessment and the insightful questions. We have revised our manuscript accordingly.
>
> ---
>
> ## Weaknesses
>
> ---
> ### W1: Failure case analysis
> > It would be better if the authors could provide some failure case analysis. In Table 1, the performance of QwQ-32B on the 2-choice task is even worse than the baseline. Is this degradation due to autocorrelation in the generated random strings, inappropriate mapping, or possible execution/calculation errors?
>
> Thank you for the great suggestion. We agree with the reviewer that the failure case analysis is important, and we investigated the failure case of QwQ-32B on the 2-choice task.
>
> **Analysis**: We analyzed 1000 generated strings and responses. We found a significant bias: 947 out of 1000 strings started with the digit "7". Furthermore, in 50 responses, the model adopted an overly simplified strategy of determining the outcome based only on the first character's ASCII code. In 45 of those cases, the string started with "7", leading to a large skew (43 tails vs. 2 heads). Notably, excluding these cases recovers a near-ideal distribution, confirming that the issue lies in the specific strategy-bias interaction rather than the string quality alone.
>
> **Conclusion**: Failure occurs when two conditions meet: (1) the generated string has some form of bias (e.g., fixed first character), and (2) the model chooses a strategy that fails to extract entropy from the whole string (e.g., only using the first character). We have added this failure analysis to Appendix D.5 and elaboration on this point in the Limitations section (Section 7).
>
> ---
> ### W2: Dependence on LLM's reasoning capability
>
> > The models used in the experiments are all quite large. Considering that SSoT relies on the model itself to decide the mapping strategy and execute it, this raises doubts that the performance may be strongly limited by the capability (especially in reasoning) of the models. It also raises concerns about scalability to more complex PIF tasks, as it remains unclear whether LLMs can reliably come up with and implement more complex operations.
>
> We thank the reviewer for pointing out this critical dependency.
> The characteristic that effectiveness scales with model capability is consistent with other prompting techniques, such as Zero-shot Chain-of-Thought (CoT), which is known to yield gains primarily in larger models only (See Figure 3 of Kojima et al., 2022). Just as the limited performance of Zero-shot CoT on smaller models does not diminish its utility for frontier models, we believe SSoT remains a valuable technique for models that possess the necessary reasoning capability.
>
> To reveal SSoT's effectiveness and limitations on smaller LLMs, we conducted additional experiments on smaller LLMs, ranging from 1.5B to 8B parameters, including the Qwen3 family and reasoning-distilled models. The results are summarized below:
>
> (JS Divergence $\times 1000$; Lower is better)
> |Model|Method|2-choice|biased 2-choice|3-choice|biased 3-choice|biased 9-choice|
> |-|-|-|-|-|-|-|
> |Qwen3-8B|Baseline|16.92 ± 7.45|86.55 ± 16.61|66.46 ± 26.42|87.30 ± 14.99|293.72 ± 8.61|
> |Qwen3-8B|**SSoT**|7.36 ± 9.02 (↓56%)|24.16 ± 8.16 (↓72%)|7.72 ± 5.03 (↓88%)|14.51 ± 9.57 (↓83%)|60.23 ± 19.14 (↓79%)|
> |Qwen3-4B|Baseline|1.40 ± 0.92|117.28 ± 0.00|31.16 ± 13.32|115.53 ± 5.51|300.16 ± 0.00|
> |Qwen3-4B|**SSoT**|7.48 ± 4.11 (↑436%)|42.35 ± 15.55 (↓64%)|55.82 ± 8.24 (↑79%)|17.98 ± 8.44 (↓84%)|104.15 ± 19.51 (↓65%)|
> |Qwen3-1.7B|Baseline|20.82 ± 10.45|94.47 ± 18.26|305.56 ± 10.93|90.04 ± 5.07|300.16 ± 0.00|
> |Qwen3-1.7B|**SSoT**|106.70 ± 14.00 (↑413%)|10.92 ± 5.77 (↓88%)|259.41 ± 20.95 (↓15%)|14.17 ± 6.79 (↓84%)|229.36 ± 13.51 (↓24%)|
> |DeepSeek-R1-Distill-Llama-8B|Baseline|0.71 ± 0.79|75.06 ± 23.55|123.05 ± 24.88|49.76 ± 17.14|244.47 ± 25.87|
> |DeepSeek-R1-Distill-Llama-8B|**SSoT**|0.38 ± 0.62 (↓46%)|9.86 ± 5.58 (↓87%)|4.71 ± 4.83 (↓96%)|2.91 ± 1.73 (↓94%)|75.17 ± 17.80 (↓69%)|
> |Qwen3-thinking-4B| Baseline | 93.26 ± 19.56| 117.28 ± 0.00| 189.82 ± 19.14| 117.28 ± 0.00| 258.57 ± 23.93|
> |Qwen3-thinking-4B| **SSoT** | 10.85 ± 8.01 (↓88%)| 2.13 ± 3.25 (↓98%)| 5.01 ± 4.51 (↓97%)| 13.31 ± 6.96 (↓89%)| 31.07 ± 13.69 (↓88%)|
> |Nemotron-Qwen-1.5B|Baseline|128.18 ± 22.90|62.00 ± 16.67|258.94 ± 25.43|112.38 ± 7.94|221.08 ± 17.13|
> |Nemotron-Qwen-1.5B|**SSoT**|32.17 ± 12.24 (↓75%)|51.49 ± 16.17 (↓17%)|75.45 ± 21.07 (↓71%)|11.35 ± 6.48 (↓90%)|118.35 ± 17.27 (↓46%)|
> |PRNG|–|1.85 ± 2.58|1.93 ± 2.80|3.36 ± 2.48|2.85 ± 3.15|13.72 ± 4.21|
>
> **Analysis**: As shown in the table, while SSoT yields performance gains for the 8B model and reasoning-enhanced small models (e.g., DeepSeek-R1-Distill-Llama-8B), the improvements are inconsistent or absent for the smaller Qwen3 models (1.5B and 4B). This empirically suggests that sufficient reasoning capability is a prerequisite for SSoT.
>
> We have added these results to Appendix D.3 and explicitly listed "Dependence on Reasoning Capability" in Section 7 for clarification.
>
> ---

---

> > ### Author Response · Authors · 2025-11-21
> > **Response to Reviewer kxrX (2/3)**
> >
> > ---
> > ### W3: Transparency and stability of strategies
> >
> > > The intrinsic reasoning process of generating random strings and mapping is not transparent. Although the authors provide analysis of the models’ reasoning behavior, it remains unclear whether future versions of these models will change their strategies, or whether different models would adopt different strategies, which makes the algorithm unstable and difficult for users to control. This lack of transparency also weakens the connection between the authors’ theoretical analysis and practical implementation.
> >
> > We acknowledge the concern regarding the autonomous nature of the strategy selection. However, we believe SSoT remains robust and practical for the following reasons:
> >
> > 1. **Consistency with Prompting Paradigm**: Relying on autonomous reasoning is an inherent characteristic of advanced prompting techniques like Zero-shot Chain-of-Thought (CoT). The trade-off for this lack of transparency is the method's high versatility and zero-shot applicability.
> > 1. **Robustness of Converged Strategies**: Our analysis (Section 5.3.1) shows that models do not select arbitrary logic but converge on fundamental arithmetic operations (e.g., Sum-Mod, Rolling Hash). Since these are natural computational choices, it is highly probable that future, more capable models will continue to adopt and precisely follow those strategies.
> >
> > Furthermore, because the reasoning is explicit in the context window, the strategy is observable. If strict stability is required, users can easily steer the LLM by instructing it to use a specific successful strategy (e.g., "Use Sum-Mod") via the system prompt. We have added this discussion on steerability to the Limitations section.
> >
> > ---
> > ### W4: Quantitative verification of theoretical assumptions
> >
> > > Considering the assumptions on the conditional distribution of the string in Theorems 4.1 and 4.2, it would be better to verify them quantitatively.
> >
> > We appreciate this suggestion to bridge the gap between theory and practice. We conducted a statistical analysis of 5000 strings generated by deepseek-r1 and added the full results to Appendix E.
> >
> > **Summary of Findings**:
> >
> > 1. **$\delta$-SV Assumption**: The unigram frequency distribution decays linearly rather than following a power law (Figure 12), indicating no extreme mode collapse. This suggests the generated strings approximately satisfy the $\delta$-Santha-Vazirani condition required for Theorem 4.1.
> > 1. **Independence Assumption**: We measured the total variation distance between the joint and marginal distributions of ASCII codes for lags $n=1, 2, 3$. The resulting distances were negligible (all $< 0.07$, see Table 11), confirming that the numerical values used for the Sum-Mod strategy are effectively independent, satisfying the assumption for Theorem 4.2.
> >
> > Collectively, these analyses bridge the gap between theory and practice, confirming that the generated strings possess sufficient entropy and approximate independence to support the theoretical guarantees of SSoT.
> >
> > ---

---

> > > ### Author Response · Authors · 2025-11-21
> > > **Response to Reviewer kxrX (3/3)**
> > >
> > > ---
> > > ## Questions
> > >
> > > ---
> > > ### Q1: NoveltyBench metrics and categories.
> > >
> > > > I hope the authors could provide a more detailed introduction to the datasets for the DAG task and the related metrics. I checked the reference to understand how “distinct” and “utility” are calculated. It seems that distinct evaluates pure diversity, while utility combines both diversity and quality. However, I'm still unclear what the columns in Table 2, such as "creativity" and “naming”, represent. If they are subsets of the dataset, improvements in some tasks, including "creativity", would be valuable. However, I feel we shouldn't expect diversity increases in subsets like "facts", especially when the utility metric can't decompose the changes in quality.
> > >
> > > We thank the reviewer for carefully checking the reference and raising this important question regarding the metrics and dataset categories.
> > >
> > > In summary, the reviewer correctly identifies the nature of the metrics and creative tasks. However, regarding the "Facts" category, we clarify that these tasks consist of questions with multiple valid answers (unlike single-answer QA). Consequently, increasing diversity in "Facts" directly contributes to higher Utility.
> > >
> > > We clarify the details below:
> > >
> > > **Metrics**
> > >
> > > As the reviewer noted, Distinct measures pure diversity (number of unique equivalence classes), while Utility combines both diversity and quality. Crucially, Utility is calculated by summing the reward scores of only the novel generations. Therefore, simply generating high-quality but repetitive answers will result in a low Utility score. High Utility requires the model to generate diverse and high-quality responses simultaneously.
> > >
> > > **Categories and SSoT's Impact**
> > >
> > > * **Creativity & Naming (Agreement)**: "Creativity" (e.g., "Write a short love poem with 4 lines.") and "Naming" (e.g., "Name the dog character in my story about a loyal dog.") are inherently open-ended. The performance improvement SSoT achieves here (Table 2) confirms its strength in exploring creative spaces.
> > > * **Facts (Clarification)**: This category contains queries where multiple valid answers exist, but the user asks for one instance (e.g., "What is a cool fact about Pittsburgh?"), and does not consist of single-answer questions (e.g., "What is the capital of France?"). In this context, if a baseline model repeats the same "cool fact" 10 times, its Utility is low. If SSoT generates 10 different but correct facts, its Utility increases significantly. Thus, SSoT is valuable even for factual tasks when the goal is to cover a broader range of knowledge.
> > >
> > > To resolve this ambiguity, we have added a detailed description of the NoveltyBench categories and metrics to Appendix F.
> > >
> > > ---

---

> > > > ### Author Response · Authors · 2025-11-28
> > > > **Additional Response to Reviewer kxrX**
> > > >
> > > > Dear Reviewer kxrX,
> > > >
> > > > Thank you very much for your thoughtful and constructive feedback. We have substantially revised the manuscript in response to your comments and those of the other reviewers, and we believe the paper has significantly improved.
> > > >
> > > > As your current rating is slightly below the acceptance threshold, we would be grateful if you could let us know whether the revised manuscript and our responses adequately address your concerns. If there are any remaining issues, ambiguities, or points where further clarification would help, we would be very keen to address them before the discussion period ends.
> > > >
> > > > Building on Reviewer KwFB’s feedback, we conducted additional experiments comparing SSoT with baselines using external random modules. Reviewer KwFB explicitly acknowledged these improvements by raising their evaluation score. Given the importance of these results in empirically demonstrating the advantages of SSoT, we share the relevant part of our response below:
> > > >
> > > > ---
> > > > ### Empirical comparison with seed injection and external tool calls
> > > >
> > > > To compare SSoT with alternatives that delegate randomness generation to external modules, we conducted experiments on the NoveltyBench (Curated) dataset using deepseek-r1-0528 and the following two baselines, which implement steps (1) and (2) outside the LLM while leaving step (3) to the model:
> > > >
> > > > 1. **Seed Injection**: For each query, we sample a fresh random string from an external PRNG, insert it into the user prompt, and instruct the LLM to use it when generating a diverse response.
> > > > 1. **Tool Call**: We provide the LLM with two tools, `random_int` (to generate a random integer within a specified range) and `random_string` (to generate a random string of a specified length), and explicitly instruct the model to call these tools whenever it needs randomness.
> > > >
> > > > We then evaluated Distinct and Utility on NoveltyBench. The results are summarized below (full details in Appendix D.6):
> > > >
> > > > (Cells show Distinct (Utility); higher is better.)
> > > > |Method|Creativity|Naming|Facts|Product Recs|Random|Opinions|Overall|
> > > > |-|-|-|-|-|-|-|-|
> > > > |Baseline|4.60 (5.61)|6.00 (6.13)|4.14 (5.35)|4.14 (6.02)|6.07 (5.10)|4.35 (4.08)|4.70 (5.17)|
> > > > |Seed Injection|5.60 (5.90)|7.43 (6.46)|5.75 (6.47)|6.43 (**7.99**)|6.67 (5.41)|5.65 (4.13)|6.00 (5.76)|
> > > > |Tool Call|5.60 (5.10)|7.00 (6.41)|5.39 (6.34)|**6.71** (7.72)|5.73 (4.75)|5.52 (3.60)|5.72 (5.33)|
> > > > |**SSoT**|**5.90 (6.44)**|**7.57 (6.62)**|**6.04 (6.71)**|5.86 (6.63)|**6.87 (5.49)**|**5.87 (4.35)**|**6.19 (5.92)**|
> > > >
> > > > As shown, SSoT consistently outperforms Seed Injection and Tool Call in Distinct and Utility across most categories. In particular, on the "Creativity" task, Seed Injection yields only a modest Utility gain and Tool Call even lowers Utility, whereas SSoT achieves high scores in both metrics, demonstrating its ability to enhance diversity without compromising response quality.
> > > >
> > > > We attribute this performance gap to several factors:
> > > > * **Limitations of Seed Injection**: Providing a single external string (one fresh seed per query) limits the model's ability to employ strategies that leverage randomness multiple times or for local decisions, as observed in our CoT analysis in Section 5.3.1. Furthermore, as shown in the rightmost panel of Figure 3 (Section 5.1.2), PIF performance with an injected seed remains consistently below that of SSoT, where the model internally generates and manipulates its own random string.
> > > > * **Instability of Tool Calls**: While the Tool Call baseline, in principle, allows multiple requests for randomness, its performance proved unstable in our setting. The use of custom tools defined at inference time led to frequent tool-calling failures (e.g., incorrect function names or arguments), which degraded overall effectiveness.
> > > >
> > > > ---
> > > > ### Practical accessibility and engineering implications
> > > >
> > > > Beyond these empirical advantages, where SSoT achieves strong quality and diversity without additional tuning, we also emphasize the practical accessibility and engineering implications.
> > > >
> > > > Since SSoT is implemented purely at the prompt level, it can be used directly from standard chat interfaces or instruction fields without any programming or infrastructure changes, and is therefore available to anyone who can issue prompts to the model. In contrast, approaches based on external modules are only available to users who can set up custom tools, manage an execution environment, and automate multi-turn interactions with the LLM.
> > > >
> > > > Even for such users with sufficient programming expertise, integrating tool calls into a pipeline introduces additional overhead: it requires maintaining the execution environment for the tools and necessitates multiple API calls (multi-turn) to the LLM, which complicates efficient use of batch APIs. By comparison, SSoT requires only a single prompt modification, works seamlessly with batch APIs, and offers the flexibility and ease of use required for immediate deployment.
> > > >
> > > > ---

---

> > > > > ### Comment · Reviewer_kxrX · 2025-11-28
> > > > >
> > > > > I thank the authors for providing the additional experiments, which effectively addressed my concerns. I would be very happy to increase my score. Unfortunately, due to recent technical issues on OpenReview and the consequences, the system no longer allows me to modify my score. I sincerely regret this situation, but I would be glad to update my score accordingly if an opportunity becomes available later.

---

> > > > > > ### Author Response · Authors · 2025-11-29
> > > > > > **Response to Reviewer kxrX**
> > > > > >
> > > > > > Dear Reviewer kxrX,
> > > > > >
> > > > > > We sincerely thank you for dedicating your time and effort to the additional discussion. We deeply appreciate your acknowledgement that your concerns have been addressed, as well as your decision to raise the score above the acceptance threshold. We are pleased that the manuscript has been significantly improved thanks to your insightful comments and our constructive discussion.

---

### Official Review · Reviewer_yb85 · 2025-10-28

**Soundness:** 2
**Presentation:** 3
**Contribution:** 3
**Rating:** 6
**Confidence:** 3

**Summary:**

String Seed of Thought (SSoT) is a proposed prompting method that injects a random string as a “seed” into an LLM’s chain-of-thought to induce more faithful probabilistic behavior and greater output diversity. The technique works by first asking the model to generate a random string, then having the model algorithmically manipulate that string to select a final answer according to a target probability distribution. The authors provide both theoretical guarantees (showing that with sufficiently long random strings the output distribution can approach the desired distribution) and extensive empirical evidence that SSoT yields significantly improved distribution alignment and more diverse responses across multiple models and tasks.

**Strengths:**

1. The SSoT prompting strategy is conceptually simple to implement (just adding a brief instruction to generate and use a random string) and is applicable to a wide range of LLMs without any model modifications.

2. SSoT dramatically improves an LLM’s ability to follow probabilistic instructions, achieving empirical sampling frequencies very close to the target probabilities and also boosts the diversity of open-ended generations, outperforming other diversity-promoting baselines like prompt paraphrasing or higher-temperature sampling while maintaining output quality. Experiments across five state-of-the-art LLMs and various scenarios consistently show that SSoT yields superior distribution-faithful performance and surpasses strong prompting baselines in both accuracy of sampling and response diversity.

3. The paper provides a solid theoretical foundation, proving (informally) that as the length of the internally generated string increases, the total variation distance between the model’s output distribution and the target distribution diminishes even if the string characters are not fully independent.

**Weaknesses:**

1. This method may cause potential errors and hurt answer quality. While the method focuses on matching distributions and diversity, the paper provides little discussion on whether the use of SSoT could inadvertently affect the correctness or factuality of outputs in tasks where a specific correct answer is expected.

2. Most experiments involve tasks with a small, discrete set of outcomes (binary or a few categories), so it remains uncertain how well SSoT would scale to more complex distributions or tasks with a larger set of possible outputs (or continuous output spaces).

3. The theoretical guarantees rest on assumptions like sufficiently long random strings and bounded bias in character generation; in practice, these conditions cannot be strictly met, so the gap between the idealized guarantees and real model behavior is not fully understood.

**Questions:**

1. How sensitive is SSoT’s effectiveness to the length of the random string used, and how can we provide for choosing an appropriate string length to balance randomness and efficiency in practice?

2. Have the authors tested SSoT on tasks with highly skewed or arbitrary target distributions (beyond uniform cases like fair coin flips), and if so, how well does the method maintain the correct proportions in those more challenging scenarios?

3. Does the inclusion of a random string (and its manipulation) ever degrade the perceived quality or coherence of the final answer, and how can one ensure that this random seed is hidden or handled so that it doesn’t confuse end-users in a real application?

---

> ### Author Response · Authors · 2025-11-21
> **Response to Reviewer yb85 (1/2)**
>
> ---
> We thank the reviewer for their detailed and constructive feedback. We have revised our manuscript accordingly.
>
> ---
> ## Weaknesses
>
> ---
> ### W1: Potential impact on correctness and factuality
> > This method may cause potential errors and hurt answer quality. While the method focuses on matching distributions and diversity, the paper provides little discussion on whether the use of SSoT could inadvertently affect the correctness or factuality of outputs in tasks where a specific correct answer is expected.
>
> Thank you for raising this important concern. Regarding the impact on answer quality, we would like to highlight that the Utility metric used in our NoveltyBench experiments explicitly incorporates response quality via a reward model. As shown in Table 2, SSoT improves Utility, indicating that it enhances diversity without sacrificing the model's ability to generate correct or high-quality answers.
>
> However, we acknowledge that for tasks with exactly one correct answer (e.g., math tasks), inducing diversity is not effective and potentially harmful. We have explicitly added this point to the Limitations in Section 7 of the revised manuscript.
>
> ---
> ### W2: Scaling to complex distributions and continuous output spaces
> > Most experiments involve tasks with a small, discrete set of outcomes (binary or a few categories), so it remains uncertain how well SSoT would scale to more complex distributions or tasks with a larger set of possible outputs (or continuous output spaces).
>
> Thank you for this constructive suggestion. First, we would like to highlight that, regarding complex output spaces, NoveltyBench includes creative writing tasks where the output space is effectively unbounded and not a small, predefined discrete option set. Tables 2 and 3 demonstrate SSoT's effectiveness in this domain.
>
> Secondly, to address your concern regarding continuous distributions, we conducted additional experiments on three continuous distributions: Uniform $U([0, 1])$, Normal $\mathcal{N}(0, 1)$, and Beta $B(2, 5)$. We evaluated the goodness-of-fit using Kolmogorov-Smirnov and Cramér–von Mises statistics (lower is better).
>
> | Distribution | Method| Kolmogorov–Smirnov| Cramér–von Mises|
> |-------------|----------|------------------------------|-----------------------------|
> |Uniform| Baseline | 0.220 ± 0.023| 0.999 ± 0.163|
> |Uniform| SSoT| **0.187 ± 0.034 (↓15%)**| **0.839 ± 0.405 (↓16%)**|
> |Normal| Baseline| 0.129 ± 0.022| 0.358 ± 0.146|
> | Normal| SSoT| **0.108 ± 0.026 (↓16%)**| **0.310 ± 0.195 (↓14%)**|
> | Beta| Baseline | 0.177 ± 0.039| 0.798 ± 0.271|
> | Beta| SSoT| **0.113 ± 0.040 (↓36%)**| **0.293 ± 0.208 (↓63%)**|
>
> As shown above, SSoT consistently improves the goodness-of-fit metrics for continuous distributions, showing more faithful sampling from the target distribution. We have added the details of this experiment in Appendix D.1.
>
> ---
> ### W3: Gap between theoretical assumptions and practice
>
> > The theoretical guarantees rest on assumptions like sufficiently long random strings and bounded bias in character generation; in practice, these conditions cannot be strictly met, so the gap between the idealized guarantees and real model behavior is not fully understood.
>
> We appreciate the reviewer pointing out this gap between theory and practice. To bridge this, we conducted a quantitative analysis of the generated strings in Appendix E. We summarize the key findings below.
>
> Regarding the general bias of the generated strings, we observed that the unigram frequency decays linearly and gradually rather than following a steep power-law (Figure 12), albeit not perfectly uniform. This confirms that the generation avoids extreme mode collapse, ensuring a sufficient baseline level of entropy.
>
> Additionally, our analysis yields two key findings that support the applicability of our theorems:
> 1. **$\delta$-SV Assumption (Theorem 4.1)**: Since the $\delta$-Santha-Vazirani source is defined by bounds on conditional probabilities, we analyzed the bigram distributions $P(s|c)$. As shown in Figure 11 in Appendix E, the upper bound condition is well-satisfied, and the lower bound condition approximately holds.
> 1. **Independence Assumption (Theorem 4.2)**: In the context of the Sum-Mod strategy, we verified the independence of the ASCII values modulo $m=2,3,9$. We found that the total variation distance between the joint distribution and the product of marginals is negligible (Table 11 in Appendix E). This confirms that, despite the autoregressive nature of the text, the numerical values used for modular arithmetic can be treated as approximately independent.
>
> These empirical results suggest that our theoretical bounds provide valid insights into the method's effectiveness, even under practical conditions.
>
> ---

---

> > ### Author Response · Authors · 2025-11-21
> > **Response to Reviewer yb85 (2/2)**
> >
> > ---
> >
> > ## Questions
> >
> > ---
> > ### Q1: Sensitivity to random string length
> > > How sensitive is SSoT’s effectiveness to the length of the random string used, and how can we provide for choosing an appropriate string length to balance randomness and efficiency in practice?
> >
> > We thank the reviewer for raising this question. To address this, we performed an additional study varying the target string length $n$ specified in the prompt for a PIF task. The result is shown below:
> >
> > | n   | Generated String Length | JS Divergence         | KL Divergence         | TV Distance           |
> > |-----|-------------------------|-----------------------|-----------------------|-----------------------|
> > | 2   | 2.0| 0.178 ± 0.022| 0.705 ± 0.086| 0.527 ± 0.029|
> > | 4   | 4.2| 0.228 ± 0.017| 0.887 ± 0.071| 0.581 ± 0.021|
> > | 8   | 8.0| 0.138 ± 0.033| 0.581 ± 0.128| 0.446 ± 0.058|
> > | 16  | 16.2| 0.037 ± 0.011| 0.156 ± 0.050| 0.222 ± 0.040|
> > | 24  | 25.9| **0.019 ± 0.007**| **0.075 ± 0.027**| **0.141 ± 0.034**|
> > | 32  | 32.4| 0.025 ± 0.007| 0.092 ± 0.026| 0.151 ± 0.030|
> > | 40  | 40.0| 0.032 ± 0.010| 0.116 ± 0.032| 0.191 ± 0.028|
> > | 48  | 50.7| 0.030 ± 0.011| 0.108 ± 0.039| 0.179 ± 0.035|
> > | 64  | 69.6| 0.044 ± 0.011| 0.162 ± 0.036| 0.244 ± 0.029|
> > | 128 | 142.1| 0.052 ± 0.016| 0.187 ± 0.054| 0.264 ± 0.035|
> >
> > **Key findings**: The generated string lengths align well with the specified target $n$, indicating that the model follows the length constraint. Performance is poor at very short lengths ($n\leq 8$) but improves dramatically as length increases, peaking around $n=24$. Beyond this point, performance degrades slightly. We attribute this degradation to the increased likelihood of arithmetic errors by the LLM when performing operations (like sum-mod) on very long strings.
> >
> > This additional experiment supports our argument that a longer string length is important for better performance in PIF. We added the details of this experiment in Appendix D.2.
> >
> > ---
> > ### Q2: Performance on highly skewed distributions
> >
> > > Have the authors tested SSoT on tasks with highly skewed or arbitrary target distributions (beyond uniform cases like fair coin flips), and if so, how well does the method maintain the correct proportions in those more challenging scenarios?
> >
> > We thank the reviewer for the question. We would like to highlight that the "Biased PIF" settings in Section 5.1.2 represent highly skewed cases. As shown in Figure 3, SSoT maintains near-ideal performance even in these skewed scenarios. This is a notable capability, as LLMs must not only diversify outputs but do so with precision to match the target skew in those settings.
> >
> > To clarify this point, we added the following sentence to Section 5.1.2.
> >
> > Biased PIF presents a greater challenge than unbiased PIF. This is because it requires not only mitigating mode collapse to achieve uniformity, but also necessitates the reasoning ability to derive appropriate arithmetic operations to sample from the target distribution and the computational capability to execute them.
> >
> > ---
> > ### Q3: Impact on perceived quality and handling the random seed in production.
> >
> > > Does the inclusion of a random string (and its manipulation) ever degrade the perceived quality or coherence of the final answer, and how can one ensure that this random seed is hidden or handled so that it doesn’t confuse end-users in a real application?
> >
> > We thank the reviewer for raising this question.
> >
> > **Impact on Quality**: For open-ended and probabilistic tasks, our results on NoveltyBench (specifically the Utility metric, which penalizes low-quality responses) demonstrate that SSoT enhances diversity without degrading the coherence or quality of the final answer. However, we acknowledge that for tasks requiring a single, deterministic answer (e.g., math tasks), forcing a generation and manipulation of a random string is not effective and could theoretically distract the model, potentially degrading performance. We have explicitly added this point to the newly created Limitations section (Section 7) to clarify the method's appropriate scope.
> >
> > **Handling for End-Users**: Regarding the user experience, our SSoT prompt is designed with a clear three-stage structure: (1) random string generation, (2) manipulation of it, and (3) generation of a final answer. In real-world applications, the system can easily parse and display only the content in the third part of the whole response. This is conceptually similar to how deployed reasoning models are typically used in practice, where their thinking content is hidden and only the final answer is shown to end-users, providing a clean and confusion-free experience.
> >
> > ---

---

> > > ### Author Response · Authors · 2025-11-28
> > > **Additional Response to Reviewer yb85**
> > >
> > > Dear Reviewer yb85,
> > >
> > > Thank you again for your thoughtful and constructive feedback. In response to your comments and those of the other reviewers, we have substantially revised the manuscript, and we believe the paper has significantly improved.
> > >
> > > As the discussion period is coming to a close, we would greatly appreciate it if you could let us know whether the revised manuscript and our responses adequately address your concerns. If there are any remaining issues or points that are still unclear, we are keen to clarify them or provide any additional analysis.
> > >
> > > Building on Reviewer KwFB’s feedback, we conducted additional experiments comparing SSoT with baselines using external random modules. Reviewer KwFB explicitly acknowledged these improvements by raising their evaluation score. Given the importance of these results in empirically demonstrating the advantages of SSoT, we share our response below:
> > >
> > > ---
> > > ### Empirical comparison with seed injection and external tool calls
> > >
> > > To compare SSoT with alternatives that delegate randomness generation to external modules, we conducted experiments on the NoveltyBench (Curated) dataset using deepseek-r1-0528 and the following two baselines, which implement steps (1) and (2) outside the LLM while leaving step (3) to the model:
> > >
> > > 1. **Seed Injection**: For each query, we sample a fresh random string from an external PRNG, insert it into the user prompt, and instruct the LLM to use it when generating a diverse response.
> > > 1. **Tool Call**: We provide the LLM with two tools, `random_int` (to generate a random integer within a specified range) and `random_string` (to generate a random string of a specified length), and explicitly instruct the model to call these tools whenever it needs randomness.
> > >
> > > We then evaluated Distinct and Utility on NoveltyBench. The results are summarized below (full details in Appendix D.6):
> > >
> > > (Cells show Distinct (Utility); higher is better.)
> > > |Method|Creativity|Naming|Facts|Product Recs|Random|Opinions|Overall|
> > > |-|-|-|-|-|-|-|-|
> > > |Baseline|4.60 (5.61)|6.00 (6.13)|4.14 (5.35)|4.14 (6.02)|6.07 (5.10)|4.35 (4.08)|4.70 (5.17)|
> > > |Seed Injection|5.60 (5.90)|7.43 (6.46)|5.75 (6.47)|6.43 (**7.99**)|6.67 (5.41)|5.65 (4.13)|6.00 (5.76)|
> > > |Tool Call|5.60 (5.10)|7.00 (6.41)|5.39 (6.34)|**6.71** (7.72)|5.73 (4.75)|5.52 (3.60)|5.72 (5.33)|
> > > |**SSoT**|**5.90 (6.44)**|**7.57 (6.62)**|**6.04 (6.71)**|5.86 (6.63)|**6.87 (5.49)**|**5.87 (4.35)**|**6.19 (5.92)**|
> > >
> > > As shown, SSoT consistently outperforms Seed Injection and Tool Call in Distinct and Utility across most categories. In particular, on the "Creativity" task, Seed Injection yields only a modest Utility gain and Tool Call even lowers Utility, whereas SSoT achieves high scores in both metrics, demonstrating its ability to enhance diversity without compromising response quality.
> > >
> > > We attribute this performance gap to several factors:
> > > * **Limitations of Seed Injection**: Providing a single external string (one fresh seed per query) limits the model's ability to employ strategies that leverage randomness multiple times or for local decisions, as observed in our CoT analysis in Section 5.3.1. Furthermore, as shown in the rightmost panel of Figure 3 (Section 5.1.2), PIF performance with an injected seed remains consistently below that of SSoT, where the model internally generates and manipulates its own random string.
> > > * **Instability of Tool Calls**: While the Tool Call baseline, in principle, allows multiple requests for randomness, its performance proved unstable in our setting. The use of custom tools defined at inference time led to frequent tool-calling failures (e.g., incorrect function names or arguments), which degraded overall effectiveness.
> > >
> > > ---
> > > ### Practical accessibility and engineering implications
> > >
> > > Beyond these empirical advantages, where SSoT achieves strong quality and diversity without additional tuning, we also emphasize the practical accessibility and engineering implications.
> > >
> > > Since SSoT is implemented purely at the prompt level, it can be used directly from standard chat interfaces or instruction fields without any programming or infrastructure changes, and is therefore available to anyone who can issue prompts to the model. In contrast, approaches based on external modules are only available to users who can set up custom tools, manage an execution environment, and automate multi-turn interactions with the LLM.
> > >
> > > Even for such users with sufficient programming expertise, integrating tool calls into a pipeline introduces additional overhead: it requires maintaining the execution environment for the tools and necessitates multiple API calls (multi-turn) to the LLM, which complicates efficient use of batch APIs. By comparison, SSoT requires only a single prompt modification, works seamlessly with batch APIs, and offers the flexibility and ease of use required for immediate deployment.
> > >
> > > ---

---

### Author Response · Authors · 2025-11-21
**General response to all reviewers**

---
We thank all the reviewers for their time and effort in reviewing our manuscript and providing constructive feedback and suggestions. We have revised our manuscript to reflect these comments and believe the quality of the manuscript has been significantly improved.

Before addressing specific comments, we would like to clarify a critical aspect regarding the scope of our method, particularly concerning Diversity-Aware Generation (DAG).

---

**TLDR**: SSoT is not limited to closed-set sampling (like coin flips); it is highly effective for open-ended creative tasks where external random modules cannot be applied.

We utilized NoveltyBench as a primary benchmark for DAG. Notably, this benchmark includes open-ended creative writing tasks (e.g., "Write a short fable about a lemur and a light bulb"). As shown in Table 2 in our manuscript, SSoT significantly improved diversity in these creative tasks. However, increasing diversity alone is insufficient if response quality degrades. The Utility metric in NoveltyBench quantifies this trade-off by incorporating both diversity and quality (measured by a reward model). Our results show that SSoT improves Utility alongside diversity, demonstrating that it generates diverse outputs without compromising quality.

This capability highlights a key practical advantage of SSoT over external randomization methods. In open-ended tasks, such as free-form creative writing or questions with multiple valid answers but no fixed option set (e.g., "Tell me a cool fact about Pittsburgh"), it is not sufficient to rely solely on external random modules, because the candidate set is not predefined. In contrast, SSoT can generate a diverse answer by internally generating randomness and integrating it directly into the reasoning process. We have added a detailed explanation of NoveltyBench and this distinction in Appendix F to clarify the method's versatility.

---

---

### Author Response · Authors · 2025-12-04
**Summary for Area Chair**

Dear Area Chair,

We provide a concise summary of our rebuttal progress to help the AC understand the status.

---
## **Overview of Reviewer Consensus**

We have comprehensively addressed all questions and weaknesses raised by the reviewers through extensive additional experiments and discussions. The discussion phase concluded with a positive consensus, where every reviewer confirmed their support for acceptance:

* **Reviewer yb85: 6** — Positive initial assessment; no outstanding concerns.
* **Reviewer kxrX: 4 → 6** — Confirmed the decision to raise the score to 6, though the update was blocked by the administrative freeze on reviews.
* **Reviewer KwFB: 2 → 6** — Concerns regarding motivation and baselines were fully resolved through a constructive dialogue, leading to a decision to raise the score twice (2 → 4 → 6).

---
## **Summary of Key Issues and Resolutions**

Below is a summary of the major concerns raised and how we successfully addressed them.

### **Advantage and Necessity over External Tools (Reviewer KwFB)**

**Issue:** Whether external pseudo-random number generators (PRNGs) would suffice.

**Resolution:** We clarified that external tools are inapplicable to open-ended creative tasks where options are undefined, and we revised the manuscript to make this distinction clearer. Furthermore, we conducted comparative experiments with "Seed Injection" and "Tool Call" baselines, demonstrating that SSoT outperforms them in both diversity and quality.

### **Gap between Theoretical Assumptions and Practice (All Reviewers)**

**Issue:** Whether autoregressive LLM generation satisfies the theoretical randomness assumptions of our method.

**Resolution:** We conducted a large-scale statistical analysis of the generated strings. We quantitatively proved that the strings approximately satisfy the conditions of our theorems ($\delta$-Santha-Vazirani source) and possess sufficient entropy and approximate independence.

### **Scope of Applicability and Clarification of Limitations (Reviewer yb85, kxrX)**

**Issue:** Applicability to complex probability distributions and dependence on model size.

**Resolution:** We demonstrated effectiveness on continuous distributions (e.g., Normal distribution) through additional experiments. Additionally, we conducted experiments across various model sizes to elucidate how SSoT's effectiveness depends on the model's reasoning capability, and explicitly stated this boundary in the Limitations section.

---

We hope this summary assists in your final assessment. Thank you for your time and coordination throughout the review process.

Sincerely,

Authors of Submission 8824

---

### Meta-Review · Area_Chair_Li2t · 2025-12-21

**Summary:**

The idea of this work is simple, broadly applicable, and requires no model modifications. Reviewers agree that the paper offers solid theoretical intuition and strong empirical results. The behavioral analysis showing how models internally adopt sum-mod or rolling-hash schemes is also interesting. However, reviewers raise concerns about practical robustness, theoretical assumptions, and generality. The method relies on LLM-generated strings that are not truly random, and the theory assumes independence-like properties that do not strictly hold. The gap between the idealized guarantees and actual model behavior is not fully addressed. Experiments focus on small outcome spaces, leaving scalability to more complex distributions uncertain. SSoT can also degrade answer quality in some models, yet failure modes are not analyzed. Prompt design lacks consistency, raising questions about generalization. Some reviewers also find the motivation unclear when external randomness could achieve exact sampling.

**Reviewer Concerns:**

Authors largely expanded the experimental section and successfully addressed all the major concerns from reviewers.

**Reviewer Scores:**

6,6,6

---

### Decision · Program_Chairs · 2026-01-26

Accept (Poster)